# Memristor-based feature learning for pattern classification

Tuo Shi [1], Lili Gao[1], Yang Tian[1], Shuangzhu Tang[1], Jinchang Liu[1], Yiqi Li[1], Ruixi Zhou[1], Shiyu Cui[1], Hui Zhang[1], Yu Li [2], Zuheng Wu[3], Xumeng Zhang [2], Taihao Li[1], Xiaobing Yan [4] ✉ & Qi Liu [2] ✉

Inspired by biological processes, feature learning techniques, such as deep learning, have achieved great success in various fields. However, since biological organs may operate differently from semiconductor devices, deep models usually require dedicated hardware and are computation-complex. High energy consumption has made deep model growth unsustainable. We present an approach that directly implements feature learning using semiconductor physics to minimize disparity between model and hardware. Following this approach, a feature learning technique based on memristor drift-diffusion kinetics is proposed by leveraging the dynamic response of a single memristor to learn features. The model parameters and computational operations of the kinetics-based network are reduced by up to 2 and 4 orders of magnitude, respectively, compared with deep models. We experimentally implement the proposed network on 180 nm memristor chips for various dimensional pattern classification tasks. Compared with memristor-based deep learning hardware, the memristor kinetics-based hardware can further reduce energy and area consumption significantly. We propose that innovations in hardware physics could create an intriguing solution for intelligent models by balancing model complexity and performance.

In recent decades, artificial intelligence (AI) has significantly advanced and has had a profound impact on human life. Feature learning, also known as representation learning, enables machines to automatically discover appropriate features from raw data[1]. Inspired by the discoveries in neuroscience[2], deep neural network (DNN) have emerged as the most effective feature learning method for pattern classification. However, current feature learning techniques have very high demands for computing power and data storage, resulting in significant energy consumption. In some scenarios, such as edge-computing, these demands can hardly be satisfied.

The implementation of neuroscience-inspired feature learning techniques using existing computing hardware is often accompanied by significant hardware-cost and energy consumption, as these neural

systems may operate differently from computing hardware built upon hardware physics. Other than the above approach, researchers have found success in incorporating ideas from physics into AI, demonstrated in works like simulated annealing[3], and Hamiltonian neural networks[4]. While these techniques are largely "inspired by physics", they are not "compute-with-physics" or physics-based, still facing issues with hardware and energy costs. For example, the simulated annealing usually involves compute-intensive processes inside a highly interconnected interaction network and stochastic search algorithms that necessitate random number generation with an exponentially decaying probability distribution[5]. While the Hamiltonian neural network computes frequently the partial differential equations (PDEs) of the Hamiltonian function with respect to momentum vectors. The

[1]Zhejiang Laboratory, Hangzhou 311122, China. [2]Frontier Institute of Chip and System, Fudan University, Shanghai 200433, China. [3]School of Integrated Circuits, Anhui University, Hefei 230601, China. [4]Key Laboratory of Brain-like Neuromorphic Devices and Systems of Hebei Province, Hebei University, Baoding 071002, China. ✉e-mail: yanxiaobing@ime.ac.cn; qi_liu@fudan.edu.cn

solving of PDEs often requires the use of expensive numerical solvers[6]. Despite the potential of emerging in-memory computing approaches to enhance computational efficiency significantly, they remain constrained by limited memory capacity[7] and incompatible algorithms[8]. A physics-based feature learning method that is computationally efficient remains elusive.

Memristors are being explored for their potential to revolutionize data storage and processing technologies. In the past few years, the switching endurance, data retention time, energy consumption, switching time, integration density, and price of memristive non-volatile memories has been remarkably improved (depending on the materials used, values up to ~$10^{15}$ cycles, >10 years, ~0.1 pJ, ~10 ns, 256 gigabits per die, and ≤$0.30 per gigabit have been achieved)[9]. Resistive Random Access Memory (RRAM), leveraging its non-volatility, multi-bit storage, and Complementary Metal-Oxide-Semiconductor Transistor (CMOS) compatibility, emerges as one of the most promising contenders for next-generation non-volatile memory (NVM). Recent advancements have seen major semiconductor manufacturers, including Taiwan Semiconductor Manufacturing Company (TSMC) and Semiconductor Manufacturing International Corporation (SMIC), successfully establish commercial RRAM production lines. TSMC has achieved mass production of 40 nm node chips and is currently producing embedded RRAM with 28 nm and 22 nm nodes[10,11]. Additionally, TSMC has also implemented 32 Mb RRAM with 12 nm ultra-low power FinFET technology[12]. Meanwhile, companies like Intel, Fujitsu, Infineon, and Panasonic have actively developed embedded RRAM products, showcasing significant progress in technology and increasing commercial viability[13–16]. Fujitsu officially announced the launch of a 12 Mb RRAM chip, targeting its application in hearing aids and wearable devices like watches[14]. Infineon's AURIX™ TC4x series introducing RRAM as the NVM for automotive applications in 2024[16]. These developments underscore the rapid progress in RRAM technology and its increasing commercial viability across various applications.

Memristor is widely recognized as a promising "compute-with-physics" device that directly implements matrix-vector multiplication (MVM) using physical laws, namely Ohm's law for multiplication and Kirchhoff's law for summation[9]. As MVM is the most frequently used operation in deep learning, this implementation has resulted in greatly improved energy efficiency[17–28]. However, since memristors are tailored to fit these algorithms, problems like large number of parameters and computation operations still exist. Moreover, current AI models mostly rely on the thermodynamic equilibrium states of devices, like the binary states of transistors and the multiple conductance states of memristors, to represent information. The device kinetic processes between thermodynamic steady states can yield additional time dimensional information under the same energy input, leading to enhanced information gain per energy consumed. The current use of memristors as variable resistors only scratches the surface of their full potential as their abundant kinetics like coupled ionic and electronic migration[29,30], have yet to be fully utilized. The majority of research and development in memristor materials and devices has been centered around modifying device features to align with current computing models or algorithms. However, this approach often oversimplifies or disregards the potential of emerging device features. Meanwhile, there has been minimal endeavor in algorithm design to incorporate these recent discoveries in materials and devices.

Here, we introduce drift-diffusion kinetics (DDK) model, which leverages drift-diffusion kinetics in resistive switching (RS) to enable feature learning. Firstly, a drift-diffusion-based physical model is presented to describe RS dynamics. The DDK model is proposed as a general form of this model. Secondly, the model is experimentally demonstrated on a single memristive device (TiN/TaO$_x$/HfO$_x$/TiN), with externally applied electrical pulses serving to tune the model parameters. Thirdly, construction of feature maps is demonstrated by applying electrical pulses to the device and recording its conductance

responses. Finally, the DDK neural network is experimentally implemented on 180 nm memristor chips using hardware-software co-optimization techniques to handle device intrinsic variation, especially in the kinetics. Compared with state-of-the-art feature extraction/learning algorithms, the DDK neural network exhibits exceptional performance on various tasks, with the number of parameters and computation operations (evaluated by multiply–accumulate operations (MACs)) greatly reduced. For example, in recognition task of 10 speakers in Speakers in the Wild (SITW[31]) dataset, the DDK network shows significantly better average accuracy (93.5%) than CNN (sample-level CNN[32], 88.1%). Moreover, the DDK network exhibits a reduction in the number of parameters and computational operations by approximately 296 and 6972 times, respectively, in comparison to the CNN. The reduction of parameters and operations across all layers may be attributed to the low number of adjustable parameters and the low complexity of the DDK layer, as well as the simplified classification layer resulting from the uncomplicated DDK layer. Even compared with memristor DNN hardware, the memristor DDK hardware can further reduce energy and area consumption by at least about 83 and 1128 times, respectively. The latency and energy consumption of the DDK operation are on the order of nano-second and pico-joule, respectively. Our approach significantly decreases the computational complexity of AI models by leveraging device physics, resulting in faster and more energy-efficient AI hardware.

## Results
### Memristor DDK model

We propose a RS model with simplified drift-diffusion kinetics (Fig. 1a) according to R. Stanley Williams et al[33]. The memristive device is a sandwiched structure with an insulation layer of length $L$ between two electrodes. The RS layer is divided into two regions, one has high concentration of dopants with resistance $R_{ON}$ and length $w$ while the other has almost zero dopants with resistance $R_{OFF}$. By assuming ohmic electronic conduction, linear ionic drift, linear dopant distribution in the dopant region and uniform electric field[33–35], we obtain

$$v(t) = \left( R_{ON} \frac{w(t)}{L} + R_{OFF} \left( 1 - \frac{w(t)}{L} \right) \right) i(t) \quad (1)$$

$$\frac{dw(t)}{dt} = \mu \frac{v(t)}{L} + \frac{D}{w(t)} \quad (2)$$

where $v(t)$ is externally applied voltage, $i(t)$ is current flow through the device, $\mu$ is mobility of dopants and $D$ is diffusion coefficient of dopants. Equation (1) describes current-voltage (I-V) relation in a voltage-controlled memristor. The first and second term on the right side of Eq. (2) describes the drift and diffusion velocity of dopants, respectively. A typical bipolar RS behavior of the model is shown in Fig. 1b, 1c. Other typical curves, e.g., abrupt switching and negative differential resistance (NDR) are shown in Fig. S1. Equation (2) can be generalized into the following forms by assuming input or independent variable $x$ is always non-negative:

$$\frac{dy}{dt} = \frac{\alpha}{y} + \beta x \ (\text{SET}) \quad (3)$$

$$\frac{dy}{dt} = \frac{\alpha}{y} - \beta x \ (\text{RESET}) \quad (4)$$

where $y$ is output or state variable, $\alpha$ and $\beta$ are positive constants controlling increase and decrease speed of $y$, respectively. Equations (3) and (4) describe the dynamics behavior of $y$ at input $x$. In memristors, $\alpha = D$, $\beta = \mu/L$, $y = w(t)$ and $x = v(t)$. $y$ is output or the state variable of the memristor device that represents the length of the doped region within the resistive switching layer after the diffusion and

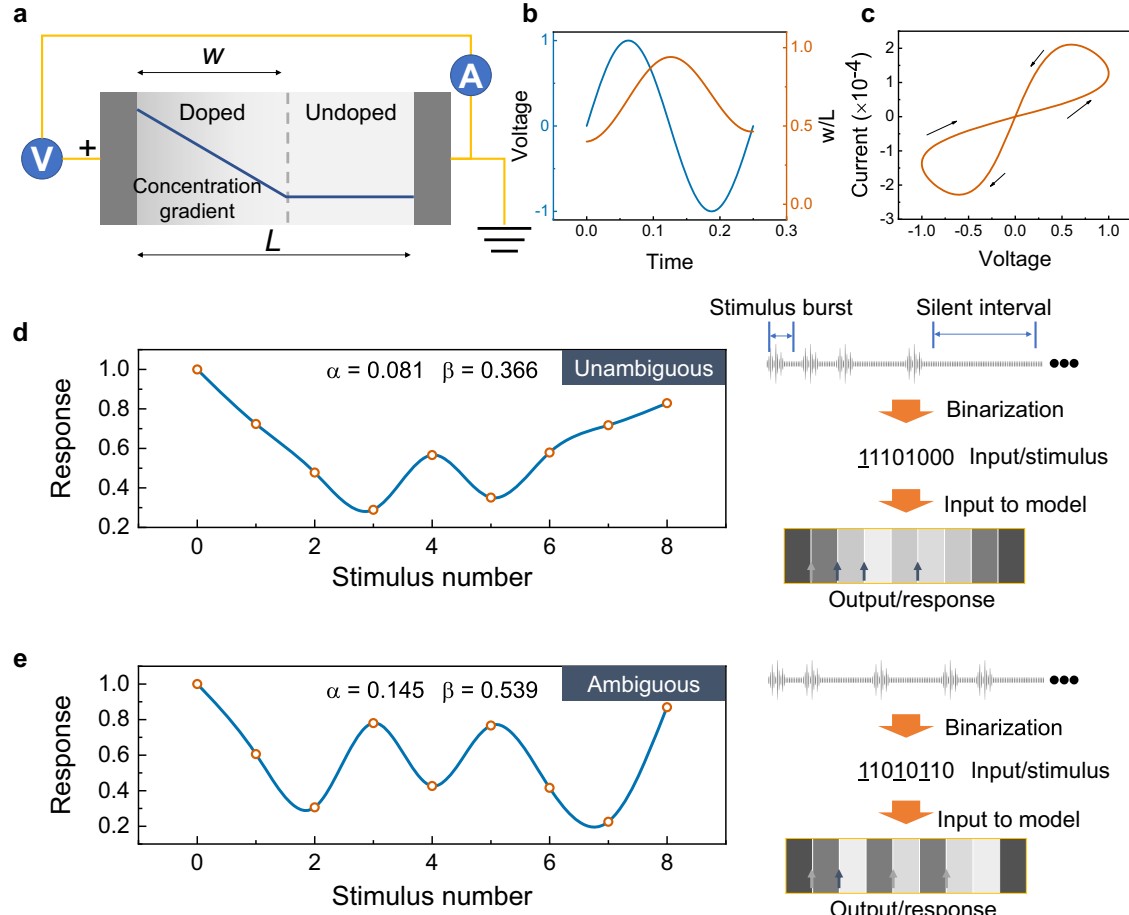

**Fig. 1 | RS model and its principle for temporal feature learning. a** A simplified drift-diffusion model for RS. Internal state of a memristor is described by length of a doped region $w$. The length of an un-doped region is $L - w$. Time evolution of the internal state is related to the drift and diffusion velocity of the doped region. **b** Variation of $w/L$ with a sinusoidal input voltage $v_O\sin(\omega_O t)$. The axes are dimensionless, with $v_O = 1$ V, $\omega_O = 10$, $\mu = 3.2 \times 10^{-14}$ m²V⁻¹s⁻¹, $R_{OFF}/R_{ON} = 20$ and $L = 60$ nm. **c** A typical simulated $I$-$V$ curve of the model shows electric hysteresis. Response of the model to unambiguous (**d**) and ambiguous (**e**) sounds, respectively. The sounds

are firstly binarized into vectors. The vectors are input to the algorithm and its output (response) are recorded. Then, the time evolution of response with input number is obtained. For the unambiguous sound, the strongest response (indicated by a grey arrow) occurs only at the start when the sound is repeated one-by-one. However, for the ambiguous sound, the response shows three possible starting points. The predicted results are highly in coincidence with psychological experiments.

migration of dopants. From Eq. (1), it is evident that since $R_{ON}$ is significantly smaller than $R_{OFF}$ ($R_{ON} << R_{OFF}$), the resistance of the device is primarily influenced by the second term ($R_{OFF}\left(1 - \frac{w(t)}{L}\right)$). Specifically, as the variable $w(t)$ increases, the resistance of the device decreases, clearly illustrating a negative correlation between $w$ and the device resistance. Therefore, the $y$ in Eqs. (3) and (4) is positively correlated with the conductivity of the memristor device. $\alpha$ and $\beta$ are positive constants controlling increase and decrease speed of $y$, respectively. Equations (3) and (4) describe the dynamics behavior of $y$ at input $x$.

Here we show that Eq. (4) can be used to extract feature from temporal pattern. Two temporal pattens, namely unambiguous (Fig. 1d) and ambiguous (Fig. 1e), are presented as examples. Here "1" stands for the present of a sound, while "0" present a silence. For unambiguous pattern, the responses at the start input are more pronounced (Fig. 1d). On the contrary, the responses of ambiguous pattern at the first, fourth and sixth inputs are almost equal (Fig. 1e). Thus, by employing the RS model at RESET (Eq. (4)), critical features in temporal patterns can be obtained for discrimination. As shown in Fig. S2, $\alpha$ is close related to an asymptote position of $y$, while $\beta$ mainly controls the decrease speed. This indicates that the diffusion coefficient of dopants ($D = \alpha$) mainly determines the final conductance, while the mobility of dopants

($\mu = \beta L$) influences primarily the rate of conductivity change. With the increasing of $\beta$, the output $y$ decreases more rapidly to asymptote so that it keeps almost constant even under subsequent input signals. This observation suggests that as the mobility of dopants increases, the rate of conductance declines more precipitously. Therefore, combination of diffusion coefficient of dopants and mobility of dopants can effectively control not only the feature learning process, but also the length of information where feature learning is performed. The different tunability of $\alpha$ and $\beta$ are discussed in the following. From the physical meanings in Eq. (2), $\alpha$ ($=D$) and $\beta$ ($=\mu/L$) should be insensitive to external electric field. However, the normalized $x(t)$, other than the real voltage configuration it is mapped to, is used as input in the curve-fitting, thus, the impact of pulse configuration or initial state is reflected in $\beta$ value. This method can overcome difficulties in representing pulse configuration and initial state that are physical multiparameter as a single mathematical value.

## Memristive devices and characterization

The choice of appropriate $\alpha$ and $\beta$ values directly influences the learned features. Hence, it is imperative to consider the adjustability of $\alpha$ or $\beta$ in order to achieve a comprehensive hardware realization of the DDK.

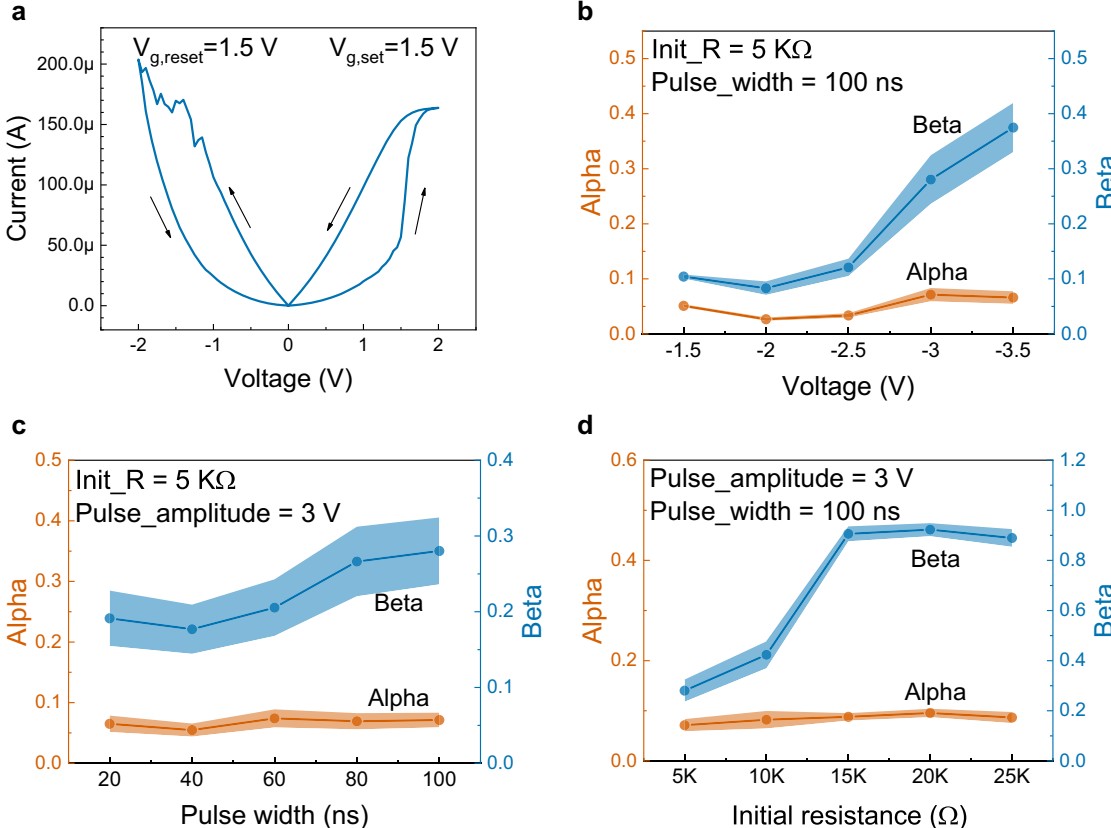

**Fig. 2 | Characterization of TiN/TaO$_x$/HfO$_x$/TiN memristive crossbar array cells. a** A typical *I-V* curve of the bipolar RS behavior. The SET process is more abrupt than the RESET one. Gate voltages for SET ($V_{g,set}$) and RESET ($V_{g,reset}$) are both 1.5 V. Tunability of parameters $\alpha$ and $\beta$ in the proposed algorithm with input pulse configurations applied on a single memristor. The influences from pulse amplitude (**b**), pulse width (**c**) and initial resistance (**d**) are studied in cycle-to-cycle test (100 cycles), respectively. Statistics of $\alpha$ (orange line) and $\beta$ (blue line) are presented in error band diagrams (95% confident interval). Parameter $\alpha$ is almost insensitive to input pulse and initial resistance, while parameter $\beta$ is positively proportional to the above factors. Moreover, when the initial resistance becomes larger, $\beta$ saturates.

The adjustability of $\alpha$ or $\beta$ is experimentally studied in a TiN/TaO$_x$/HfO$_x$/TiN memristor-based crossbar array. The fabrication and characterization of the array is detailed in Methods. The device structure is confirmed by high resolution transmission electron microscope (HRTEM, Fig. S3). A typical *I-V* curve of the device in RS are illustrated in Fig. 2a. The RS mechanism is studied by comprehensive physical modelling detailed in Methods. The physical parameters used for device simulation is listed in Table S1 and the results are shown in Fig. S4, Figures S5 and S6. The formation and rupture of the conductive filament driven by the drift-diffusion kinetics of oxygen vacancy probably induces the RS. The pulsed *I-V-t* curves are shown in Fig. S7, showing linear and symmetric conductance change with an operation region from 6.1 µS to 202.1 µS.

From Eq. (2), the switching dynamics is influenced by input voltage, charge carrier mobility or diffusion coefficient and thickness of the switching layer. The only possible influence factor that can be controlled in-situ is input voltage. Moreover, initial resistance state is another factor since it determines the ON/OFF ratio. Therefore, the tunability of $\alpha$ and $\beta$ is studied under various pulse configurations with different initial resistances. The input $x(t)$ in Eq. (4) is assumed to be a vector with a constant value 1 and a length of 100. Then $x(t)$ is mapped to a certain pulse configuration with 1 being represented by pulse amplitude or width. Afterwards, voltage pulses in this configuration are applied on the device and the conductance change with pulse number is measured. At last, the conductance change region is normalized. Figure 2b-d shows error band diagrams (95% confident interval) of fitted $\alpha$ and $\beta$ for the RESET process in cycle-to-cycle tests (100 cycles) with various pulse amplitudes, pulse widths and initial resistances, respectively. It can be concluded that $\alpha$ value is almost

independent on the above influence factors, while $\beta$ value is positively proportional to the above factors. $\beta$ becomes almost constant at initial resistance of ≥15 KΩ, since the first applied voltage pulse can cause a nearly complete RESET in this situation with a low ON/OFF ratio. Device-to-device tests (100 devices, Fig. S8) draw identical conclusions. The error bars in Fig. S8 denote a 95% confidence interval. In RESET, $\beta$ describes the decease speed of device conductance. With large values of amplitude, width or initial resistance, RESET process becomes more rapidly.

### Feature learning with DDK model and memristor device

Construction of the DDK feature maps is a necessary step for subroutine tasks, e.g., pattern classification. The process of feature map construction is demonstrated using 1-D raw waveform as an example. Two methods are proposed. One is that the feature is extracted from the whole waveform. Since Eq. (4) is calculated sequentially, this method is time-consuming and may not fully exploit the parallel nature of memristor array. The other is that the high-dimensional waveform is sub-sampled into several low-dimensional utterances, as shown in Fig. 3a. Then utterance features can be extracted in parallel. This method is faster than the first one but uses more devices. A 1-D pooling layer can be used in the above two methods for dimension reduction. For algorithm-level implementation, the constraints on the variables in Eq. (4) are removed. The input vectors are firstly normalized to a range of 0 to 1. The corresponding output is the extracted feature. For the hardware implementation with memristors, the input vectors are linearly mapped to the amplitude or width of voltage pulses, suggesting that the samples and voltage pulses have equal number. Then the voltage pulse train is applied on

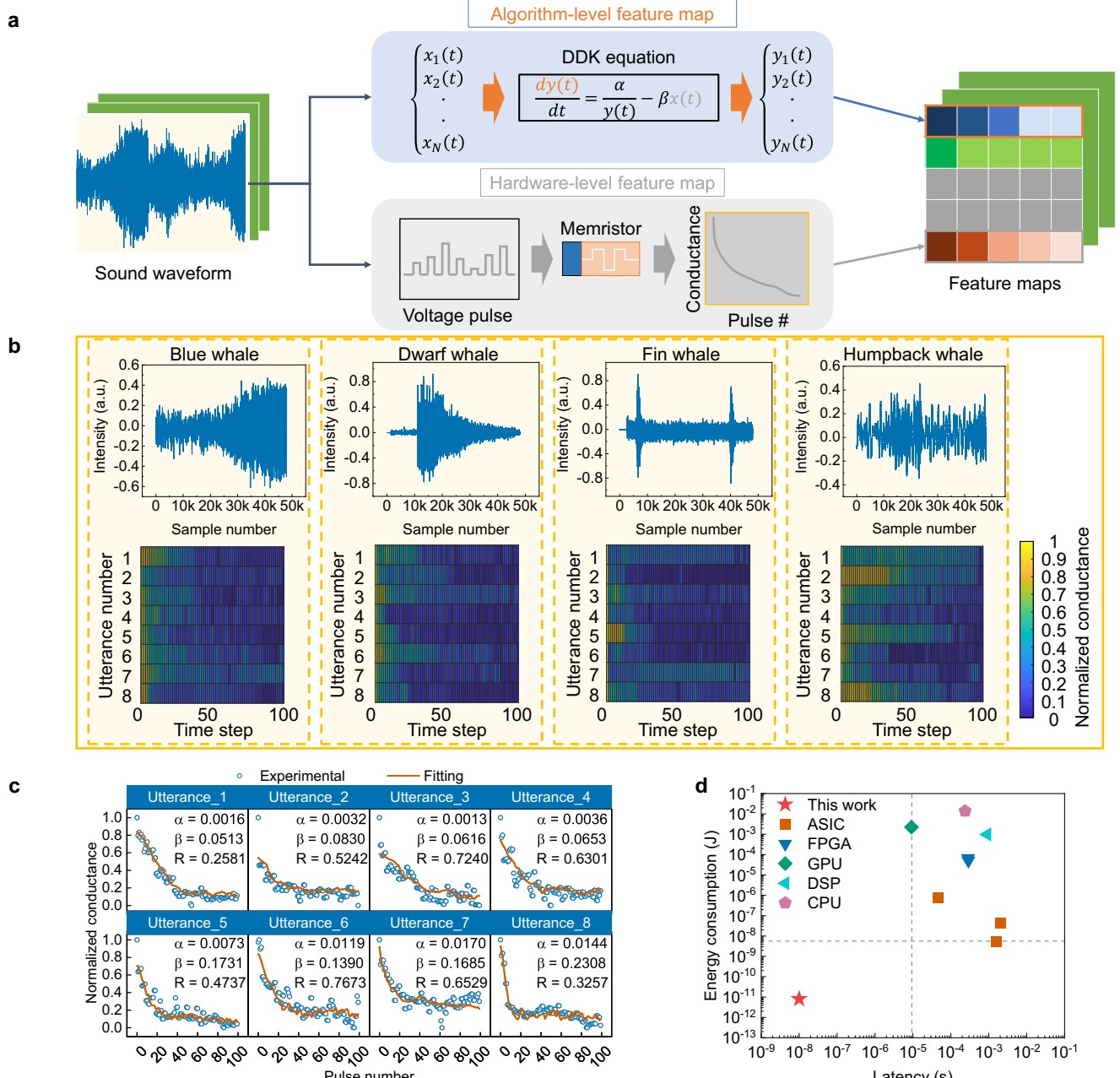

**Fig. 3 | Construction of memristor-based feature maps. a** A flow chart for feature map construction from waveforms. Firstly, the waveform is input into the drift-diffusion kinetics (DDK) equation (algorithm) or transformed to a series of voltage pulses (hardware). Then, the equation solution or memristor conductance response constitutes the feature map. **b** Hardware-level feature extraction from whale sounds in WMMS. The sound waveforms of blue, dwarf, fin and humpback whales (upper panel) are sub-sampled into 8 utterances and average pooled

(1 × 60). The corresponding feature maps are shown in the lower panel. **c** The device conductance change of the hardware feature map (blue whale). The fitted *I-V* curves of alpha and beta parameters are shown by orange solid lines, while the measured curves are depicted as blue scattered dots. The fitting accuracy is represented as sum of the squared residuals (R). **d** Comparison of our method in terms of latency and energy consumption with MFCC-based hardware.

the memristor, and the resultant conductance after each pulse constitutes the feature map. Feature map construction from high-dimensional raw data follows the same procedure by decomposing the tensors into several 1-D vectors.

Sound waveforms from 4 whales, namely blue, dwarf, fin and humpback whale, in Watkins Marine Mammal Sound (WMMS) database[36], are presented in upper panels of Fig. 3b to experimentally demonstrating the feature map construction. Each whale soundwave of 48000 samples is firstly divided into 8 utterances. Secondly, the 6000 samples in each utterance are reduced in dimension by an average 1-D pooling layer with a size of 1 × 60. Thirdly, the 100 samples

of each pooled utterance are mapped to voltage pulse amplitudes ranging from 2 to 3 V, pulse width and initial resistance are fixed at 100 ns and 5 KΩ, respectively. The conductance after each pulse applied on the device is measured. Finally, an 8 × 100 size feature map is constructed, as shown in lower panel of Fig. 3b. The conductance evolution of 8 utterances from blue whale sound is illustrated as an example in Fig. 3c. The fitted *I-V* curves of different pairs of alpha and beta parameters are shown by orange solid lines, while the measured *I-V* curves are depicted as blue scattered dots. The fitting accuracy is represented as sum of the squared residuals (represented as *R*) shown in individual plots.

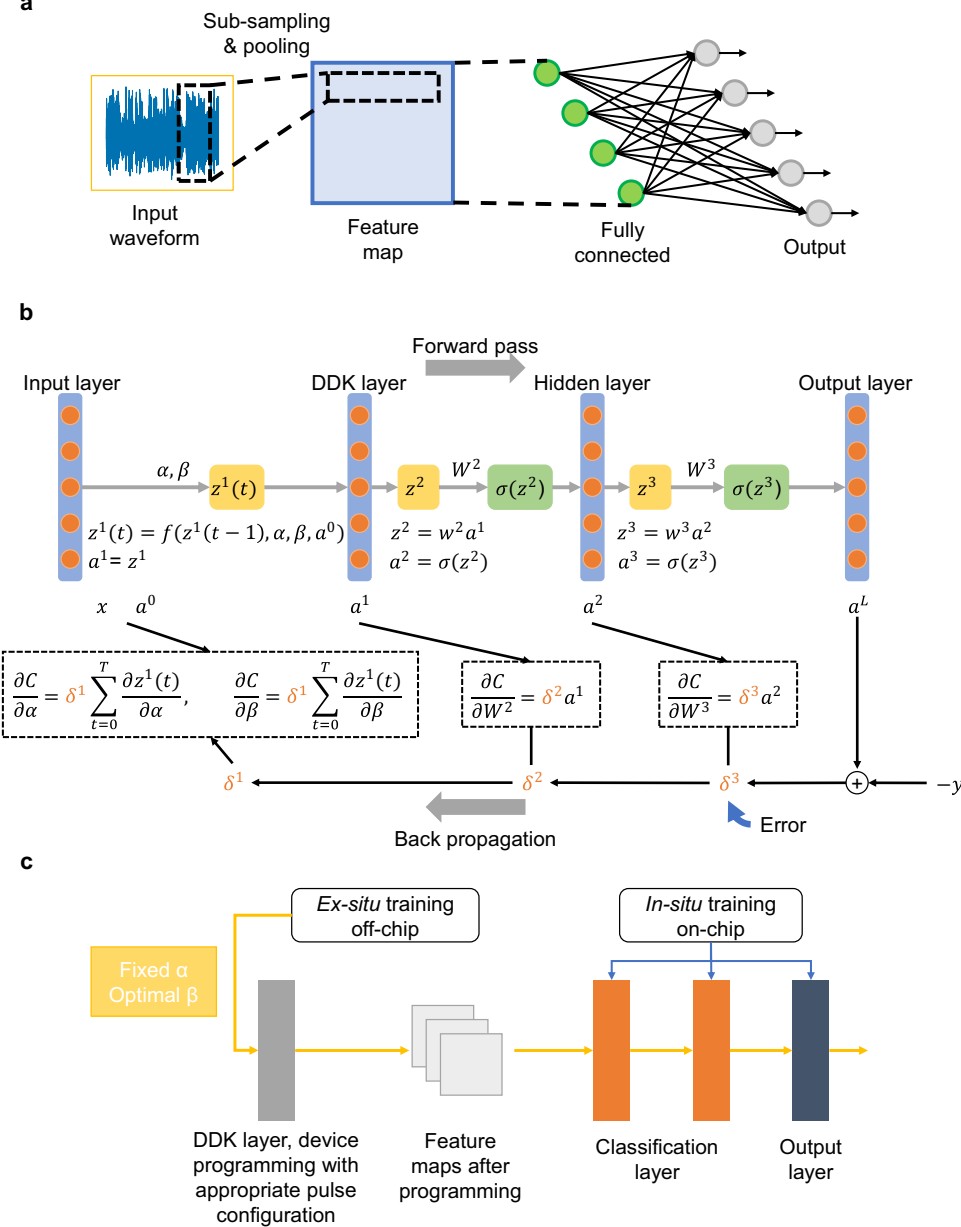

**Fig. 4 | Implementation of the DDK network. a** A typical architecture of DDK neural networks. Raw data is directly input to the DDK layer. After DDK layer, feature maps are obtained and multilayer perceptron is used for classification. **b** Back-propagation-based learning of parameters $\alpha$ and $\beta$. In contrast to conventional machine learning techniques where features are extracted manually, features in the proposed algorithm are learned automatically from datasets via back-propagation-based optimizer. **c** Hybrid training method. The DDK layer is trained ex-situ while the classification layer is trained in-situ.

Here the Mel-frequency cepstral coefficient (MFCC) method is presented for comparison with the DDK method. The MFCC-based method is a dominant feature extraction method that is widely used in audio signal processing[37]. It typically requires heavy preprocessing to guarantee a high-quality spectrum for subsequent coefficient calculation. In comparison with MFCC-based approaches (Fig. S9), hardware implementation of the proposed algorithm with memristors is very simple and shows great advantages in latency and energy consumption. As shown in Fig. 3d and Table S2, the latency and energy consumption of a single DDK operation of the TiN/TaO$_x$/HfO$_x$/TiN cell are estimated to be 10 ns and 0.8 pJ, using an average voltage of 2 V and an average resistance of 50 KΩ (detailed in Supplementary Notes), which are reduced by 917 (graphics processing unit, GPU[38]) ~1.6 × 10$^6$ (application specific integrated circuit, ASIC[39]) times and 6800 (ASIC[39]) ~1.725 × 10$^{10}$ (central processing

unit, CPU[40]) times in comparison with the MFCC operation, respectively.

## Neural network deployment and optimization

The DDK system architecture is shown in Fig. S10. To make the network training process be aware of hardware non-idealities, the imprecision-based neural network training framework incorporates limited device conductance range, output noise and device failure in network training (detailed in Supplementary Notes)[41]. Figure 4a shows a typical DDK network structure for pattern recognition. Figure 4b shows network training procedure, in which the parameters $\alpha$ and $\beta$ are learned via back-propagation method (detailed in Methods). After modeling, the network is deployed on the hardware (Fig. S11) via a compiler (detailed in Supplementary Notes). For task scheduling, a hybrid training technique that involves off-chip pre-training and on-

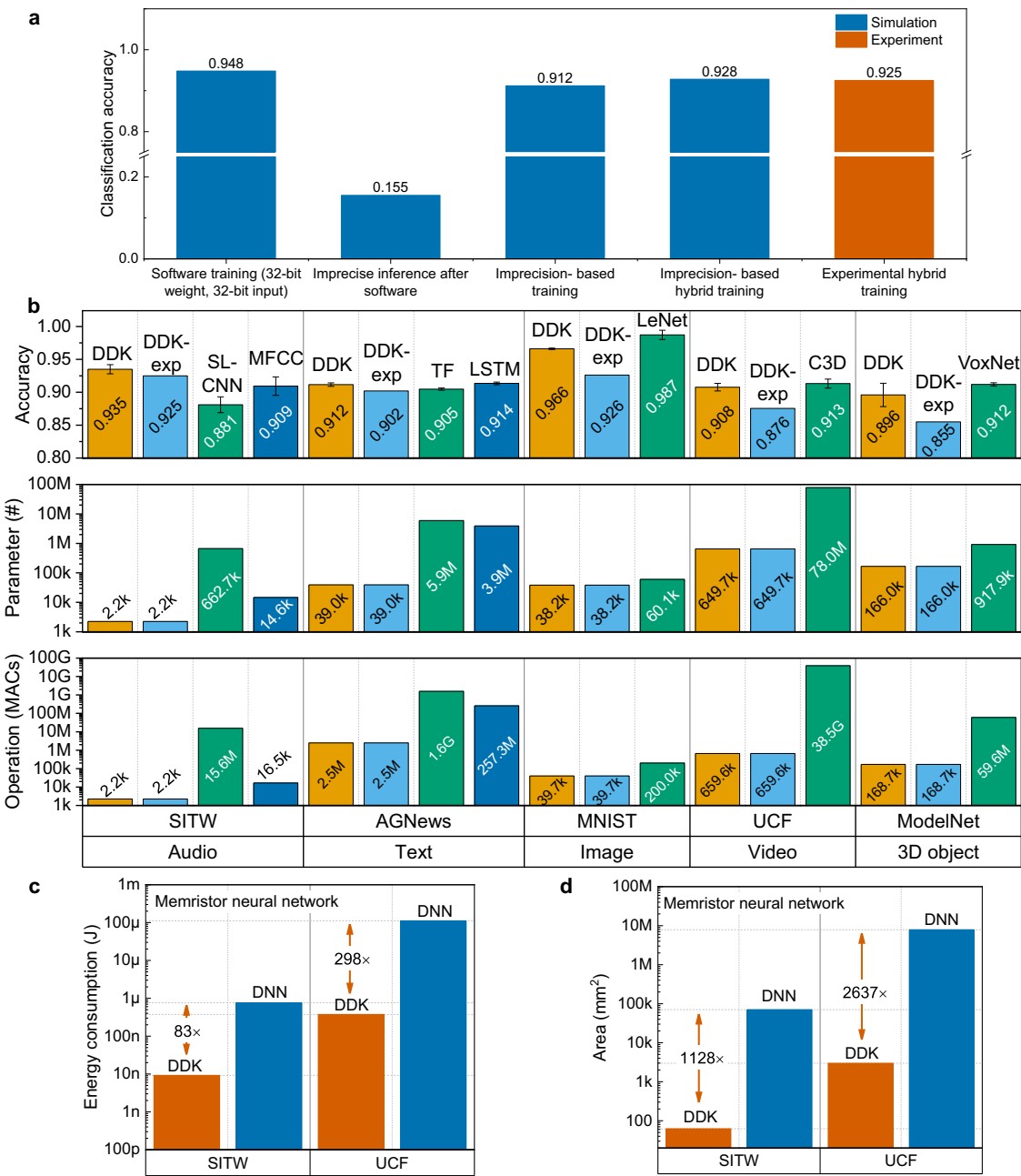

**Fig. 5 | Hardware-software co-optimization and comparison of DDK with deep learning and MFCC-based methods. a** Effectiveness of imprecision-based training and hybrid training in accuracy improvement. **b** Comparison of the DDK network (software: DDK; hardware experiment: DDK-exp) with other feature learning (DNN) and feature extraction (MFCC) techniques. Error bar is obtained by running simulations for 5 times. TF: Transformer. **c** Energy consumption comparison between the DDK network and memristor-based DNN. **d** Area comparison between the DDK network and memristor-based DNN.

chip fine-tuning is proposed (Fig. 4c) considering the large variation in DDK. In this technique, the DDK layer is trained ex-situ. The reason is that in hardware implementation, $\alpha$ and $\beta$ are implicit and obtained by curve fitting of the conductance evolution with time, but not by a direct conductance measurement as in the classification layer, thus evaluation and tuning of $\alpha$ and $\beta$ in-situ is time-consuming. The significant variations of parameters $\alpha$ and $\beta$ in in-situ training may introduce instability, impeding the network ability to converge rapidly.

The off-chip pre-training follows the procedure of back-propagation method in Fig. 4b. It includes the hardware constraints of $\alpha$ and $\beta$ in training, according to the experimental results in Fig. 2b-d. The on-chip fine-tuning phase is as follows: (1) initialization: with the parameters $\alpha$ and $\beta$ of the DDK layer already pre-trained, appropriate pulse configurations for performing feature extraction in the

memristor DDK layer are selected according to the experimental results. (2) Mapping pre-trained weights: the pre-trained weights of the classification layer are written into the RRAM devices by adjusting their conductance values using write-and-verify technique. The write-and-verify technique, weight updating and MVM output accuracy are shown in Fig. S12. This process ensures that the initial state of the memristor or resistance random access memory (RRAM) devices corresponds to the optimized weights obtained during the off-chip pre-training phase. (3) Forward pass: given a set of input data, the input features are firstly processed by the DDK layer. Feature maps are obtained by applying electrical pulses with such configuration to memristive devices in the DDK layer and record their conductance evolution with the pulses. These feature maps are then fed into the memristor classification layer. The computation results from the

RRAM arrays in the classification layer are used to generate predictions. (4) Loss calculation: the predictions generated by the memristor classification layer are compared against the actual targets to compute the loss function. (5) Backward pass and gradient calculation: the loss function is then differentiated with respect to the conductance values of the RRAM devices in the classification layer. This involves calculating the gradients of the loss function with respect to the RRAM output, which in turn requires calculating the gradients through the entire network (but focusing on the classification layer for updates). Since the DDK layer parameters are fixed, the gradients are only propagated through the classification layer. (6) In-situ weight update: using the calculated gradients, the conductance values of the RRAM devices are updated in-situ. The goal is to reduce the loss by adjusting the weights (represented by the RRAM conductance) in the direction that improves the model predictions. (7) Iteration and convergence: the forward pass, loss calculation, backward pass, and in-situ weight update steps are repeated iteratively until the model converges. (8) Evaluation: once the fine-tuning process is complete, the model performance is evaluated on a validation or test dataset to assess its generalization ability and robustness.

The advantage of our approach is that the feature extraction is performed by the differential equation that is defined by only two parameters, namely alpha and beta. While in the competing approaches, such as attention mechanism, convolution and recurrent connections, the feature extraction is defined by a set of weight matrices. Therefore, not only the feature extraction is simplified, but also the subsequent classification layer is minimized due to the small number of features to be processed. Moreover, our approach also accelerates the feature extraction and classification based on computing-in-memory. For other RRAM-based approaches that implements existing AI models, they only accelerate the MVM, but cannot simplify the feature extraction and classification layers.

## Pattern classification of the DDK neural networks

The effectiveness of the proposed imprecision-based training and hybrid training is studied in a 10-class speaker recognition task in the Speakers in the Wild (SITW) dataset[31], as shown in Fig. 5a. The details of the DDK network are shown in Supplementary Notes, Fig. S13a, and Table S3. When the entire network is exclusively trained based on software without non-idealities (referred to as software training), the accuracy of software testing can achieve 94.8%. Nevertheless, when simulating the hardware inference process by incorporating experimental noise into the aforementioned network (imprecise inference after software training), the accuracy is merely 15.5%. The reason is that software training cannot account for hardware imperfections, leading to a diminished level of network resilience. If the DDK and classification layer are all trained on software with noises (imprecision-based training), the test accuracy can reach 91.2%. The performance can be enhanced further by adopting the fine-tuning training method on the classification layer (imprecision-based hybrid training). This approach involves preserving or fixing the pre-trained parameters of the DDK layers, which were originally obtained through software training with noise, while solely retraining the classification layer on software with noise. Using this strategy, we ultimately achieve an enhanced robustness and attain an accuracy of 92.8%, reflecting a substantial improvement in performance. The DDK network was experimentally trained on memristive chips using the hybrid training method. The feature maps of the ten speakers in dataset are experimentally obtained with memristors, as shown in Fig. S14. The DDK layer is firstly trained on software and then deployed on hardware, while the classification layer is trained in-situ on hardware (experimental hybrid training). Training procedure and parameters are shown in Fig. S15 and Table S4, respectively. This experimental accuracy of 92.5% demonstrates the effectiveness of this strategy in enhancing classification accuracy on practical memristive hardware. The impact of cycle-to-

cycle variability on neural network accuracy is simulated in the speaker recognition task (Fig. S16). The variation is modeled as Gaussian noise[42–46]. The noise is added to the target conductance in the RRAM synaptic devices in the classification layer and the $\alpha$ and $\beta$ of the RRAM conductance in the DDK layer. The noise value equals to the standard deviation of the distribution. In the simulation process, the study range for $\alpha$ and $\beta$ noise is set between 0 and 0.5, while the range for synaptic noise is set between 0 and 2.5 (unit: $\mu S$). The results further confirm the effectiveness of the proposed hybrid training strategy.

The performances of the DDK algorithm and hardware on 1-D, 2-D and 3-D raw data are studied and compared with mainstream deep models, as shown in Fig. 5b. The details of the neural networks are presented in Supplementary Notes, Fig. S13, Table S3 and S5. The statistical findings were collected from a series of five network training sessions. Various pattern classification tasks, such as speaker recognition (SITW dataset), text classification (AG News dataset[47]), image recognition of handwritten digits (MNIST dataset[48]), action recognition from video stream (UCF dataset[49]) and 3D object recognition based on voxel (ModelNet dataset[50]) are performed on representative datasets. Among the five types of pattern classification tasks, the DDK network has a similar level of performance to the other networks being compared. The number of network parameters and operations are reduced by up to 296 times (speaker recognition) and 58,441 times (action recognition), respectively. To better disclose the importance of feature extraction in the neural networks and demonstrate the significance of the proposed DDK layer in simplifying the networks, the number of MACs and their proportions in the feature extraction, classification, and the total number of MACs in the DDK network and other comparative DNNs are listed in Tables S6 and S7. The statistical results show that in the LeNet-5 for MNIST dataset classification, the MACs in the feature extraction stage account for 70.5% of the total MACs. For other models such as SL-CNN for SITW dataset, VoxNet for ModelNet dataset, LSTM and Transformer for AGNews dataset, and C3D for UCF dataset, the proportion of MACs in the feature extraction stage is over 98% of the total MACs in the network. This finding highlights the significant impact that the feature extraction part has on the overall computational complexity of neural network models. It shows that the feature extraction part is the most computation-intensive one in the entire AI models. On the contrary, the proportion of MACs in the feature extraction stage in the proposed DDK networks only ranges from 1.2% to 10.39%. And the total MACs of the DDK network is also significantly lower than its counterparts.

Therefore, compared with mainstream DNNs, the proposed DDK layer not only simplifies the feature extraction, but also makes the entire AI model more lightweight. This improvement may not only reduce energy consumption, but also decrease latency, thereby enhancing the overall performance of the model when deployed on hardware. This is especially important for resource-constrained application scenarios, such as edge computing.

CNN use kernels to address phrase dependencies. The quantity of kernels needed to encompass the interconnections among every word combination in a phrase would be vast and impractical due to the exponential growth of combinations while augmenting the maximum length of input sentences. Furthermore, the dependency length is determined by kernel size, which is a manually selected hyperparameter. In recurrent deep learning methods such as recurrent neural network (RNN)[51] or long short-term memory (LSTM)[52], long-term dependencies are ignored due to exploding or vanishing gradient problems. Moreover, the dependency length is dictated by network structure that is hand-crafted hyperparameter, resulting in a larger number of parameters that need to be trained. Transformers eliminate recursion by processing entire sentences in parallel. Additionally, they mitigate performance degradation due to long dependencies by utilizing multi-head attention mechanisms and positional embeddings to learn relationships between words. This parallel processing capability

requires more parameters, especially when applying transformers to extended sequences. Transformers can only capture dependencies within predetermined input size.

The proposed DDK network explicitly models dependencies like transformers, eliminating the possibility of performance degradation in proportion to sequence length. Table S8 shows the comparison of network layers based on computational efficiency metrics. The DDK layer shows the reported low per-layer complexity and maximum path length, indicating that it can be computed more efficiently without losing long-term information. Although the DDK layer shares some similarity with RNNs in the processing of sequential operations, the problem of parallelization is less pronounced since the DDK network does not use deep structures to represent dependencies. Thus, the parallelization of the DDK layer can be performed on one of the two dimensions defining the input. Besides, a great advantage of the DDK model is that the dependency length is automatically determined from data, rather than relying on pre-determined hyperparameters as in other deep learning techniques. However, the DDK network shows a comparatively lower accuracy (2.11% decrease) in image classification than the CNN. The likely reason is that the handwritten digit images do not possess temporal relevance information. It is suggested that the DDK may have better performance in datasets where temporal relation among samples is existed and important.

In contrast to the outcomes obtained from software simulations, the experimental implementations generally exhibit a minor but acceptable decrease in accuracy ($\leq 4.1\%$), probably owing to the nonidealities in device and array. Quantization performance of the DDK network is evaluated on SITW dataset by using post-training quantization (PTQ) and quantization-aware training (QAT), as shown in Fig. S17, proving that quantization is feasible. In terms of quantization precision, there are slight accuracy loss ($\leq 4.8\%$, with activation function quantization) for the PTQ if the precision is $\geq 6$ bit. While for the QAT, even the accuracy loss of 4-bit quantization (3.6%, with and without activation function quantization) is still acceptable. Therefore, a 4-bit or higher weight precision is acceptable for the DDK network in this example, showing that the DDK hardware does not necessarily need a high-precision analog-to-digital convertor (ADC) or digital-to-analog convertor (DAC). The effect of endurance on neural network performance is studied in Supplementary Notes, Figs. S18 and S19. Probably because of the "second order memristor"[53], the write pulse with long width in the DDK layer leads to non-gradual resistive switching, making the device reaches a stable saturation state from the initial. As a result, the decrease in classification accuracy is suppressed. However, the long pulse width may result in a long write latency. This demonstrates that there is a trade-off between latency and accuracy, considering the endurance of memristor devices.

In the hardware benchmark, we evaluate the hardware performance using 8-bit ADC and DAC, which may be a reasonable selection for most situations. The energy-efficiency of the DDK hardware is evaluated in the speaker recognition task considering four layout strategies (Fig. S20). Based on the hardware parameters in Table S9, the latency, area and energy consumption of the circuits can be calculated (Table S10). The energy-efficiency is evaluated to reach 2.77 TOP s$^{-1}$ W$^{-1}$ at integer 8-bit (INT8) at 180 nm technology node (hardware benchmark is detailed in Supplementary Notes and the result is listed in Table S11, the number of operations of the networks used for evaluation are shown in Tables S12-S15), outperforming a Tesla V100 GPU by approximately 27.6 times[54]. The time and energy of a single feature extraction of the DDK hardware is estimated to be 35.4 ns and 236.68 pJ, respectively. Compared to the DNN implemented with memristors, the DDK networks demonstrate notable benefits in terms of energy (Fig. 5c) and area (Fig. 5d). One can expect a decrease in energy and area consumption by approximately 2 and 3 orders of magnitude, respectively.

## Discussion

In this work, we presented the drift-diffusion kinetics (DDK) network, the previously unexplored feature learning model based entirely on physical kinetics of memristors. It has reported low number of trainable parameters, layer complexity, and maximum path length of the DDK layer in comparison with existing deep models. Furthermore, it can be implemented fully using memristor crossbar arrays. The DDK feature learning can be realized directly using the resistive switching kinetics of the array cell by simply applying programming electrical signals and reading the corresponding device states after each signal. The classification layer is implemented utilizing the MVM computing capability of the arrays. The tunability limitations of the parameters present in a single memristor can be overcome by decomposing the DDK operation into several MVM ones. Our approach represents a significant step towards AI model design based on the properties of emerging materials and devices. We believe that it has the potential to spur the development of powerful AI techniques grounded in real-world physics, ultimately leading to the creation of fast and energy-efficient computing models and systems.

## Methods

### Device fabrication and characterization

The crossbar arrays are in one-transistor-one-resistor (1T1R) configuration. A 1T1R cell consists of a TiN/TaO$_x$/HfO$_x$/TiN memristive device stacked on via 5 of metal layer 5 and a NMOS transistor that acts as a selector to suppress sneak current and controls compliance current. The arrays are fabricated in a hybrid fashion where metal layers 1 to 4 are fabricated in a commercial standard 180 nm process and metal layers 5 to 6 are fabricated in a laboratory 180 nm process. The thickness of the TiN electrodes, TaO$_x$ layer and HfO$_x$ layer are ~30 nm, 45 nm and 9 nm, respectively (Fig. S3). The TaO$_x$ layer is heavily-doped with high oxygen vacancy concentration and act as an oxygen vacancy reservoir. After fabrication, the arrays are sliced and packaged.

To conduct electrical characterization, the chip is subjected to testing using a self-made hardware test system. The TiN electrode at the HfO$_x$/TiN interface is always grounded in electrical test. The material characterizations of the device, including structure and chemical component are performed by HRTEM. The flake samples are prepared by focused ion beam (FIB) technique. The chemical component is characterized by energy dispersive X-ray spectroscopy (EDS).

### Device modeling

Figure S4a shows the axisymmetric 2D model geometry of the TiN/TaO$_x$/HfO$_x$/TiN device. The RRAM device stack consists of a heavily-doped TaO$_x$ layer (45 nm) on top of a HfO$_x$ (9 nm) layer sandwiched by top and bottom TiN electrodes (TE and BE, 30 nm). In order to focus on the switching behavior of the memristor, the simulation starts immediately after the electroforming process, where a continuous conductive filament (CF) has been formed in the center of HfO$_x$ with a radius of 9 nm and a height of 9 nm. The CF corresponds to a region of high oxygen vacancy ($V_o^{..}$) concentration and thus exhibits locally high conductivity.

The local current and $V_o^{..}$ concentration distribution during the set and reset in this system are calculated by solving three coupled differential equations through a numerical solver (COMSOL Multiphysics). The differential equations comprise heat transfer equation (Eq. 5), current continuity equation (Eq. 6) and drift-diffusion equation (Eq. 7) for $V_o^{..}$ transport.

$$\rho C_p \frac{\partial T}{\partial t} + \nabla \cdot (-k \nabla T) = \frac{J^2}{\sigma} \tag{5}$$

$$\nabla \cdot J = \nabla \cdot (\sigma \nabla V) = 0 \tag{6}$$

$$\frac{\partial n_{V_o^{\cdot\cdot}}}{\partial t} = \nabla \cdot (D\nabla n_{V_o^{\cdot\cdot}} + z_i \frac{D_i}{RT} F n_{V_o^{\cdot\cdot}} \nabla V + DSn_{V_o^{\cdot\cdot}} \nabla T) \tag{7}$$

where

$J$: current density,
$\sigma$: electrical conductivity,
$V$: electrical potential,
$T$: temperature,
$k$: thermal conductivity,
$C_P$: heat capacity,
$\rho$: density of mass,
$t$: time,
$n_{V_o^{\cdot\cdot}}$: concentration of $V_o^{\cdot\cdot}$ [mol/cm$^3$],
$D$: diffusion coefficient of $V_o^{\cdot\cdot}$ [cm$^2$s$^{-1}$],
$S$: Soret diffusion coefficient [1/K],
$z_i$: (dimensionless) is the charge number of the ionic species,
$R$: the molar gas constant,
$F$: Faraday's constant.

When CF region is in low resistance state, a large amount of Joule heat will be generated, so the influence of temperature on the electrical conductivity ($\sigma_{th}$) and thermal conductivity ($k_{th}$) of TaO$_x$ and HfO$_x$ region should be considered. With this, $\sigma$ and $k$ is modeled according to Eqs. (8) and (9):

$$\sigma_{th} = \sigma_{th0} \exp(-E_{ac}/k_B T) \tag{8}$$

$$k_{th} = k_{th0}(1 + \lambda(T - T_0)) \tag{9}$$

$k_B$: the Boltzmann constant,
$T_0$: initial temperature,
$\lambda$: thermal coefficient.
$N_A$: the Avogadro's constant.
In the Eq. (7), D and S are the function of $T$, shown in Eqs. (10–11):

$$D = 0.5a^2 f \exp(-E_a/(k_B T)) \tag{10}$$

$$S = -E_a/(k_B T^2) \tag{11}$$

The boundary conditions applied in this model are shown in Fig. S4a. The boundary conditions for the current continuity equation are $V = 0$ V and $V = V_D$ at the BE and TE, respectively. Besides, the rest of the outer boundary is electrically insulated. The initial potential at $t_O = 0$ s is $V(t_O) = 0$ V in all regimes. The thermal boundary conditions at the outermost boundaries of the device are defined by Dirichlet boundary conditions in terms of a constant value for $T_0 = 293.15$ K. The initial value of the temperature is $T_0$ in all regimes. For the $V_o^{\cdot\cdot}$ drift-diffusion, no flux was assumed at the TE/TaO$_x$ and HfO$_x$/BE interfaces. A uniform concentration of $1 \times 10^{21}$ cm$^{-3}$ was assumed within the CF and TaO$_x$ layer as the initial state (i.e., the forming state). The material parameters used in simulation are summarized in Table S1.

Figure S4b shows the measured and calculated I-V characteristics during the set and reset processes. The stimulated I-V curves are consistent with measured results. The reset transition starts at negative voltage of about −1.25 V. The resistance gradually increases and finally reaches a high resistance after reset. Similarly, the set transition occurs at a positive voltage of near 1.4 V and recovers the original low resistance state.

Figures S5 and S6 illustrate the distribution of $n_{V_o^{\cdot\cdot}}$ and the temperature in the TaO$_x$ and HfO$_x$ regions during the set and reset processes, respectively. As observed from Fig. S5, the highest temperature is recorded in the conductive filament (CF) area, with peak values of approximately 420 °C (Fig. S5b) and 430 °C (Fig. S5f) at the edge of the reset and set points, respectively. As evident from Fig. S6,

during the reset process, a marked decrease of $n_{V_o^{\cdot\cdot}}$ is observed near the bottom electrode within the CF region at the edge of reset point (Fig. S6b, corresponding to Point B in Fig. S4b), indicating the onset of filament rupture, leading the device into a low-conductance state. Conversely, during the set process, a sudden increase of $n_{V_o^{\cdot\cdot}}$ near the bottom electrode within the CF region at the edge of set points (Fig. S6f, corresponding to Point F in Fig. S4b) indicates the growth of the CF, leading to the device transitioning to a high-conductance state.

The connection between COMSOL simulations and Eqs. (3) and (4) lies in their description of different levels of physical phenomena, but both involve the behavior of internal ions or defects (such as oxygen vacancies, $V_o^{\cdot\cdot}$) during the resistive switching process of the device. Equations (3) and (4) describe a simplified model that captures the fundamental dynamics of conductive filament (CF) growth and rupture during resistive switching based on the highly idealized assumptions of ohmic electron conduction, linear ion drift, linear dopant distribution in the dopant region and uniform electric field. These equations focus on the change in CF length $w(t)$ over time, where $y$ can be considered an alternative variable for $w(t)$, and $x$ represents the externally applied voltage or current. Equation (3) corresponds to the SET process, while Eq. (4) corresponds to the RESET process. The term $\beta x$ describes the drift motion of ions or defects, and the term $\frac{\alpha}{y}$ reflects the diffusion effect. On the other hand, COMSOL simulations employ a more detailed physical model. Equation (7) describes the changes in the concentration of oxygen vacancies ($n_{V_o^{\cdot\cdot}}$), considering diffusion, drift, and the influence of temperature gradients. Specifically, the term $D\nabla n_{V_o^{\cdot\cdot}}$ and $z_i \frac{D_i}{RT} F n_{V_o^{\cdot\cdot}} \nabla V$ stand for Fick diffusion flux and drift flux, respectively. The term $DSn_{V_o^{\cdot\cdot}} \nabla T$ is the Soret diffusion flux. Soret diffusion is the movement of molecules along a temperature gradient. Equations (3) and (4), as well as the COMSOL simulation model, both describe the drift and diffusion movements of ions or vacancies within the memristor. However, Eqs. (3) and (4) represent a simplified model based on highly idealized assumptions, whereas the COMSOL simulation model considers a wider range of practical factors, thus providing a more comprehensive and realistic representation of the device behavior.

## Curve fitting
The experimental I-V curves are fitted by Eq. (4) (assuming the alpha and beta are all nonnegative). The fitting is a nonlinear least squares problem. The minimization of the difference between experimental data and simulations is performed by the trust-region-reflective algorithm[55]. Before fitting, the experimental data is normalized to a range from 0 to 1. The upper and lower boundaries for alpha and beta are set to 0 and 1, respectively. The initial guess values of alpha and beta are random values in the range from 0 to 1.

## Backpropagation-based feature learning
DDK networks rely on the backpropagation algorithm to execute the process of parameter updates and optimization during training. The training process is presented in the following. Firstly, the preprocessed input data is propagated forward through each layer of the network to obtain the predicted output. Then, the loss between the prediction and the actual target is calculated. Next, the gradients of the loss function with respect to $\alpha$ and $\beta$ in the DDK layer, as well as each weight in the classification layer, are computed using the chain rule. Finally, the computed gradients are used to update $\alpha$ and $\beta$ in the DDK layer and the weights in the classification layer, aiming to minimize the loss function. The specific methods for preprocessing the data, forward propagation, backward propagation, and parameter updating are as follows.

1. Input data pre-processing
Each utterance or vector from the raw dataset is normalized to [0, 1].

2. Forward pass
1) DDK layer

$$\mathbf{z}^1(0) = [1, 1, 1 \cdots 1] \tag{12}$$

for $t = 1$ to $T$, do

$$\mathbf{z}^1(t) = \mathbf{z}^1(t-1) + \frac{\alpha}{\mathbf{z}^1(\mathbf{t-1})} - \beta \mathbf{x}(t-1) \tag{13}$$

$$\mathbf{a}^1 = \mathbf{z}^1 \tag{14}$$

end for.

$T$ is the total length of utterance or vector. $\mathbf{z}^k(t)$ is the output of the $k$th layer (the first layer is the DDK layer) at input index $t$. $\mathbf{a}^1$ is the neuron output of the DDK layer.

2) Hidden layers:
for $k = 2$ to $L$-1 do

$$\mathbf{z}^k = \mathbf{W}^k \mathbf{a}^{k-1} \tag{15}$$

$$\mathbf{a}^k = \sigma(\mathbf{z}^k) \tag{16}$$

end for.

3) Output layer:

$$\mathbf{z}^L = \mathbf{W}^L \mathbf{a}^{L-1} \tag{17}$$

$$\mathbf{a}^L = SoftMax(\mathbf{z}^L) \tag{18}$$

$L$ is the total number of layers in the network. $\sigma$ is an activation function. $\mathbf{W}^k$ is the weight matrix of the $k$th layer.

3. Backward pass
1) Output layer
a. Calculate the gradient of $\mathbf{W}$ in output layer:

$$\frac{\partial C(\alpha, \beta, \mathbf{W})}{\partial \mathbf{W}} = (\mathbf{a}^L - \mathbf{y}) \mathbf{a}^{L-1} \tag{19}$$

b. Update $\mathbf{W}$:

$$\Delta \mathbf{W} = -\eta \cdot \frac{\partial C(\alpha, \beta, \mathbf{W})}{\partial \mathbf{W}} \tag{20}$$

$\mathbf{y}$ is truth label, $\eta$ is learning rate, $C(\alpha, \beta, \mathbf{W})$ is loss function.

2) Hidden layer
a. Calculate the error $\delta^k$ and gradient of $\mathbf{W}$ in the $k$th layer:

$$\boldsymbol{\delta}^k = \mathbf{W}^{k+1} \boldsymbol{\delta}^{k+1} \odot \sigma'(\mathbf{z}^k) \tag{21}$$

$$\frac{\partial C(\alpha, \beta, \mathbf{W})}{\partial \mathbf{W}} = \boldsymbol{\delta}^k \mathbf{a}^{k-1} \tag{22}$$

b. Update $\mathbf{W}$:

$$\Delta \mathbf{W} = -\eta \cdot \frac{\partial C(\alpha, \beta, \mathbf{W})}{\partial \mathbf{W}} \tag{23}$$

3) DDK layer

a. Calculate the gradient of $\alpha$ and $\beta$:

$$\begin{cases} \frac{\partial \mathbf{z}^1(t)}{\partial \alpha} = \frac{1}{\mathbf{z}^1(t-1)}, & t = 1 \\ \frac{\partial \mathbf{z}^1(t)}{\partial \alpha} = \frac{1}{\mathbf{z}^1(t-1)} + \prod_{t=2}^{T} \frac{\partial \mathbf{z}^1(t)}{\partial \mathbf{z}^1(t-1)} \cdot \frac{\partial \mathbf{z}^1(1)}{\partial \alpha}, & 2 \leq t \leq T \end{cases} \tag{24}$$

$$\begin{cases} \frac{\partial \mathbf{z}^1(t)}{\partial \beta} = -\mathbf{x}(t-1), & t = 1 \\ \frac{\partial \mathbf{z}^1(t)}{\partial \beta} = -\mathbf{x}(t-1) + \prod_{t=2}^{T} \frac{\partial \mathbf{z}^1(t)}{\partial \mathbf{z}^1(t-1)} \cdot \frac{\partial \mathbf{z}^1(1)}{\partial \beta}, & 2 \leq t \leq T \end{cases} \tag{25}$$

$$\frac{\partial C(\alpha, \beta, \mathbf{W})}{\partial \alpha} = \boldsymbol{\delta}^1 \sum_{t=1}^{T} \frac{\partial \mathbf{z}^1(t)}{\partial \alpha} \tag{26}$$

$$\frac{\partial C(\alpha, \beta, \mathbf{W})}{\partial \beta} = \boldsymbol{\delta}^1 \sum_{t=1}^{T} \frac{\partial \mathbf{z}^1(t)}{\partial \beta} \tag{27}$$

b. Update $\alpha$ and $\beta$:

$$\Delta \alpha = -\eta \cdot \frac{\partial C(\alpha, \beta, \mathbf{W})}{\partial \alpha} \tag{28}$$

$$\Delta \beta = -\eta \cdot \frac{\partial C(\alpha, \beta, \mathbf{W})}{\partial \beta} \tag{29}$$

This process ensures that the DDK network, like other neural networks, can effectively learn and adjust its parameters through the backpropagation algorithm. In our approach, the data transformation is performed by the differential equation that is defined by two parameters $\alpha$ and $\beta$. While in the competing approaches, such as attention mechanism, convolution and recurrent connections, this transformation is defined by a set of weight matrices. Consequently, our approach can achieve minimal computational complexity, as evidenced by a limited number of trainable parameters and computational operations.

In hardware implementation, the backpropagation is performed on CPU using experimentally measured device conductance and current. The gradient of the loss function with respect to the conductance of the RRAM devices is calculated as follows:

$$\boldsymbol{\delta}^k = \mathbf{G}^{k+1} \boldsymbol{\delta}^{k+1} \odot \sigma'(\mathbf{z}^k) \tag{30}$$

$\boldsymbol{\delta}^{k+1}$ is the error in the $(k+1)$th layer, which is transmitted back to the $(k+1)$th layer by the gradient of the loss function to softmax. $\mathbf{G}^{k+1}$ is value of the conductance of the synaptic weight mapping in the $(k+1)$th layer. $\sigma'(\mathbf{z}^k)$ is the derivative of the activation function with respect to output current in the $k$th layer. The calculation of gradient of the loss function with respect to the conductance of the RRAM devices as follow:

$$\frac{\partial C(\alpha, \beta, \mathbf{W})}{\partial \mathbf{G}} = \boldsymbol{\delta}^k \mathbf{a}^{k-1} \tag{31}$$

$\mathbf{a}^{k-1}$ is the output in the $(k-1)$th layer. Specifically, it is the value of the output current of the $(k-1)$th layer after it has been processed by the activation function. The activation function is implemented on CPU.

## Data availability
All other data are available from the corresponding authors on request. Source data are provided with this paper.

## Code availability

The codes are available at github[56].

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

## Acknowledgements

This work was supported by the National Key R&D Program of China under Grant No. 2021YFB3601200, the National Natural Science Foundation of China under Grant Nos. U20A20220, U22A6001, 61821091, 61888102, and 61825404, the Key R&D Program of Zhejiang (No. 2022C01048), the Strategic Priority Research Program of the Chinese Academy of Sciences under Grant XDB44000000.

## Author contributions

T.S. conceived the idea. T.S., and L.G. implemented neural network simulation. T.S., Y.T., and YuL. performed device modeling and finite element simulation. T.S., S.T., R.Z., S.C., and Z.W. performed device test and characterization. J.L. set up the hardware and conducted the measurements. Y.L., J.L., and T.S. developed the software toolchain. Y.L., and H.Z. benchmarked the system performance. T.S., L.G., J.L., S.T., S.C., X.Z., T.L., X.Y., and Q.L. performed the data analysis. T.S., Y.T., J.L., L.G., Y.L., X.Y., and Q.L. wrote the manuscript. X.Y., and Q.L. supervised the project. All authors discussed the results and implications and commented on the manuscript at all stages.

## Competing interests
The authors declare no competing interests.
