## [Peer Review File · Nature Communications]

REVIEWER COMMENTS

Reviewer #1 (Remarks to the Author):

The submitted work presents an innovative way of using ReRAM kinetics for modelling the generation of features provided to an AI model. The work claims many orders of magnitude better energy and latency than CPU and GPU. Main points of the proposed work are 1- a drift-diffusion model for the ReRAM 2- the experimental characterization of ReRAM devices, and 3- the proposed model embedding is experimentally validated on a custom test system.

While the DDK model itself is not novel, the use of dynamic properties of memristors has not been explored yet in literature, and the idea is itself very interesting. On the other hand, the reviewer finds that the current draft does not address some important points, such as:

- Most of the experimental characterization presented in the supplementary material compares the feature extraction with DDK with the MFCC method. While there is a clear advantage in the feature extraction phase, it is unclear to the reviewer how much the feature extraction itself impacts the energy and latency of training and feed-forwarding for the entire deployed model. If the feature extraction is just a small part, then the results presented are not relevant. The data points provided should provide more details about which model is being accelerated, how do competitive approaches perform, and how much that task is contributing towards the overall energy/latency metrics of the deployed model.

- Data points are provided for competitive CPU, GPU, and ASIC approaches, but the reviewer finds disappointing that insufficient information is given about how these data points are extracted. While there is no established framework for simulating easily of all these components, it is the author's responsibility to show that the comparisons were fair; IMC will only be really accepted by all the hardware community if demonstrations of its usefulness come with great clarity and transparency.

- In the reviewer's opinion, the subsection about backpropagation in the Methods section is not sufficient to address this subject as is. It would need a dedicated subsection in the Results and Discussion section, with the purpose of providing an intuition of how the feature are feed-forwarded and backpropagated along the new hardware implementation of the model based on using the differential equation that governs the ReRAM and why this is, from the point of view of the entire deployed model, more computationally efficient than other ReRAM-based approaches to IMC. This discussion would allow to also introduce how much the DDK method impacts the following stages in terms of saved energy and latency.

- Furthermore, endurance of ReRAMs might be an issue if the input layer is rewritten for every batch of new inputs. Please address the trade-offs of latency vs accuracy vs endurance involved with re-writing frequently the ReRAMs.

The presented data are sufficient for demonstrating the validity of the approach, but it is not possible for the reviewer to evaluate the quality of data presented because of the points that were raised. I suggest that the authors address these points in a very major review of the manuscript.

Here some minor suggestions to further improve the draft:

(line 36-37) ambiguous sentence, rephrase

(line 45) "still facing issues with hardware and energy costs": please argument this statement further and add possibly add citations

(line 46) Why "physics-based feature learning method that is computationally efficient remains elusive"? Please list a few examples where the computational efficiency of emerging techniques based on in-memory computing such as DPE is not sufficient and needs to be changed with new approaches such as the one presented here.

(line 48) "recently" sounds inappropriate since ReRAM research started more than 15 years ago

(line 65) "DDK" reintroduce acronym in the body of the article.

(line 75) reducing the number of parameters of a model while keeping the same accuracy is very strong statement, it should be supported by the specific model for which this happens, the dataset over which it is trained, the method it is compared against to, and an intuitive reason why the parameter reduction happens (just for the first layer, or for all layers?).

(line 107-114) the purpose of this paragraph and why it is relevant to the rest of the discussion is unclear

(line 151) please specify if the error bars in S5 are the 95% intervals

(line 152) please provide literature references for the characterization of this specific device, or extend the supplementary material with information regarding: the writing procedure, the achievable conductance range, the on/off ratio, and the endurance of the device (and how these are tied together in a trade-off of latency vs accuracy vs endurance).

(line 184) Fig 3d, Table S2: MFCC is based on FFT + filters. Please provide comparisons also with methods that don't extract features with heavy pre-processing, if they exist. If they don't exist, provide the reason why preprocessing is required.

(line 187-188) The profiling of GPU, CPU, and ASIC needs to be further extended and presented in detail in the methods section. Also, why it is claimed an improvement of “at least 10^3 ”, instead of providing an exact comparison?

(line 192) please provide literature reference for imprecision-based NN

(line 216) Please discuss also the trade-offs involved with the device variability, such as: What is the precision of the readout? What is the time it takes to extract these features? If this is part of the feedforward mechanism, this can become a relevant contribution to throughput. Also, which peripherals are used around the memory array? How much energy they need per conversion? How many levels the readout needs?

(line 520) this is in the reviewer's opinion the most important subsection; it needs to be rewritten, with added details and clarity

(line 532) please provide an intuitive explanation of why transforming the data using the differential equation that governs the ReRAM is more efficient than competing approaches

(line 293-294 of supplementary material) These numbers are very ambitious for a 180nm technology node, please provide literature reference

Reviewer #2 (Remarks to the Author):

The manuscript is interesting; however, there are issues that should be solved prior to publication.

1.-At the introduction, the industrial use of memristor and current state-of-the-art features should be included, to let the reader have an exact picture of memristors in the nanoelectronics landscape.

2.-The fitting of beta and alpha is messy. Do the authors fit the experimental I-V curve with equations 3 and 4 and determine the parameters alpha and beta to obtain the best fit? Any algorithm to minimize the difference between experimental data and simulations? This issue should be explained in depth. In addition, the I-V curves of different pairs of alpha and beta parameters should be shown in comparison with measured I-V curves, to see the accuracy of the fitting.

3.- The authors should explain how COMSOL simulations are connected to the model in equations 3 and 4? In addition, temperature plots obtained from COMSOL should be included in one or two panels (at the edge of the reset and set points) to describe the role of thermal effects in the resistive switching of the devices. Oxygen vacancies in the Ta₂O₅ layer should also be plotted in some panels for the set and reset processes. The simulation parameters are fitted to reproduce the I-V curves with just numerical criteria or they are physically based on DFT calculations or extracted from the literature? The negative activation energy for conduction in TaO_x can be supported by previous literature?

4.- The role of cycle-to-cycle variability is not explained. How a simple model such as the one presented here can deal with it?, and how it affects the accuracy of the neural network, when variability affects the synaptic weights? See the complexity of variability compact modeling in memristors in the following reference (<https://onlinelibrary.wiley.com/doi/10.1002/aisy.202200338>).

5.-The typical model developed in the classical paper of S. Williams group, and revisited in this manuscript, could be good enough to fit the I-V curves. However, in the back-propagation algorithm, the synaptic weights derivatives are needed (see Figure 4 in the main manuscript). The authors should comment on this issue. A plot on the capacity to reproduced experimental current derivatives would be welcome to clarify this issue.

6.-In the final part of the manuscript, the neural network is described. However, the authors use a confusing writing style where software and hardware parts are intertwined. It is necessary to explain which part of the network are hardware-based, and the corresponding experimental data

obtained, and which part are software-based. How is the connection between experimental data and software-based data done?

Reviewer #2 (Remarks on code availability):

Readme files are included. The code is sorted out.

Reviewer #3 (Remarks to the Author):

The authors fabricated a bilayer memristor with structure TiN/TaO/HfO/TiN, and built a 2 parameter-related (drift-diffusion) device-level model based on its electrical characteristics. The model is further developed to implement feature learning for pattern classification with higher energy efficiency. The hardware-based feature learning is interesting and the topic is suitable for the journal. I would in principle recommend publication, while some minor revisions are still needed.

The following suggestions and questions may be helpful to the authors.

1. Please clarify why the device needs a higher gate voltage for RESET than SET? Especially when the RESET current is lower than the SET current. The same gate voltage in both SET and RESET process could ease the control. See Nature 2023 Hybrid 2D-CMOS microchips for memristive applications. <https://www.nature.com/articles/s41586-023-05973-1>
2. The pulsed I-V-t curves of the memristor are needed apart from Fig.2a, as Fig.2b-c are based on pulse operation, with different voltage/pulse width/initial resistance. So the readers can see the real operation region of the device.
3. Please define “y” more specifically, like resistivity or conductivity? Figure 1 and Fig.S2 should be better explained with real physical meaning. For example, instead of describing the clear mathematic relation like “Fig. S2, α is close related to an asymptote position of y, while β mainly controls the decrease speed... With the increasing of β , the output y decreases more rapidly to asymptote so that it keeps almost constant even under subsequent input signals.”, which could be

further explained as “dopants mobility could induce higher conductance or resistance change” “dopants diffusion coefficient determines the final conductance or resistance level could be achieved”, if I understand correctly.

4. The authors define $\alpha=D$ =dopant diffusion coefficient, $\beta=u/L$ =dopant mobility/L, while D and u normally are fixed number once the material combination has been determined. The authors need to explain how pulse configurations changes the dopant diffusion coefficient and dopant mobility.

If not, please specify which methods could be used to adjust u and D, as variables.

Minor flaws:

1. Please provide a semi-log curve for Fig. S1.
2. Modify Fig. S4c-e using the same scale to show better visualization.

The followings are our point-to-point responses to the reviewer's comments.

Reviewer 1

The submitted work presents an innovative way of using ReRAM kinetics for modelling the generation of features provided to an AI model. The work claims many orders of magnitude better energy and latency than CPU and GPU. Main points of the proposed work are 1- a drift-diffusion model for the ReRAM 2- the experimental characterization of ReRAM devices, and 3- the proposed model embedding is experimentally validated on a custom test system.

While the DDK model itself is not novel, the use of dynamic properties of memristors has not been explored yet in literature, and the idea is itself very interesting. On the other hand, the reviewer finds that the current draft does not address some important points, such as:

Major issues:

Comment 1. *Most of the experimental characterization presented in the supplementary material compares the feature extraction with DDK with the MFCC method. While there is a clear advantage in the feature extraction phase, it is unclear to the reviewer how much the feature extraction itself impacts the energy and latency of training and feed-forwarding for the entire deployed model. If the feature extraction is just a small part, then the results presented are not relevant. The data points provided should provide more details about which model is being accelerated, how do competitive approaches perform, and how much that task is contributing towards the overall energy/latency metrics of the deployed model.*

Answer: Thanks for your suggestion. We highly agree with you that a detailed discussion on the contribution of the feature extraction for the entire model is necessary. Although energy consumption and latency are important metrics for evaluating hardware performance, they may be inappropriate for evaluating the AI models since they typically have a strong correlation with a specific hardware platform. Therefore, from the aspect of AI models, we think that the appropriate metric for evaluating the contribution of the feature extraction for the entire model is the number of operations¹⁻³. For deep neural networks (DNNs), most of the operations are MACs (multiply-accumulate operations)^{4,5}. To better understand the performance of different models in terms of

computational complexity, the number of MACs and their proportions in the feature extraction, classification, and the total number of MACs in the DDK network and other comparative DNNs are listed in Table 1 (Table S6 in the revised manuscript) in the response letter.

Table 1 | Comparison of DDK network with other DNNs in terms of operation numbers (MACs) in feature extraction and classification

Dataset	Network	MACs/ feature extraction	MACs/ classifier	MACS/ Total	Proportion of MACs in the feature extraction
SITW	DDK	256	2.208K	2.464K	0.1039
	SL-CNN	15.322M	0.267M	15.589M	0.9829
AGNews	DDK	38.4K	2.459M	2.497M	0.0154
	Transformer	1.5749G	76.8K	1.575G	0.9999
	LSTM	257.294M	2048	257.296M	0.9999
MNIST	DDK	1568	38.17K	39.738K	0.0395
	LeNet	141.048K	58.92K	199.968K	0.705
UCF	DDK	9.6K	649.984K	659.584K	0.0146
	C3D	38.497G	0.05G	38.547G	0.9987
ModelNet	DDK	2048	166.656K	168.704K	0.012
	VoxNet	58.752M	0.886M	59.638M	0.9851

The statistical results show that in the LeNet-5 for MNIST dataset classification, the MACs in the feature extraction stage account for 70.5% of the total MACs. For other models such as SL-CNN for SITW dataset, VoxNet for ModelNet dataset, LSTM and Transformer for AGNews dataset, and C3D for UCF dataset, the proportion of MACs in the feature extraction stage is over 98% of the total MACs in the network. This finding highlights the significant impact that the feature extraction part has on the overall computational complexity of neural network models. It shows that the feature extraction part is the most computation-intensive one in the entire AI models. On the contrary, the proportion of MACs in the feature extraction stage in the proposed DDK networks only ranges from 1.2% to 10.39%. And the total MACs of the DDK network is also significantly lower than its counterparts.

Therefore, compared with mainstream DNNs, the proposed DDK layer not only simplifies the feature extraction, but also makes the entire AI model more lightweight. This improvement may not only reduce energy consumption, but also decrease latency, thereby enhancing the overall performance of the model when deployed on hardware. This is especially important for resource-constrained application scenarios, such as edge computing.

The corresponding text in the manuscript has been revised as follows: “Among the five types of pattern classification tasks, the DDK network has a similar level of performance to the other networks being compared. The number of network parameters and operations are reduced by up to 296 times (speaker recognition) and 58,441 times (action recognition), respectively. To better disclose the importance of feature extraction in the neural networks and demonstrate the significance of the proposed DDK layer in simplifying the networks, the number of MACs and their proportions in the feature extraction, classification, and the total number of MACs in the DDK network and other comparative DNNs are listed in Table S6. The statistical results show that in the LeNet-5 for MNIST dataset classification, the MACs in the feature extraction stage account for 70.5% of the total MACs. For other models such as SL-CNN for SITW dataset, VoxNet for ModelNet dataset, LSTM and Transformer for AGNews dataset, and C3D for UCF dataset, the proportion of MACs in the feature extraction stage is over 98% of the total MACs in the network. This finding highlights the significant impact that the feature extraction part has on the overall computational complexity of neural network models. It shows that the feature extraction part is the most computation-intensive one in the entire AI models. On the contrary, the proportion of MACs in the feature extraction stage in the proposed DDK networks only ranges from 1.2% to 10.39%. And the total MACs of the DDK network is also significantly lower than its counterparts.

Therefore, compared with mainstream DNNs, the proposed DDK layer not only simplifies the feature extraction, but also makes the entire AI model more lightweight. This improvement may not only reduce energy consumption, but also decrease latency, thereby enhancing the overall performance of the model when deployed on hardware. This is especially important for resource-constrained application scenarios, such as edge computing.

Supplementary information

Table S6 | Comparison of DDK network with other DNNs in terms of operation numbers (MACs) in feature extraction and classification

Dataset	Network	MACs/ feature extraction	MACs/ classifier	MACS/ Total	Proportion of MACs in the feature extraction
SITW	DDK	256	2.208K	2.464K	0.1039
	SL-CNN	15.322M	0.267M	15.589M	0.9829
AGNews	DDK	38.4K	2.459M	2.497M	0.0154
	Transformer	1.5749G	76.8K	1.575G	0.9999
	LSTM	257.294M	2048	257.296M	0.9999
MNIST	DDK	1568	38.17K	39.738K	0.0395
	LeNet	141.048K	58.92K	199.968K	0.705
UCF	DDK	9.6K	649.984K	659.584K	0.0146
	C3D	38.497G	0.05G	38.547G	0.9987
ModelNet	DDK	2048	166.656K	168.704K	0.012
	VoxNet	58.752M	0.886M	59.638M	0.9851

”

Comment 2. *Data points are provided for competitive CPU, GPU, and ASIC approaches, but the reviewer finds disappointing that insufficient information is given about how these data points are extracted. While there is no established framework for simulating easily of all these components, it is the author's responsibility to show that the comparisons were fair; IMC will only be really accepted by all the hardware community if demonstrations of its usefulness come with great clarity and transparency.*

Answer: Thanks for your comments. MFCC and the proposed DDK method both extract features by the grain size of frame. When comparing performance among different hardware, we focus on the energy consumption per frame. From the point of fairness, the hardware performance benchmark only takes into account the feature extraction, other stages like classification will be

ignored. And for the different kinds of hardware in the reference papers, like ASIC, FPGA, DSP, GPU, and CPU, the energy consumption is calculated in different ways. Here we represent the computing details and modify some parts of the benchmark in Table S2 in the revised manuscript.

(1) ASIC: These ASIC circuits are newly proposed, and the performance details are given in the paper. The energy consumption can be calculated indirectly or obtained directly based on the information provided in the papers.

For ASIC⁶, the latency of MFCC per frame is 16 ms (4-stage pipeline with 64 ms total latency) and the power of MFCC is 0.34 μ W. As a result, the energy consumption of MFCC per frame is 5.44 nJ (16 ms \times 0.34 μ W).

For ASIC⁷, the latency of MFCC per frame is calculated from the data in the paper, which is 2 ms (calculated from the time sampled from Figure in the reference paper, the start time of MFCC is 4.5 ms, and the end time of MFCC is 6.5 ms). In the same way, the average power of MFCC is calculated, which is 21.23 μ W (11 points of the power consumption sampled from 4.5 ms to 6.5 ms to calculate the average power of MFCC, which are 21 μ W, 22.5 μ W, 26 μ W, 29 μ W, 30 μ W, 30 μ W, 29 μ W, 15 μ W, 13 μ W, 9 μ W, 9 μ W). As a result, the energy consumption of MFCC per frame is 42.46 nJ (2 ms \times 21.23 μ W).

For ASIC⁸, the performance can be directly obtained. The energy consumption of MFCC per frame is 0.72 μ J, and the latency of MFCC per frame is 45.79 μ s.

(2) FPGA: FPGAs are mature products, and their performance details may not be given in the paper. However, we can evaluate power using AMD Power Estimator (XPE) to get a rough power consumption result, which is based on the resources used in the paper. Then the energy consumption of the MFCC per frame can be calculated using the power and the latency mentioned in the paper.

For FPGA⁹, the MFCC latency is 14601 cycles per frame, which is 14601 \times 20 ns (clock frequency is 50 MHz) = 292 μ s. In AMD Power Estimator (XPE), we set the device as XC4VLX15 (the same device as the one used in the paper), and the logic part as resources mentioned in the paper, such as the number of LUT and Flip flops, and then we get an estimated power (172 mW), as shown in Fig. 1. Finally, the energy consumption of MFCC per frame is calculated to be 50.22 μ J (172 mW \times 292 μ s).

Fig. 1 | The result of XPE for XC4VLX15

For FPGA¹⁰, the MFCC latency is directly mentioned in the paper, which is 285.44 μs. In AMD Power Estimator (XPE), we set the device as XC3S2000, and the logic part as resources mentioned in the paper, such as the number of LUT and Flip flops, and then we get an estimated power (234 mW), as shown in Fig. 2. Finally, the energy consumption of MFCC per frame is calculated to be 66.79 μJ (234 mW×285.44 μs).

Fig. 2 | The result of XPE for XC3S2000

(3) DSP, GPU, and CPU: DSPs, GPUs, and CPUs are also mature products, and their performance details may not be given in the paper. However, their power consumptions can be evaluated from the product specifications. Then, the energy consumption of MFCC can be simply calculated with the latency of MFCC mentioned in the paper.

For GPU¹¹, the MFCC latency per frame is 9.17 μ s. The power consumption of GTX580 GPU from product specification is 244 W¹². Finally, the energy consumption of MFCC per frame is 2.24 mJ (244 mW \times 9.17 μ s).

For DSP¹³, the MFCC latency is 0.912 ms. The power consumption of TMS320C6713 from product specification is 1.067 W¹⁴. Finally, the energy consumption of MFCC per frame is 0.973 mJ (1.067 W \times 0.912 ms).

For CPU¹⁰, the MFCC latency is 238.77 μ s. The power consumption of Intel P4 CPU from product specification is 57.8 W¹⁵. Finally, the energy consumption of MFCC per frame is 13.8 mJ (57.8 mW \times 238.77 μ s).

According to the above calculations, we revised the Table S2 and Fig. 3d in the manuscript.

The corresponding text in the manuscript has been revised as follows: “

Supplementary information

Benchmark of the MFCC hardware

MFCC and the proposed DDK method both extract features by the grain size of frame. When comparing performance among different hardware, we focus on the energy consumption per frame. From the point of fairness, the hardware performance benchmark only takes into account the feature extraction, other stages like classification will be ignored. And for the different kinds of hardware in the reference papers, like ASIC, FPGA, DSP, GPU, and CPU, the energy consumption is calculated in different ways. Here we represent the computing details and modify some parts of the benchmark in Table S2.

(1) ASIC: These ASIC circuits are newly proposed, and the performance details are given in the paper. The energy consumption can be calculated indirectly or obtained directly based on the information provided in the papers.

For ASIC⁶, the latency of MFCC per frame is 16 ms (4-stage pipeline with 64 ms total latency) and the power of MFCC is 0.34 μ W. As a result, the energy consumption of MFCC per frame is 5.44 nJ (16 ms \times 0.34 μ W).

For ASIC⁷, the latency of MFCC per frame is calculated from the data in the paper, which is 2 ms (calculated from the time sampled from Figure in the reference paper, the start time of MFCC is 4.5 ms, and the end time of MFCC is 6.5 ms). In the same way, the average power of MFCC is calculated, which is 21.23 μ W (11 points of the power consumption sampled from 4.5 ms to 6.5 ms to calculate the average power of MFCC, which are 21 μ W, 22.5 μ W, 26 μ W, 29 μ W, 30 μ W, 30 μ W, 29 μ W, 15 μ W, 13 μ W, 9 μ W, 9 μ W). As a result, the energy consumption of MFCC per frame is 42.46 nJ (2 ms \times 21.23 μ W).

For ASIC⁸, the performance can be directly obtained. The energy consumption of MFCC per frame is 0.72 μ J, and the latency of MFCC per frame is 45.79 μ s.

(2) FPGA: FPGAs are mature products, and their performance details may not be given in the paper. However, we can evaluate power using AMD Power Estimator (XPE) to get a rough power consumption result, which is based on the resources used in the paper. Then the energy consumption of the MFCC per frame can be calculated using the power and the latency mentioned in the paper.

For FPGA⁹, the MFCC latency is 14601 cycles per frame, which is 14601 \times 20 ns (clock frequency is 50 MHz) = 292 μ s. In AMD Power Estimator (XPE), we set the device as XC4VLX15 (the same device as the one used in the paper), and the logic part as resources mentioned in the paper, such as the number of LUT and Flip flops, and then we get an estimated power (172 mW). Finally, the energy consumption of MFCC per frame is calculated to be 50.22 μ J (172 mW \times 292 μ s).

For FPGA¹⁰, the MFCC latency is directly mentioned in the paper, which is 285.44 μ s. In AMD Power Estimator (XPE), we set the device as XC3S2000, and the logic part as resources mentioned in the paper, such as the number of LUT and Flip flops, and then we get an estimated power (234 mW). Finally, the energy consumption of MFCC per frame is calculated to be 66.79 μ J (234 mW \times 285.44 μ s).

(3) DSP, GPU, and CPU: DSPs, GPUs, and CPUs are also mature products, and their performance details may not be given in the paper. However, their power consumptions can be evaluated from the product specifications. Then, the energy consumption of MFCC can be simply calculated with the latency of MFCC mentioned in the paper.

For GPU¹¹, the MFCC latency per frame is 9.17 μ s. The power consumption of GTX580 GPU from product specification is 244 W¹². Finally, the energy consumption of MFCC per frame is 2.24 mJ (244 mW \times 9.17 μ s).

For DSP¹³, the MFCC latency is 0.912 ms. The power consumption of TMS320C6713 from product specification is 1.067 W¹⁴. Finally, the energy consumption of MFCC per frame is 0.973 mJ (1.067 W \times 0.912 ms).

For CPU¹⁰, the MFCC latency is 238.77 μ s. The power consumption of Intel P4 CPU from product specification is 57.8 W¹⁵. Finally, the energy consumption of MFCC per frame is 13.8 mJ (57.8 mW \times 238.77 μ s).

Table S2 | Comparison of various hardware for MFCC feature extraction

	Tech. node	Name	Frequency	Power	Latency	Energy
This work	180 nm	N.A.	100 MHz	80 μ W	10 ns	0.8 pJ
ASIC ⁶	28 nm	N.A.	0.04 MHz	0.34 μ W	16 ms	5.44 nJ
ASIC ⁷	65 nm	N.A.	0.25 MHz	21.23 μ W	2 ms	42.46 nJ
ASIC ⁸	180 nm	N.A.	N.A.	N.A.	45.79 μ s	0.72 μ J
FPGA ⁹	90 nm	XC4VLX15	50 MHz	172 mW	292 μ s	50.22 μ J
FPGA ¹⁰	90 nm	XC3S2000	50 MHz	234 mW	285.44 μ s	66.79 μ J
GPU ¹¹	40 nm	GTX580	772 MHz(Core)	244 W	9.17 μ s	2.24 mJ
DSP ¹³	130 nm	TMS320C6713	225 MHz	1.067 W	0.912 ms	0.973 mJ

CPU ¹⁰	180 nm	Intel P4	1.5 GHz	57.8 W	238.77 μ s	13.8 mJ
-------------------	--------	----------	---------	--------	----------------	---------

Fig. 2d Comparison of the proposed feature map construction method in terms of hardware latency and energy consumption with main-stream MFCC-based hardware. Our method shows significantly reduced latency and energy consumption.”

Comment 3. *In the reviewer’s opinion, the subsection about backpropagation in the Methods section is not sufficient to address this subject as is. It would need a dedicated subsection in the Results and Discussion section, with the purpose of providing an intuition of how the feature are feed-forwarded and backpropagated along the new hardware implementation of the model based on using the differential equation that governs the ReRAM and why this is, from the point of view of the entire deployed model, more computationally efficient than other ReRAM-based approaches to IMC. This discussion would allow to also introduce how much the DDK method impacts the following stages in terms of saved energy and latency.*

Answer: Thank you for the insightful suggestion. We have accepted your suggestion and added a new section in the Results and Discussion to explain the forward and back propagation process of DDK network. This section primarily introduces the two stages of hybrid training for DDK neural networks: off-chip pre-training and on-chip fine-tuning, and details the specific implementation processes of forward and backward propagation during these two stages. Finally, it presents the advantages of this training technique in terms of energy consumption and latency.

The specific steps for off-chip pre-training are as follows: During the off-chip pre-training phase, the DDK neural network is trained off-chip while considers the hardware constraints of α and β , according to the experimental results in Fig. 2b-d in the original manuscript. The backpropagation algorithm is employed to update the network parameters. The backpropagation process involves the following steps: Firstly, the input data is propagated forward through each layer of the network to obtain the predicted output. Then, the loss between the prediction and the actual target is calculated. Next, the gradients of the loss function with respect to α and β in the DDK layer, as well as each weight in the classification layer, are computed using the chain rule. Finally, the computed gradients are used to update α and β in the DDK layer and the weights in the classification layer, aiming to minimize the loss function.

The specific implementation process for the on-chip fine-tuning phase is as follows:

1. Initialization: With the parameters α and β of the DDK layer already pre-trained and fixed, appropriate pulse configurations for performing feature extraction in the memristor DDK layer are selected according to the experimental results. The pre-trained weights of the classification layer are prepared and ready to be mapped onto the conductance states of the RRAM devices.
2. Mapping pre-trained weights: The pre-trained weights of the classification layer are written into the RRAM devices by adjusting their conductance values using write-and-verify technique. The write-and-verify technique, weight updating and MVM output accuracy are shown in Fig. S12 in the revised manuscript. This process ensures that the initial state of the RRAM devices corresponds to the optimized weights obtained during the off-chip pre-training phase.
3. Forward pass: Given a set of input data, the input features are first processed by the DDK layer. Feature maps are obtained by applying electrical pulses with such configuration to memristive devices in the DDK layer and record their conductance evolution with the pulses. These feature maps are then fed into the memristor classification layer. The computation results from the RRAM arrays in the classification layer are used to generate predictions.
4. Loss calculation: The predictions generated by the memristor classification layer are compared against the actual targets to compute the loss function. This loss value quantifies the error between the model predictions and the desired outputs.

5. Backward pass and gradient calculation: The loss function is then differentiated with respect to the conductance values of the RRAM devices in the classification layer. This involves calculating the gradients of the loss function with respect to the RRAM output, which in turn requires calculating the gradients through the entire network (but focusing on the classification layer for updates). Since the DDK layer parameters are fixed, the gradients are only propagated through the classification layer.
6. In-situ weight update: Using the calculated gradients, the conductance values of the RRAM devices are updated in-situ. This is done by applying voltage pulses to the devices in a manner that modifies their conductance based on the gradient direction and magnitude. The goal is to reduce the loss function by adjusting the weights (represented by the RRAM conductance) in the direction that improves the model predictions.
7. Iteration and convergence: The forward pass, loss calculation, backward pass, and in-situ weight update steps are repeated iteratively until the model converges.
8. Evaluation: Once the fine-tuning process is complete, the model performance is evaluated on a validation or test dataset to assess its generalization ability and robustness.

From the aspect of energy and latency, the advantage of our approach is that the feature extraction is performed by the differential equation that is defined by only two parameters, namely alpha and beta. While in the competing approaches, such as attention mechanism, convolution and recurrent connections, the feature extraction is defined by a set of weight matrices. As discussed in Table S8 in the revised manuscript, the layer complexity ($O(n \cdot d)$) of our approach is the lowest among self-attention, recurrent, convolution and fully connected layers. Therefore, not only the feature extraction is simplified, but also the subsequent classification layer is minimized due to the small number of features to be processed. This leads to greatly reduced number of parameters and computational operations (Fig. 5b in the revised manuscript), suggesting saved energy and latency when deploying such models on hardware.

However, other RRAM-based approaches are based on existing AI models. They accelerate the matrix-vector multiplication (MVM) based on computing-in-memory, but cannot decrease the number of MVM operation in the models. On the contrary, our approach not only accelerates the

MVM and DDK operators, but also greatly reduces the number of MVM operations by replacing the conventional feature extraction layers using the DDK layers. Therefore, as shown in Fig. 5c in the revised manuscript, our approach can greatly save energy compared with RRAM-based deep neural networks.

The corresponding text in the manuscript has been revised as follows: “Fig. 4a shows a typical DDK network structure for pattern recognition. Fig. 4b shows network training procedure, in which the parameters α and β are learned via back-propagation method (detailed in Methods). After modeling, the network is deployed on the hardware (Fig. S11) via a compiler (detailed in Supplementary Notes). For task scheduling, a hybrid training technique that involves off-chip pre-training and on-chip fine-tuning is proposed (Fig. 4c) considering the large variation in DDK. In this technique, the DDK layer is trained *ex-situ*. The reason is that in hardware implementation, α and β are implicit and obtained by curve fitting of the conductance evolution with time, but not by a direct conductance measurement as in the classification layer, thus evaluation and tuning of α and β *in-situ* is time-consuming. The significant variations of parameters α and β in *in-situ* training may introduce instability, impeding the network ability to converge rapidly.

The off-chip pre-training follows the procedure of back-propagation method in Fig. 4b. It includes the hardware constraints of α and β in training, according to the experimental results in Fig. 2b-d. The on-chip fine-tuning phase is as follows: (1) initialization: with the parameters α and β of the DDK layer already pre-trained, appropriate pulse configurations for performing feature extraction in the memristor DDK layer are selected according to the experimental results. (2) Mapping pre-trained weights: the pre-trained weights of the classification layer are written into the RRAM devices by adjusting their conductance values using write-and-verify technique. The write-and-verify technique, weight updating and MVM output accuracy are shown in Fig. S12. This process ensures that the initial state of the memristor or resistance random access memory (RRAM) devices corresponds to the optimized weights obtained during the off-chip pre-training phase. (3) Forward pass: given a set of input data, the input features are firstly processed by the DDK layer. Feature maps are obtained by applying electrical pulses with such configuration to memristive devices in the DDK layer and record their conductance evolution with the pulses. These feature

maps are then fed into the memristor classification layer. The computation results from the RRAM arrays in the classification layer are used to generate predictions. (4) Loss calculation: the predictions generated by the memristor classification layer are compared against the actual targets to compute the loss function. (5) Backward pass and gradient calculation: the loss function is then differentiated with respect to the conductance values of the RRAM devices in the classification layer. This involves calculating the gradients of the loss function with respect to the RRAM output, which in turn requires calculating the gradients through the entire network (but focusing on the classification layer for updates). Since the DDK layer parameters are fixed, the gradients are only propagated through the classification layer. (6) *In-situ* weight update: using the calculated gradients, the conductance values of the RRAM devices are updated *in-situ*. The goal is to reduce the loss by adjusting the weights (represented by the RRAM conductance) in the direction that improves the model predictions. (7) Iteration and convergence: the forward pass, loss calculation, backward pass, and in-situ weight update steps are repeated iteratively until the model converges. (8) Evaluation: once the fine-tuning process is complete, the model performance is evaluated on a validation or test dataset to assess its generalization ability and robustness.

The advantage of our approach is that the feature extraction is performed by the differential equation that is defined by only two parameters, namely alpha and beta. While in the competing approaches, such as attention mechanism, convolution and recurrent connections, the feature extraction is defined by a set of weight matrices. Therefore, not only the feature extraction is simplified, but also the subsequent classification layer is minimized due to the small number of features to be processed. Moreover, our approach also accelerates the feature extraction and classification based on computing-in-memory. For other RRAM-based approaches that implements existing AI models, they only accelerate the MVM, but cannot simplify the feature extraction and classification layers.”

Comment 4. *Furthermore, endurance of ReRAMs might be an issue if the input layer is rewritten for every batch of new inputs. Please address the trade-offs of latency vs accuracy vs endurance involved with re-writing frequently the ReRAMs.*

Answer: Thanks for your comments. We agree with you that the endurance of RRAMs may be an

issue causing performance degradation in their application scenarios, such as memory and artificial neural network. Therefore, the trade-offs between latency, accuracy, and endurance are indeed worthy of further investigation. Since the latency involved in re-writing the RRAMs is directly determined by the pulse width that is used by the writing pulse to program the device, the trade-offs are studied by investigating the relations between pulse width, endurance and neural network accuracy. To be specific, we study the influence of pulse width on device endurance, and then the impact of conductance degradation found in the endurance test on the classification accuracy of the DDK neural network. Since the re-writing occurs primarily in the DDK layer, the device conductance in the classification layers is assumed to be constant to simplify the study and highlight the effect of re-writing. The study is performed based on simulation using models extracted from experimental data following the specific analysis process shown in Fig. 3.

We firstly study the influence of pulse width on device endurance. To make the endurance test more reliable, the endurance of the device is tested using the same method as in the DDK feature extraction. In this method, a large and constant gate voltage is applied so that the primary role of gate voltage is to select the specific device but not control the conductance states by limiting the compliant currents. Therefore, the conductance states are determined by the voltages applied on BL and SL. For the LTP process, the voltage of WL for set operation is fixed at 2 V, while the voltage of BL increases gradually (from 0.7 V to 1.7 V) and SL is ground. For the LTD process, the voltage of WL for reset operation is fixed at 3 V, while the voltage of SL increases gradually (from -1.4 V to -2.4 V) and BL is ground. The pulse width in the writing operation is fixed at 50 ns, 200 ns, 500 ns, respectively. The read operation uses a fixed voltage (0.2 V) and pulse width (20 μ s). Fig. 4 shows the endurance test results of the devices. Fig. 4a shows the LTD and LTP characteristics around initial, 10^3 , 10^6 , 10^7 , and 10^9 pulses. The conductance range and on/off ratio of 25 devices for each pulse width configuration in the endurance test are statistically calculated in Fig. 4b and Fig. 4c, respectively. Though the dynamic range and on/off ratio degrade as the increase of update pulses, there is an on/off ratio larger than 1 after 10^9 update pulses. Previous study has demonstrated that under weak weight update pulses in a low resistance region, the incremental switching cycles of RRAM can be increased for more than 5 orders of magnitude (up to 10^{11}) compared with full window switching under strong programming pulses¹⁶. Although the

switching region in our endurance test is not constrained in a low resistance region, the conclusions are still in consistence with the previous study. Besides, it can also be observed that the high resistance states in the three pulse configurations all shows firstly a upward then a downward trend, and the trend of the 500 ns configuration is flatter in the pulse number $\leq 10^7$ than the other two. This may be attribute to the effect of the “second order memristor”, where non-gradual resistive switching behavior is typically obersevd under long pulse width, because sufficient heat accumulation can occur even inside a single set pulse when the pulse is sufficiently long¹⁷.

Secondly, to study the impact of conductance degradation found in the endurance test on the classification accuracy of the DDK neural network, the effect of conductance degradation on neural network parameters, namely, α and β , should be modeled. Based on experimental data in Fig. 2c and 2d in the original manuscript, we respectively demonstrate the specific impacts of different pulse widths and different initial resistances on the α and β parameters. By observing Fig. 2c in the original manuscript, it can be clearly seen that as the pulse width gradually increases, the β exhibits an increasing trend, while the α remains almost unchanged throughout this process. Similarly, in Fig. 2d in the original manuscript, as the initial resistance continues to rise, the β also shows a corresponding increasing trend and eventually stabilizes, while the α value remains relatively stable.

Given this, we will next conduct an in-depth analysis of the specific impacts of pulse width and initial resistance on the β , and explore how these impacts further affect classification accuracy. During the network inference process, we can adjust parameters such as pulse amplitude, initial resistance, and pulse width to match the β of the device with the β obtained from network training. However, it is worth noting that after multiple operations, the β will inevitably undergo degradation. Therefore, we analyze in detail the effect of β decay on the accuracy with the increase of pulse number at different pulse widths.

(1) Evaluation of the changes in initial resistance (high conductance state) at different pulse widths. Here we use the average high conductance states of the 25 test devices for each pulse configuration

as the initial resistance. Changes in initial resistance at different pulse widths as shown in Fig. 6a. The initial resistance shown on the y-axis in Fig. 6a is based on the average of the initial resistances measured from average 25 devices in Fig. 4. In consistence with previous analysis, the fluctuation of the initial resistance for pulse number $\leq 10^7$ is the smallest when the pulse width is 500 ns. After 10^7 cycles, the initial resistances of the three configurations all degrades.

(2) Modeling the relation between initial resistance and β at arbitrary pulse width. For simplicity, we construct a linear model between initial resistance and β using the relation of initial resistance and β at 5 K Ω , 10 K Ω and 15 K Ω initial resistances in Fig. 2d in the original manuscript. β exhibits an increasing trend as the initial resistance increases. Since this extracted linear model is based on the data measured at pulse width of 100 ns, this model should be extended to situations where arbitrary pulse width is used. The linear model we construct at pulse width of 100 ns is shown as follows:

$$\beta = (9.66 \times 10^{-5}) \times R_{init} - 0.21 \quad (1)$$

where R_{init} is the initial resistance. We assume that at arbitrary pulse width, the slope of the extended linear model is unchanged, but the data point it passes through is determined by another linear model (Fig. 5) between pulse width and β , using the experimental data in Fig. 2c in the original manuscript. The linear model between pulse width and β is shown as follows:

$$\beta = 0.00134 \times T + 0.14378 \quad (2)$$

where T is the pulse width. Using the slope and the data points it passes through, the extended linear model that describes the relation between initial resistance and β at arbitrary pulse width can be constructed. From the experimental data, the β values typically fall within the range of 0 to 1, for β values that exceed this range, they are truncated directly. The extended linear model at pulse width of 50ns 200ns and 500ns is shown as equations (3-5):

$$\beta = (9.66 \times 10^{-5}) \times R_{init} - 0.27 \quad (3)$$

$$\beta = (9.66 \times 10^{-5}) \times R_{init} - 0.07 \quad (4)$$

$$\beta = (9.66 \times 10^{-5}) \times R_{init} + 0.33 \quad (5)$$

(3) Modeling the change rate of β at arbitrary pulse width. The change rate of β with the number of pulses at different pulse widths can be derived from the extended linear model between β and the initial resistance, as well as the degradation process of initial resistance with pulse number (Fig. 6a). As shown in Fig. 6b, the rate of β change fluctuates the most at pulse width of 50 ns and the least at pulse width of 500 ns. Because, even though the initial resistance experiences a relatively high rate of change after the number of pulses reaches 10^9 at pulse width of 500 ns, the β value is truncated, resulting in a small change rate.

(4) Modeling the decay process of β obtained from the pre-trained network. Here, we set the decay rate of the β obtained from the pre-trained network to be the change rate of β in step (3). Fig. 6c shows the decay process of the β obtained from the pre-trained network with an increasing number of pulses at different pulse widths. It can be observed that under different pulse widths, the decay of β exhibits a trend of first increasing and then decreasing. Specifically, when the pulse width is 50 ns, the fluctuation amplitude is the largest, while when the pulse width reaches 500 ns, the fluctuation amplitude is the smallest.

(5) Study of β decay on classification accuracy. We test the impact of β decay on classification accuracy on the SITW dataset in a 10-speaker recognition task at different pulse widths (as shown in Fig. 6d). It can be observed that the classification accuracy exhibits a downward trend as the number of pulses increases for all pulse widths. Particularly, when the pulse width is 50 ns, the decline in classification accuracy is the most significant. As the pulse width increases, the downward trend in classification accuracy gradually slows down. This indicates that in endurance test, the pulse width used for re-writing has a significant impact on accuracy degradation. Specifically, small pulse width has low latency, but the degradation in classification accuracy is significant and its endurance is poor; large pulse width has high latency, but the degradation in classification accuracy is less pronounced and its endurance is good, especially when the pulse number is $\leq 10^7$.

In conclusion, probably because of the “second order memristor”, the write pulse with long width leads to non-gradual resistive switching, making the device reaches a stable saturation state from the initial (Fig. 7). As a result, the decrease in classification accuracy is suppressed. However, the long pulse width may result in a long write latency. This demonstrates that there is a trade-off between latency and accuracy, considering the endurance of memristor devices.

Fig. 3 | Analysis flowchart for latency and accuracy in endurance test.

Fig. 4 | Endurance test of the memristor device. a, Analog switching of the full dynamic range of different operation numbers (pulse width is 200 ns). **b,** Achievable conductance range for different pulses with width of 50 ns, 200 ns and 500 ns. The maximum and minimum conductance of each cycle are collected to obtain the conductance range. We collect the analog switching of 25 devices for each pulse width configuration and display the boxplot of the conductance range, which shows that the achievable conductance range is degraded as the pulse number increases. **c,** On/off ratio of the 25 devices after different numbers of update pulses, the on/off ratio can still be larger than 1 after 10^9 incremental switching.

Fig. 5 | Fitting curve of β versus pulse width

Fig. 6 | The relation of latency and accuracy in the endurance test in audio classification task.

a. Changes in initial resistance at different pulse widths. The values are average high conductance states of 25 test devices for each pulse configuration. **b.** The change rate of β at different pulse widths. **c.** The decay process of β from the pre-trained network at different pulse widths. **d.** The effect of β decay on classification accuracy in audio classification at different pulse widths.

Fig. 7 | Schematic of the endurance of a second order memristor at different pulse widths. For short pulses, gradual switching usually occurs, resulting in instability in the endurance test. While for long pulses, the fluctuation is suppressed because of the more abrupt resistive switching.

The corresponding text in the manuscript has been revised as follows: “The effect of endurance on neural network performance is studied in Supplementary Notes, Fig. S18 and Fig. S19. Probably because of the “second order memristor”¹⁸, the write pulse with long width in the DDK layer leads to non-gradual resistive switching, making the device reaches a stable saturation state from the initial. As a result, the decrease in classification accuracy is suppressed. However, the long pulse width may result in a long write latency. This demonstrates that there is a trade-off between latency and accuracy, considering the endurance of memristor devices.

Supplementary Notes

The effect of endurance on neural network performance

The endurance of RRAMs may be an issue causing performance degradation in their application scenarios, such as memory and artificial neural network. Therefore, the trade-offs between latency, accuracy, and endurance are further investigation. Since the latency involved in re-writing the RRAMs is directly determined by the pulse width that is used by the writing pulse to program the device, the trade-offs are studied by investigating the relations between pulse width, endurance and neural network accuracy. To be specific, we study the influence of pulse width on device endurance, and then the impact of conductance degradation found in the endurance test on the classification accuracy of the DDK neural network. Since the re-writing occurs primarily in the DDK layer, the device conductance in the classification layers is assumed to be constant to simplify

the study and highlight the effect of re-writing. The study is performed based on simulation using models extracted from experimental data.

We firstly study the influence of pulse width on device endurance. To make the endurance test more reliable, the endurance of the device is tested using the same method as in the DDK feature extraction. In this method, a large and constant gate voltage is applied so that the primary role of gate voltage is to select the specific device but not control the conductance states by limiting the compliant currents. Therefore, the conductance states are determined by the voltages applied on BL and SL. For the LTP process, the voltage of WL for set operation is fixed at 2 V, while the voltage of BL increases gradually (from 0.7 V to 1.7 V) and SL is ground. For the LTD process, the voltage of WL for reset operation is fixed at 3 V, while the voltage of SL increases gradually (from -1.4 V to -2.4 V) and BL is ground. The pulse width in the writing operation is fixed at 50 ns, 200 ns, 500 ns, respectively. The read operation uses a fixed voltage (0.2 V) and pulse width (20 μ s). Fig. S18 shows the endurance test results of the devices. Fig. S18a shows the LTD and LTP characteristics around initial, 10^3 , 10^6 , 10^7 , and 10^9 pulses. The conductance range and on/off ratio of 25 devices for each pulse width configuration in the endurance test are statistically calculated in Fig. S18b and Fig. S18c, respectively. Though the dynamic range and on/off ratio degrade as the increase of update pulses, there is an on/off ratio larger than 1 after 10^9 update pulses. Previous research has demonstrated that, in contrast to full-window switching under strong programming pulses, employing weaker weight update pulses in the low-resistance region can enhance the incremental switching cycles of RRAM by over five orders of magnitude, reaching approximately 10^{11} cycles¹⁶. Although the switching region in our endurance test is not constrained in a low resistance region, the conclusions are still in consistence with the previous study. Besides, it can also be observed that the high resistance states in the three pulse configurations all shows firstly an upward then a downward trend, and the trend of the 500 ns configuration is flatter in the pulse number $\leq 10^7$ than the other two. This may be attribute to the effect of the “second order memristor”, where non-gradual resistive switching behavior is typically observed under long pulse width, because even a single set pulse can result in substantial heat accumulation if its duration is sufficiently extended¹⁷.

Secondly, to study the impact of conductance degradation found in the endurance test on the classification accuracy of the DDK neural network, the effect of conductance degradation on neural network parameters, namely, α and β , should be modeled. Based on experimental data in Fig. 2c and 2d in the original manuscript, we respectively demonstrate the specific impacts of different pulse widths and different initial resistances on the α and β parameters. By observing Fig. 2c in the original manuscript, it can be clearly seen that as the pulse width gradually increases, the β exhibits an increasing trend, while the α remains almost unchanged throughout this process. Similarly, in Fig. 2d in the original manuscript, as the initial resistance continues to rise, the β also shows a corresponding increasing trend and eventually stabilizes, while the α value remains relatively stable.

Given this, we will next conduct an in-depth analysis of the specific impacts of pulse width and initial resistance on the β , and explore how these impacts further affect classification accuracy. During the network inference process, we can adjust parameters such as pulse amplitude, initial resistance, and pulse width to match the β of the device with the β obtained from network training. However, it is worth noting that after multiple operations, the β will inevitably undergo degradation. Therefore, we analyze in detail the effect of β decay on the accuracy with the increase of pulse number at different pulse widths.

(1) Evaluation of the changes in initial resistance (high conductance state) at different pulse widths. Here we use the average high conductance states of the 25 test devices for each pulse configuration as the initial resistance. Changes in initial resistance at different pulse widths are shown in Fig. S19a. The initial resistance shown on the y-axis in Fig. S19a is based on the average of the initial resistances measured from average 25 devices in Fig. S18. In consistence with previous analysis, the fluctuation of the initial resistance for pulse number $\leq 10^7$ is the smallest when the pulse width is 500 ns. After 10^7 cycles, the initial resistances of the three configurations all degrades.

(2) Modeling the relation between initial resistance and β at arbitrary pulse width. For simplicity, we construct a linear model between initial resistance and β using the relation of initial resistance

and β at 5 K Ω , 10 K Ω and 15 K Ω initial resistances in Fig. 2d. β exhibits an increasing trend as the initial resistance increases.

Since this extracted linear model is based on the data measured at pulse width of 100 ns, this model should be extended to situations where arbitrary pulse width is used. The linear model we construct at pulse width of 100 ns is shown as follows:

$$\beta = (9.66 \times 10^{-5}) \times R_{init} - 0.21 \quad (\text{S7})$$

where R_{init} is the initial resistance. We assume that at arbitrary pulse width, the slope of the extended linear model is unchanged, but the data point it passes through is determined by another linear model between pulse width and β , using the experimental data in Fig. 2c. The linear model between pulse width and β is shown as follows:

$$\beta = 0.00134 \times T + 0.14378 \quad (\text{S8})$$

where T is the pulse width. Using the slope and the data points it passes through, the extended linear model that describes the relation between initial resistance and β at arbitrary pulse width can be constructed. The extended linear model at pulse width of 50 ns, 200 ns and 500 ns is shown as S9-S11:

$$\beta = (9.66 \times 10^{-5}) \times R_{init} - 0.27 \quad (\text{S9})$$

$$\beta = (9.66 \times 10^{-5}) \times R_{init} - 0.07 \quad (\text{S10})$$

$$\beta = (9.66 \times 10^{-5}) \times R_{init} + 0.33 \quad (\text{S11})$$

From the experimental data, the β values typically fall within the range of 0 to 1, for β values that exceed this range, they are truncated directly.

(3) Modeling the change rate of β at arbitrary pulse widths. The change rate of β with the number of pulses at different pulse widths can be derived from the extended linear model between β and the initial resistance, as well as the degradation process of initial resistance with pulse number (Fig. S19a). As shown in Fig. S19b, the rate of β change fluctuates the most at pulse width of 50 ns and the least at pulse width of 500 ns. Because, even though the initial resistance experiences a

relatively high rate of change after the number of pulses reaches 10^9 at pulse width of 500 ns, the β value is truncated, resulting in a small change rate.

(4) Modeling the decay process of β obtained from the pre-trained network. Here, we set the decay rate of the β obtained from the pre-trained network to be the change rate of β in step (3). Fig. S19c shows the decay process of the β obtained from the pre-trained network with an increasing number of pulses at different pulse widths. It can be observed that under different pulse widths, the decay of β exhibits a trend of first increasing and then decreasing. Specifically, when the pulse width is 50 ns, the fluctuation amplitude is the largest, while when the pulse width reaches 500 ns, the fluctuation amplitude is the smallest.

(5) Study of β decay on classification accuracy. We test the impact of β decay on classification accuracy on the SITW dataset in a 10-speaker recognition task at different pulse widths (as shown in Fig. S19d). It can be observed that the classification accuracy exhibits a downward trend as the number of pulses increases for all pulse widths. Particularly, when the pulse width is 50 ns, the decline in classification accuracy is the most significant. As the pulse width increases, the downward trend in classification accuracy gradually slows down. This indicates that in endurance test, the pulse width used for re-writing has a significant impact on accuracy degradation. Specifically, small pulse width has low latency, but the degradation in classification accuracy is significant and its endurance is poor; large pulse width has high latency, but the degradation in classification accuracy is less pronounced and its endurance is good, especially when the pulse number is $\leq 10^7$.

Fig. S18 | Endurance test of the memristor device. a, Analog switching of the full dynamic range of different operation numbers (pulse width is 200 ns). **b,** Achievable conductance range for different pulses with width of 50 ns, 200 ns and 500 ns. The maximum and minimum conductance of each cycle are collected to obtain the conductance range. We collect the analog switching of 25 devices for each pulse width configuration and display the boxplot of the conductance range, which shows that the achievable conductance range is degraded as the pulse number increases. **c,** On/off ratio of the 25 devices after different numbers of update pulses, the on/off ratio can still be larger than 1 after 10^9 incremental switching.

Fig. S19 | The relation of latency and accuracy in the endurance test in audio classification task. a. Changes in initial resistance at different pulse widths. The values are average high conductance states of 25 test devices for each pulse configuration. **b.** The change rate of β at different pulse widths. **c.** The decay process of β from the pre-trained network at different pulse widths. **d.** The effect of β decay on classification accuracy in audio classification at different pulse widths.”

Minor issues:

Comment 5. (line 36-37) *ambiguous sentence, rephrase.*

Answer: Thanks for your comments. The sentence “However, current feature learning techniques are very expensive in terms of computation, memory usage, and power consumption, restricting their applications in some scenarios, such as edge-computing where energy and hardware resource are limited” is now rephrased to be “However, current feature learning techniques have very high demands for computing power and data storage, resulting in significant energy consumption. In some scenarios, such as edge-computing, these demands can hardly be satisfied.”

The corresponding text in the manuscript has been revised as follows: “However, current feature learning techniques have very high demands for computing power and data storage, resulting in significant energy consumption. In some scenarios, such as edge-computing, these demands can hardly be satisfied.”

Comment 6. (line 45) *“still facing issues with hardware and energy costs”*: please argument this statement further and add possibly add citations.

Answer: Thanks. The convergence of simulated annealing may be slow due to the compute-intensive processes inside a highly interconnected interaction network and stochastic search algorithms that necessitate random number generation with an exponentially decaying probability distribution¹⁹. The Hamiltonian neural network computes frequently the partial differential equations (PDEs) of the Hamiltonian function with respect to momentum vectors. The solving of PDEs often requires the use of expensive numerical solvers²⁰. Therefore, the implementation of these physics-inspired techniques is usually not hardware-friendly enough.

The corresponding text in the manuscript has been revised as follows: “While these techniques are largely “inspired by physics”, they are not “compute-with-physics” or physics-based, still facing issues with hardware and energy costs. For example, the simulated annealing usually involves compute-intensive processes inside a highly interconnected interaction network and

stochastic search algorithms that necessitate random number generation with an exponentially decaying probability distribution¹⁹. While the Hamiltonian neural network computes frequently the partial differential equations (PDEs) of the Hamiltonian function with respect to momentum vectors. The solving of PDEs often requires the use of expensive numerical solvers²⁰”

Comment 7. (line 46) *Why "physics-based feature learning method that is computationally efficient remains elusive"? Please list a few examples where the computational efficiency of emerging techniques based on in-memory computing such as DPE is not sufficient and needs to be changed with new approaches such as the one presented here.*

Answer: Thanks for your comments. Although in-memory computing based on emerging techniques, e.g., memristors, has the potential to provide 100 to 100,000 times better computing efficiency than GPUs²¹, most materials and device efforts have been focusing on tailoring device properties to fit existing computing algorithms, which readily trivializes or excludes some intriguing new device properties²². This primarily leads to two issues:

Firstly, such techniques only accelerate the implementation of the artificial intelligence models on hardware but not improve the models. For deep neural networks, the MVM (matrix-vector multiplication) operations are accelerated by in-memory computing, but the number of MVM operations cannot be reduced. This will be problematic in the future because the computational cost and size of modern large language models (LLMs) increases dramatically. The increase number of computation operations may cancel out the benefits from in-memory computing. Furthermore, emerging techniques based on in-memory computing still have limited capacity, the capacity of reported in-memory computing chips is usually on the order of Mb²³. Therefore, a cluster for distributed computing with many in-memory computing chips is needed to implement the computation task of a single LLM. This may require complex system design, high communication and memory cost, leading to huge energy consumption and latency compared with the ideal situation where the entire model is run on a single chip²⁴.

Secondly, the computational efficiency of emerging techniques based on in-memory computing is hindered by the mismatched algorithms. For example, S. Kumar et al. proposed an activity-difference training strategy that exploits the analogue behavior of devices and offers over four orders of magnitude energy advantage compared with digital approaches²⁵.

Therefore, hardware acceleration and model simplification are both required to improve the computational efficiency of emerging techniques based on in-memory computing.

The corresponding text in the manuscript has been revised as follows: “Despite the potential of emerging in-memory computing approaches to enhance computational efficiency significantly, they remain constrained by limited memory capacity²³ and incompatible algorithms²⁵. A physics-based feature learning method that is computationally efficient remains elusive.”

Comment 8. (line 48) “recently” sounds inappropriate since ReRAM research started more than 15 years ago.

Answer: Thanks. The word “recently” is inappropriate and now deleted in the revised version.

The corresponding text in the manuscript has been revised as follows: “Memristor is widely recognized as a promising “compute-with-physics” device that directly implements matrix-vector multiplication (MVM) using physical laws, namely Ohm’s law for multiplication and Kirchhoff’s law for summation”

Comment 9. (line 65) “DDK” reintroduce acronym in the body of the article.

Answer: Thanks. The acronym of “DDK” is reintroduced in the revised version.

The corresponding text in the manuscript has been revised as follows: “Here, we introduce drift-diffusion kinetics (DDK) model, which leverages drift-diffusion kinetics in resistive switching (RS) to enable feature learning.”

Comment 10. *(line 75) reducing the number of parameters of a model while keeping the same accuracy is very strong statement, it should be supported by the specific model for which this happens, the dataset over which it is trained, the method it is compared against to, and an intuitive reason why the parameter reduction happens (just for the first layer, or for all layers?).*

Answer: Thanks for your comments. The details about the comparison are as follows: for recognition task of 10 speakers in Speakers in the Wild (SITW²⁶) dataset, the DDK network shows significantly better average accuracy (93.5%) than CNN (sample-level CNN²⁷, 88.1%). Moreover, the DDK network exhibits a reduction in the number of parameters and computational operations by approximately 296 and 6972 times, respectively, in comparison to the CNN. The reduction of parameters and operations across all layers may be attributed to the low number of adjustable parameters and the low complexity of the DDK layer, as well as the simplified classification layer resulting from the uncomplicated DDK layer.

The corresponding text in the manuscript has been revised as follows: “Compared with state-of-the-art feature extraction/learning algorithms, the DDK neural network exhibits exceptional performance on various tasks, with the number of parameters and computation operations (evaluated by multiply–accumulate operations (MACs)) greatly reduced. For example, in recognition task of 10 speakers in Speakers in the Wild (SITW²⁶) dataset, the DDK network shows significantly better average accuracy (93.5%) than CNN (sample-level CNN²⁷, 88.1%). Moreover, the DDK network exhibits a reduction in the number of parameters and computational operations by approximately 296 and 6972 times, respectively, in comparison to the CNN. The reduction of parameters and operations across all layers may be attributed to the low number of adjustable parameters and the low complexity of the DDK layer, as well as the simplified classification layer resulting from the uncomplicated DDK layer.”

Comment 11. (line 107-114) *the purpose of this paragraph and why it is relevant to the rest of the discussion is unclear.*

Answer: Thanks for your comments. The paragraph that discusses the psychology research is indeed somewhat irrelevant to the rest of discussion. Thus, the paragraph is rewritten with the irrelevant contents removed.

The corresponding text in the manuscript has been revised as follows: “Here we show that equation (4) can be used to extract feature from temporal pattern. Two temporal patterns, namely unambiguous (Fig. 1d) and ambiguous (Fig. 1e), are presented as examples. Here “1” stands for the presence of a sound, while “0” presents a silence. For unambiguous pattern, the responses at the start input are more pronounced (Fig. 1d). On the contrary, the responses of ambiguous pattern at the first, fourth and sixth inputs are almost equal (Fig. 1e).”

Comment 12. (line 151) *please specify if the error bars in S5 are the 95% intervals.*

Answer: Thanks. The error bars are specified for Fig. S5. They are the 95% intervals as in Fig. 2b-d.

The corresponding text in the manuscript has been revised as follows: “Device-to-device tests (100 devices, Fig. S8) draw identical conclusions. The error bars in Fig. S8 denote a 95% confidence interval.”

Comment 13. (line 152) *please provide literature references for the characterization of this specific device, or extend the supplementary material with information regarding: the writing procedure, the achievable conductance range, the on/off ratio, and the endurance of the device (and how these are tied together in a trade-off of latency vs accuracy vs endurance).*

Answer: Thank you. The writing procedure for write-and-verify has been already demonstrated in the revised manuscript (Fig. S12). The writing procedure for conductance range and on/off ratio

measurement without write-and-verify is shown in Fig. 8a. For the long-term potentiation (LTP) process, the voltage of BL for set operation is fixed (1.6 V), while the voltage of WL increases gradually (from 1.0 V to 2.0 V) and SL is ground. For the long-term depression (LTD) process, the voltage of SL for reset operation is fixed (-2 V), while the voltage of WL increases gradually (from 1.0 V to 2.0 V) and BL is ground. The pulse width in the writing operation is fixed at 100 ns. The read operation uses a fixed voltage (0.2 V) and pulse width (20 μ s). The above two writing procedures belong to the gate-voltage-modulation method, in which the conductance states are determined not only by the voltages applied on BL and SL, but primarily by the compliant current controlled by the specific gate voltage. We use this method for write-and-verify of the devices in the classification layer because it typically can achieve linear and symmetric conductance change^{28,29}. Fig. 8b shows the measured conductance range of the memristor device using the above writing procedure and configuration. The achievable conductance range is from 6.1 μ S to 202.1 μ S. Therefore, the on/off ratio at this writing configuration is about 33.13.

The endurance of the device is tested using the same method as in the DDK feature extraction but not the gate-voltage-modulation method. In this method, a large and constant gate voltage is applied so that the primary role of gate voltage is to select the specific device but not control the conductance states by limiting the compliant currents. Therefore, the conductance states are determined by the voltages applied on BL and SL. For the LTP process, the voltage of WL for set operation is fixed at 2 V, while the voltage of BL increases gradually (from 0.7 V to 1.7 V) and SL is ground. For the LTD process, the voltage of WL for reset operation is fixed at 3 V, while the voltage of SL increases gradually (from -1.4 V to -2.4 V) and BL is ground. The pulse width in the writing operation is fixed at 50 ns, 200 ns, 500 ns, respectively. The read operation uses a fixed voltage (0.2 V) and pulse width (20 μ s). Fig. 4 shows the endurance test results of the devices. Fig. 4a shows the LTD and LTP characteristics around initial, 10^3 , 10^6 , 10^7 , and 10^9 pulses. The conductance range and on/off ratio of 25 devices for each pulse width configuration in the endurance test are statistically calculated in Fig. 4b and Fig. 4c, respectively. Though the dynamic range and on/off ratio degrade as the increase of update pulses, there is an on/off ratio larger than 1 after 10^9 update pulses.

Fig. 8 | The pulsed I - V - t curves of the memristor. a, Programming scheme. For the LTP process, the voltage of the transistor gate (WL) increases linearly (from 0.9 V to 2.0 V), where the voltage of the top electrode is fixed at 2.2 V and the voltage of the source is ground. For the LTD process, each operation includes a SET and RESET pulse. For each operation, the RESET pulse is applied first, then the SET pulse is applied. The RESET voltages of the gate, source, and top electrode are fixed (3.7 V, ground, and -3.3 V separately). The SET voltage of the gate decreases linearly (from 2.0 V to 0.9 V). The SET voltage of the top electrode is fixed at 2.2V, while the source is grounded. The pulse width in the measurement is fixed at 5 ns. The conductance is read at 0.2 V. **b**, A typical LTP/LTD curve of the memristor device.

Fig. 4 | Endurance test of the memristor device. a, Analog switching of the full dynamic range of different operation numbers (pulse width is 200 ns). **b,** Achievable conductance range for different pulses with width of 50 ns, 200 ns and 500 ns. The maximum and minimum conductance of each cycle are collected to obtain the conductance range. We collect the analog switching of 25 devices for each pulse width configuration and display the boxplot of the conductance range, which shows that the achievable conductance range is degraded as the pulse number increases. **c,** On/off ratio of the 25 devices after different numbers of update pulses, the on/off ratio can still be larger than 1 after 10^9 incremental switching.

The corresponding text in the manuscript has been revised as follows: “

Supplementary information

Fig. S18 | Endurance test of the memristor device. a, Analog switching of the full dynamic range of different operation numbers (pulse width is 200 ns). **b**, Achievable conductance range for different pulses with width of 50 ns, 200 ns and 500 ns. The maximum and minimum conductance of each cycle are collected to obtain the conductance range. We collect the analog switching of 25 devices for each pulse width configuration and display the boxplot of the conductance range, which shows that the achievable conductance range is degraded as the pulse number increases. **c**, On/off ratio of the 25 devices after different numbers of update pulses, the on/off ratio can still be larger than 1 after 10^9 incremental switching.”

Comment 14. (line 184) Fig 3d, Table S2: MFCC is based on FFT + filters. Please provide comparisons also with methods that don't extract features with heavy pre-processing, if they exist. If they don't exist, provide the reason why preprocessing is required.

Answer: Thanks for your comments. The evolution of audio feature extraction can be sub-categorized into time domain, frequency domain, joint time-frequency domain³⁰. The oldest and simplest features are extracted from the time domain. The time domain features (e.g., zero-crossing rate) evolved up to late 1950s. In around 1950s, the frequency domain features (e.g., short time Fourier transform (STFT)) were proposed. The time domain signal is converted into frequency domain signal by using Fourier transform or auto-regression analysis. In later 1960s, the joint time-frequency feature extraction algorithms (e.g., Mel frequency cepstral coefficient (MFCC)) were developed. While early time-domain approaches are comparatively straightforward, they exhibit a notable deficiency in effectiveness relative to joint time-frequency domain methods³¹⁻³³.

The joint time-frequency domain feature extraction approaches can roughly be categorized into two types, one models the sound production and the other models the auditory system³⁴. (1) For the first type, linear prediction coding (LPC) is a representative method. LPC is based on the idea that the current sound sample can be approximated by a linear combination of past sound samples. The preprocessing steps such as pre-amplification, framing and windowing are required³⁵. Moreover, the predictor coefficients are computed by minimizing the sum of squared differences between the actual samples and the linearly predicted ones using least square method³⁶. However, since LPC are too sensitive to numerical precision, hence it is desirable to transform the LPC to the cepstral domain (linear prediction cepstral coefficients, LPCC) using a recursive method³⁷. (2) For the other type, the MFCC is the most representative. Prior literature has established that MFCC-based feature extraction methods are the most prevalent, effective, and extensively utilized techniques for speaker identification feature extraction, accounting for 97% of total implementations, with 31% comprising solely MFCC³⁴.

The preprocessing of MFCC contains pre-amplification, framing and windowing like the LPC. The pre-amplification is used to enhance high-frequency components, improve signal-to-noise

ratio, and reduce signal distortion. It may work like a high pass filter. Afterwards, a framing step is performed to divide the signals into frames. Because in most cases, speech signals are non-stationary, and Fourier transform of the entire signal is meaningless because the frequency profile of the signal is lost over time. Therefore, Fourier transform is performed on short-term frames and a good approximation of the signal frequency profile can be obtained by connecting adjacent frames. At last, a windowing operation is carried out. The discontinuity introduced by segmenting the signal into frames distorts the spectrum. The distortion can be reduced by multiplying the speech frame by a window function.

Therefore, although there are some feature extraction methods that don't extract features with heavy pre-processing, they are out-of-date with unsatisfactory performances in modern audio signal processing. As a dominant feature extraction method up-to-date, MFCC requires preprocessing steps to guarantee a high-quality spectrum for subsequent coefficient calculation.

The corresponding text in the manuscript has been revised as follows: “Here the Mel-frequency cepstral coefficient (MFCC) method is presented for comparison with the DDK method. The MFCC-based method is a dominant feature extraction method that is widely used in audio signal processing³⁴. It typically requires heavy preprocessing to guarantee a high-quality spectrum for subsequent coefficient calculation. In comparison with MFCC-based approaches (Fig. S9), hardware implementation of the proposed algorithm with memristors is very simple and shows great advantages in latency and energy consumption.”

Comment 15. (line 187-188) *The profiling of GPU, CPU, and ASIC needs to be further extended and presented in detail in the methods section. Also, why it is claimed an improvement of “at least 10^3 ”, instead of providing an exact comparison?*

Answer: Thanks for your comments. The profiling of GPU, CPU and ASIC is extended and presented in detail in the methods in the revised manuscript, as discussed in the response to comment 2. The vague claim is now replaced by an exact comparison. For example, “as shown in

Fig. 3d and Table S2, the latency and energy consumption of a single DDK operation of the TiN/TaO_x/HfO_x/TiN cell are estimated to be 10 ns and 0.8 pJ, using an average voltage of 2 V and an average resistance of 50 KΩ (detailed in Supplementary Notes), which are reduced by 917 (graphics processing unit, GPU¹¹) ~ 1.6×10⁶ (application specific integrated circuit, ASIC⁶) times and 6800 (ASIC⁶) ~ 1.725×10¹⁰ (central processing unit, CPU¹⁰) times in comparison with the MFCC operation, respectively.”

The corresponding text in the manuscript has been revised as follows: “As shown in Fig. 3d and Table S2, the latency and energy consumption of a single DDK operation of the TiN/TaO_x/HfO_x/TiN cell are estimated to be 10 ns and 0.8 pJ, using an average voltage of 2 V and an average resistance of 50 KΩ (detailed in Supplementary Notes), which are reduced by 917 (graphics processing unit, GPU¹¹) ~ 1.6×10⁶ (application specific integrated circuit, ASIC⁶) times and 6800 (ASIC⁶) ~ 1.725×10¹⁰ (central processing unit, CPU¹⁰) times in comparison with the MFCC operation, respectively.”

Comment 16. (line 192) please provide literature reference for imprecision-based NN.

Answer: Thanks. The imprecision-based neural network training framework is developed by ourself based on PyTorch. In the design of the framework, we refer to the following papers for the modeling of imprecision factors and they are now added in the revised manuscript:

Le Gallo, M. et al. A 64-core mixed-signal in-memory compute chip based on phase-change memory for deep neural network inference. Nature Electronics 6, 680-693 (2023).
<https://doi.org:10.1038/s41928-023-01010-1>

This reference characterizes the MVM error to quantify the MVM precision. Three metrics are used to describe the MVM error:

$$\epsilon_{\text{total}} = y_{\text{exp}} - y_{\text{fp}} \quad (6)$$

$$\epsilon_{\text{linear}} = \widehat{W}x - y_{\text{fp}} \quad (7)$$

$$\epsilon_{\text{residual}} = y_{\text{exp}} - \widehat{W}x \quad (8)$$

Where $y_{fp}=Wx$ denotes the desired MVM result, \widehat{W} is the programmed weight, and y_{exp} is the experimental output. ϵ_{total} , including ϵ_{linear} and $\epsilon_{residual}$, is the total MVM error. ϵ_{linear} defines the error resulting from a wrong programming of the weights, and $\epsilon_{residual}$ results from residual error, such as read noise, leakage current, and IR drop etc.

In our work, we considered three main imprecision factors, i.e., available conductance range, output variation, and device failure. Like the reference, we consider the ϵ_{linear} using the writing error and $\epsilon_{residual}$ using the output variation. Moreover, we train the neural network with clipped weight to make the most conductance achievable to adapt the memristor array. Device failures such as being stuck on or off are considered to achieve better learning performance.

The corresponding text in the manuscript has been revised as follows: “To make the network training process be aware of hardware non-idealities, the imprecision-based neural network training framework incorporates limited device conductance range, output noise and device failure in network training (detailed in Supplementary Notes)³⁸”

Comment 17. (line 216) *Please discuss also the trade-offs involved with the device variability, such as: What is the precision of the readout? What is the time it takes to extract these features? If this is part of the feedforward mechanism, this can become a relevant contribution to throughput. Also, which peripherals are used around the memory array? How much energy they need per conversion? How many levels the readout needs?*

Answer: Thank you. To clarify your question, here we provide a schematic of the hardware system for testing, as shown in Fig. 9. This figure was published in our previous paper³⁹. The peripherals around the array includes digital-analog converter (DAC), transimpedance amplifier (TIA), and analog-digital converter (ADC). The DACs convert the digital input voltage signal from the FPGA to the analog input signal (voltage pulse) on the crossbar array. TIAs are used to convert the output current of the crossbar array to voltage and then feed the voltage to the ADCs. Then the ADCs

convert the analog output voltage of the TIA to digital output voltage signal. The peripherals and the memristor arrays are part of the feedforward mechanism.

For the readout, we studied the influence of readout precision on the classification accuracy of the DDK network in a task of classifying 10 speakers in the SITW dataset. This simulation study was presented in the supplementary materials (Fig. S17 in the revised manuscript). Here we present the Fig. S17 as Fig. 10 in the response letter to explain the influence of readout precision. Firstly, we train a baseline DDK network using floating point 32-bit (FP 32) data format. The test accuracy at FP 32 is 94.8%. When deploying the trained model on hardware, it is usually quantized since the resource and precision on an edge hardware is limited. We study the influence of different quantization methods (post-training quantization (PTQ) and quantization-aware training (QAT)) on the DDK network accuracy. The quantization precision varies from INT2 (2-bit integer) to INT 8. We discuss two situations, namely with activation function quantization and without activation function quantization. In the fully connected layer of the DDK network, a common linear quantization method is employed to quantize it into integers ranging from 2bit to 8bit. For the quantization of the Sigmoid function, the first step is to determine the valid range of the inputs. Considering the characteristics of the Sigmoid function, its valid input range is typically set to $[-8, 8]$, as within this interval, the output of the Sigmoid function is already close to its theoretical limit values (i.e., approaching 0 or 1). Subsequently, the clipped inputs are quantized into integers ranging from 2bit to 8bit through linear mapping. Essentially, this approach substitutes the Sigmoid function with the operations of clipping and linear mapping. The simulation is performed for 5 times for each training configuration and the maximal accuracy is used for evaluation. As shown in Fig. 10, the overall performance of QAT is better than PTQ. And networks using quantization methods with activation function quantization show slightly lower accuracy than those without activation function quantization. In terms of quantization precision, there are slight accuracy loss ($\leq 4.8\%$, with activation function quantization) for the PTQ if the precision is ≥ 6 bit. While for the QAT, even the accuracy loss of 4-bit quantization (3.6%, with and without activation function quantization) is still acceptable. Therefore, a 4-bit or higher weight precision is acceptable for the DDK network in this example, showing that our method does not necessarily need a high-precision ADC or DAC. In the hardware benchmark section in the supplementary information, we

evaluate the hardware performance using 8-bit ADC and DAC, which may be a reasonable selection for most situations based on this discussion. Thus, the readout precision can be 8-bit (256 readout levels) or lower. It should be noted that since the aim of our prototype hardware system is to systematically investigate the specifications of memristor devices and chips, for example, the maximal conductance levels that a single memristor cell can achieve, we use high-precision ADCs (16-bit) and DACs (16-bit) in the prototype system to study the potentials of the memristors. In practical implementation of memristor hardware, for example, a memristor chip, such high-precision DACs and ADCs are not necessary.

The time and energy per conversion can be estimated according to the following equation:

$$T_{\text{conversion}} = T_{\text{DAC}} + T_{\text{array}} + T_{\text{TIA}} + T_{\text{ADC}}, \quad (9)$$

$$E_{\text{conversion}} = E_{\text{DAC}} + E_{\text{array}} + E_{\text{TIA}} + E_{\text{ADC}}. \quad (10)$$

For the evaluation of the peripheral circuits, we have revised our selection of the reference circuits in the original manuscript and chosen three papers that present more suitable hardware benchmarks. The chips discussed in these papers were tested after tape-out. The corresponding literature references for the circuits, namely, TIA⁴⁰, ADC⁴¹, and DAC⁴², used in these benchmarks can now be found in the following, respectively:

Lavasani, H. M., Pan, W., Harrington, B., Abdolvand, R. & Ayazi, F. A 76dB Ω 1.7GHz 0.18 μ m CMOS tunable transimpedance amplifier using broadband current pre-amplifier for high frequency lateral micromechanical oscillators. In *2010 IEEE International Solid-State Circuits Conference - (ISSCC)*. 318-319.

Brooks, L. & Lee, H.-S. A Zero-Crossing-Based 8-bit 200 MS/s Pipelined ADC. *IEEE Journal of Solid-State Circuits* **42**, 2677-2687 (2007).

Idros, N., Rosli, A., Abdul Aziz, Z. A., Rajendran, J. & Marzuki, A. A 1.8 V high-speed 8-bit hybrid DAC with integrated rail-to-rail buffer amplifier in CMOS 180 nm. *Microelectronics International* **38**, 46-54 (2021).

The T_{DAC} , T_{TIA} , and T_{ADC} in the references are 5 ns, 0.4 ns and 20 ns respectively. The power of DAC, TIA and DAC in the references are 12.6 mW, 7.2 mW and 8.5 mW respectively.

The maximum energy cost of a single DAC, ADC, and TIA, could be estimated as:

$$E_{DAC} = 12.6 \text{ mW} \times 5 \text{ ns} = 63 \text{ pJ} \text{ (180 nm, 8-bit, 200 MHz)} \quad (11)$$

$$E_{TIA} = 7.2 \text{ mW} \times 0.4 \text{ ns} = 2.88 \text{ pJ} \text{ (180 nm, 2.5 GHz)} \quad (12)$$

$$E_{ADC} = 8.5 \text{ mW} \times 20 \text{ ns} = 170 \text{ pJ} \text{ (180 nm, 8-bit, 200 MHz)} \quad (13)$$

Thus, the $T_{\text{conversion}}$ could be estimated as

$$T_{\text{conversion}} = 5 \text{ ns} + 10 \text{ ns} + 0.4 \text{ ns} + 20 \text{ ns} = 35.4 \text{ ns} \quad (14)$$

The $E_{\text{conversion}}$ could be estimated as

$$E_{\text{conversion}} = 63 \text{ pJ} + 10 \text{ ns} \times (2 \text{ V})^2 \times 20 \text{ } \mu\text{S} + 2.88 \text{ pJ} + 170 \text{ pJ} = 236.68 \text{ pJ} \quad (15)$$

Fig. 9 | A schematic of the memristor hardware system. The system consisting of an upper computer and the PCB for memristor chip. The code of the DDK neural network is edited and debugged in an integrated development environment that provides the programming platform. After compilation, the DDK and matrix-vector multiplication (MVM) operations are executed by the memristor chip via a library-based toolchain that consists of operator libraries, runtime, APIs and drivers. While the nonlinear operations like activation functions are performed by the CPU.

The input and output signals on the memristor crossbar array are analog. Matrix switches are controlled by the FPGA to select desired array cells. We use DACs to convert the digital input voltage signal from the FPGA to the analog input signal (voltage pulse) on the crossbar array. For the output, a TIA is used to convert the output current of the crossbar array to voltage and then feed the voltage to the ADCs. Then the ADCs convert the analog output voltage of the TIA to digital output voltage signal.

Fig. 10 | Quantization performances of the DDK network on SITW dataset. The accuracy refers to the maximal accuracy in 5 simulations. The quantization methods are as follows. PTQ (sigmoid): post-training quantization with sigmoid activation function quantization; PTQ: post-training quantization without sigmoid activation function quantization; QAT (sigmoid): quantization-aware training with sigmoid activation function quantization; QAT: quantization-aware training without sigmoid activation function quantization. The overall accuracy loss of QAT is less than PTQ. For PTQ, sigmoid activation function quantization is not preferred, while for QAT, sigmoid activation function quantization can usually lead to better performance.

The corresponding text in the manuscript has been revised as follows: “Quantization performance of the DDK network is evaluated on SITW dataset by using post-training quantization (PTQ) and quantization-aware training (QAT), as shown in Fig. S17, proving that quantization is feasible. In terms of quantization precision, there are slight accuracy loss ($\leq 4.8\%$, with activation function quantization) for the PTQ if the precision is ≥ 6 bit. While for the QAT, even the accuracy loss of 4-bit quantization (3.6%, with and without activation function quantization) is still acceptable. Therefore, a 4-bit or higher weight precision is acceptable for the DDK network in this example, showing that the DDK hardware does not necessarily need a high-precision analog-to-digital convertor (ADC) or digital-to-analog convertor (DAC).

In the hardware benchmark, we evaluate the hardware performance using 8-bit ADC and DAC, which may be a reasonable selection for most situations. The energy-efficiency of the DDK hardware is evaluated in the speaker recognition task considering four layout strategies (Fig. S20). Based on the hardware parameters in Table S9, the latency, area and energy consumption of the circuits can be calculated (Table S10). The energy-efficiency is evaluated to reach $2.77 \text{ TOP s}^{-1} \text{ W}^{-1}$ at integer 8-bit (INT8) at 180 nm technology node (hardware benchmark is detailed in Supplementary Notes and the result is listed in Table S11), outperforming a Tesla V100 GPU by approximately 27.6 times⁴³. The time and energy of a single feature extraction of the DDK hardware is estimated to be 35.4 ns and 236.68 pJ, respectively.”

Comment 18. (line 520) *this is in the reviewer’s opinion the most important subsection; it needs to be rewritten, with added details and clarity.*

Answer: Thanks for your comments. This subsection has been rewritten as follows: DDK network rely on the backpropagation algorithm to execute the process of parameter updates and optimization during training. The training process is as follows. Firstly, the preprocessed input data is propagated forward through each layer of the network to obtain the predicted output. Then, the loss between the prediction and the actual target is calculated. Next, the gradients of the loss function with respect to α and β in the DDK layer, as well as each weight in the classification layer, are computed using the chain rule. Finally, the computed gradients are used to update α and β in

the DDK layer and the weights in the classification layer, aiming to minimize the loss function. The process ensures that the DDK network, like other neural networks, can effectively learn and adjust its parameters through the backpropagation algorithm. The specific methods for preprocessing the data, forward propagation, backward propagation, and parameter updating are presented in the following.

1. Input data pre-processing

Each utterance or vector from the raw dataset is normalized to [0, 1].

2. Forward pass

1) DDK layer

$$z^1(0) = 1 \tag{16}$$

for $t = 1$ to T , do

$$z^1(t) = z^1(t - 1) + \frac{\alpha}{z^1(t-1)} - \beta x(t - 1) \tag{17}$$

$$a^1 = z^1 \tag{18}$$

end for.

T is the total length of utterance or vector. $z^k(t)$ is the output of the k th layer (the first layer is the DDK layer) at input index t . a^1 is the neuron output of the DDK layer.

2) Hidden layers:

for $k=2$ to $L-1$ do

$$z^k = W^k a^{k-1} \tag{19}$$

$$a^k = \sigma(z^k) \tag{20}$$

end for.

3) Output layer:

$$z^L = W^L a^{L-1} \quad (21)$$

$$a^L = \text{SoftMax}(z^L) \quad (22)$$

L is the total number of layers in the network. σ is an activation function. W^k is the weight matrix of the k th layer.

3. Backward pass

1) Output layer

a. Calculate the gradient of W in output layer:

$$\frac{\partial C(\alpha, \beta, W)}{\partial W} = (a^L - y) a^{L-1} \quad (23)$$

b. Update W :

$$\Delta W = -\eta \cdot \frac{\partial C(\alpha, \beta, W)}{\partial W} \quad (24)$$

y is truth label, η is learning rate, $C(\alpha, \beta, W)$ is loss function.

2) Hidden layer

a. Calculate the error δ^k and gradient of W in the k th layer:

$$\delta^k = W^{k+1} \delta^{k+1} \odot \sigma'(z^k) \quad (25)$$

$$\frac{\partial C(\alpha, \beta, W)}{\partial W} = \delta^k a^{k-1} \quad (26)$$

b. Update W :

$$\Delta W = -\eta \cdot \frac{\partial C(\alpha, \beta, W)}{\partial W} \quad (27)$$

3) DDK layer

a. Calculate the gradient of α and β :

$$\begin{cases} \frac{\partial z^1(t)}{\partial \alpha} = \frac{1}{z^1(t-1)}, & t = 1 \\ \frac{\partial z^1(t)}{\partial \alpha} = \frac{1}{z^1(t-1)} + \prod_{t=2}^T \frac{\partial z^1(t)}{\partial z^1(t-1)} \cdot \frac{\partial z^1(1)}{\partial \alpha}, & 2 \leq t \leq T \end{cases} \quad (28)$$

$$\begin{cases} \frac{\partial z^1(t)}{\partial \beta} = -x(t-1), & t = 1 \\ \frac{\partial z^1(t)}{\partial \beta} = -x(t-1) + \prod_{t=2}^T \frac{\partial z^1(t)}{\partial z^1(t-1)} \cdot \frac{\partial z^1(1)}{\partial \beta}, & 2 \leq t \leq T \end{cases} \quad (29)$$

$$\frac{\partial C(\alpha, \beta, W)}{\partial \alpha} = \delta^1 \sum_{t=1}^T \frac{\partial z^1(t)}{\partial \alpha} \quad (30)$$

$$\frac{\partial C(\alpha, \beta, W)}{\partial \beta} = \delta^1 \sum_{t=1}^T \frac{\partial z^1(t)}{\partial \beta} \quad (31)$$

b. Update α and β :

$$\Delta \alpha = -\eta \cdot \frac{\partial C(\alpha, \beta, W)}{\partial \alpha} \quad (32)$$

$$\Delta \beta = -\eta \cdot \frac{\partial C(\alpha, \beta, W)}{\partial \beta} \quad (33)$$

This process ensures that the DDK network, like other neural networks, can effectively learn and adjust its parameters through the backpropagation algorithm.

The corresponding text in the manuscript has been revised as follows: “DDK networks rely on the backpropagation algorithm to execute the process of parameter updates and optimization during training. The training process is presented in the following. Firstly, the preprocessed input data is propagated forward through each layer of the network to obtain the predicted output. Then, the loss between the prediction and the actual target is calculated. Next, the gradients of the loss function with respect to α and β in the DDK layer, as well as each weight in the classification layer,

are computed using the chain rule. Finally, the computed gradients are used to update α and β in the DDK layer and the weights in the classification layer, aiming to minimize the loss function. The specific methods for preprocessing the data, forward propagation, backward propagation, and parameter updating are as follows:

1. Input data pre-processing

Each utterance or vector from the raw dataset is normalized to $[0, 1]$.

2. Forward pass

1) DDK layer

$$z^1(0) = 1 \tag{12}$$

for $t = 1$ to T , do

$$z^1(t) = z^1(t - 1) + \frac{\alpha}{z^1(t-1)} - \beta x(t - 1) \tag{13}$$

$$a^1 = z^1 \tag{14}$$

end for.

T is the total length of utterance or vector. $z^k(t)$ is the output of the k th layer (the first layer is the DDK layer) at input index t . a^1 is the neuron output of the DDK layer.

2) Hidden layers:

for $k=2$ to $L-1$ do

$$z^k = W^k a^{k-1} \tag{15}$$

$$a^k = \sigma(z^k) \tag{16}$$

end for.

3) Output layer:

$$z^L = W^L a^{L-1} \quad (17)$$

$$a^L = \text{SoftMax}(z^L) \quad (18)$$

L is the total number of layers in the network. σ is an activation function. W^k is the weight matrix of the k th layer.

3. Backward pass

4) Output layer

a. Calculate the gradient of W in output layer:

$$\frac{\partial C(\alpha, \beta, W)}{\partial W} = (a^L - y) a^{L-1} \quad (19)$$

b. Update W :

$$\Delta W = -\eta \cdot \frac{\partial C(\alpha, \beta, W)}{\partial W} \quad (20)$$

y is truth label, η is learning rate, $C(\alpha, \beta, W)$ is loss function.

5) Hidden layer

a. Calculate the error δ^k and gradient of W in the k th layer:

$$\delta^k = W^{k+1} \delta^{k+1} \odot \sigma'(z^k) \quad (21)$$

$$\frac{\partial C(\alpha, \beta, W)}{\partial W} = \delta^k a^{k-1} \quad (22)$$

b. Update W :

$$\Delta W = -\eta \cdot \frac{\partial C(\alpha, \beta, W)}{\partial W} \quad (23)$$

6) DDK layer

c. Calculate the gradient of α and β :

$$\begin{cases} \frac{\partial z^1(t)}{\partial \alpha} = \frac{1}{z^1(t-1)}, & t = 1 \\ \frac{\partial z^1(t)}{\partial \alpha} = \frac{1}{z^1(t-1)} + \prod_{t=2}^T \frac{\partial z^1(t)}{\partial z^1(t-1)} \cdot \frac{\partial z^1(1)}{\partial \alpha}, & 2 \leq t \leq T \end{cases} \quad (24)$$

$$\begin{cases} \frac{\partial z^1(t)}{\partial \beta} = -x(t-1), & t = 1 \\ \frac{\partial z^1(t)}{\partial \beta} = -x(t-1) + \prod_{t=2}^T \frac{\partial z^1(t)}{\partial z^1(t-1)} \cdot \frac{\partial z^1(1)}{\partial \beta}, & 2 \leq t \leq T \end{cases} \quad (25)$$

$$\frac{\partial C(\alpha, \beta, W)}{\partial \alpha} = \delta^1 \sum_{t=1}^T \frac{\partial z^1(t)}{\partial \alpha} \quad (26)$$

$$\frac{\partial C(\alpha, \beta, W)}{\partial \beta} = \delta^1 \sum_{t=1}^T \frac{\partial z^1(t)}{\partial \beta} \quad (27)$$

d. Update α and β :

$$\Delta \alpha = -\eta \cdot \frac{\partial C(\alpha, \beta, W)}{\partial \alpha} \quad (28)$$

$$\Delta \beta = -\eta \cdot \frac{\partial C(\alpha, \beta, W)}{\partial \beta} \quad (29)$$

This process ensures that the DDK network, like other neural networks, can effectively learn and adjust its parameters through the backpropagation algorithm.”

Comment 19. (line 532) please provide an intuitive explanation of why transforming the data using the differential equation that governs the ReRAM is more efficient than competing approaches.

Answer: Thanks for your comments. In our approach, the transforming is performed by the differential equation that is defined by two parameters, namely alpha and beta. While in the

competing approaches, such as attention mechanism, convolution and recurrent connections, the transformation is defined by a set of weight matrices. Consequently, our approach can achieve minimal computational complexity, as evidenced by a limited number of trainable parameters and computational operations.

The corresponding text in the manuscript has been revised as follows: “In our approach, the data transformation is performed by the differential equation that is defined by two parameters α and β . While in the competing approaches, such as attention mechanism, convolution and recurrent connections, this transformation is defined by a set of weight matrices. Consequently, our approach can achieve minimal computational complexity, as evidenced by a limited number of trainable parameters and computational operations.”

Comment 20. (line 293-294 of supplementary material) *These numbers are very ambitious for a 180nm technology node, please provide literature reference.*

Answer: Thanks for your professional suggestion. We have revised our selection of the reference circuits in the original manuscript and chosen three papers that present more suitable hardware benchmarks. The chips discussed in these papers were tested after tape-out. The corresponding literature references for the circuits, namely, TIA⁴⁰, ADC⁴¹, and DAC⁴², used in these benchmarks can now be found in the following, respectively.

Lavasani, H. M., Pan, W., Harrington, B., Abdolvand, R. & Ayazi, F. in *2010 IEEE International Solid-State Circuits Conference - (ISSCC)*. 318-319.

Brooks, L. & Lee, H.-S. A Zero-Crossing-Based 8-bit 200 MS/s Pipelined ADC. *IEEE Journal of Solid-State Circuits* **42**, 2677-2687 (2007). <https://doi.org/10.1109/jssc.2007.908770>

Idros, N., Rosli, A., Abdul Aziz, Z. A., Rajendran, J. & Marzuki, A. A 1.8 V high-speed 8-bit hybrid DAC with integrated rail-to-rail buffer amplifier in CMOS 180 nm. *Microelectronics International* **38**, 46-54 (2021). <https://doi.org/10.1108/mi-10-2020-0073>

The corresponding text in the manuscript has been revised as follows: “

$$E_{DAC} = 12.6 \text{ mW} \times 5 \text{ ns} = 63 \text{ pJ} \text{ (180 nm, 8-bit, 200 MHz)} \quad (\text{S14})$$

$$E_{TIA} = 7.2 \text{ mW} \times 0.4 \text{ ns} = 2.88 \text{ pJ} \text{ (180 nm, 2.5 GHz)} \quad (\text{S15})$$

$$E_{ADC} = 8.5 \text{ mW} \times 20 \text{ ns} = 170 \text{ pJ} \text{ (180 nm, 8-bit, 200 MHz)} \quad (\text{S16})$$

$$E_{total} = n_{DDK_operation} \times E_{DDK_operation} + n_{cell} \times E_{cell} + n_{DAC} \times E_{DAC} + n_{TIA} \times E_{TIA} + n_{ADC} \times$$

$$E_{ADC} = 128 \times 0.8 \text{ pJ} + 2234 \times 2 \times 0.04 \text{ pJ} + 128 \times 63 \text{ pJ} + 154 \times 2.88 \text{ pJ} + 3 \times 170 \text{ pJ} =$$

$$9206.48 \text{ pJ} \quad (\text{S17})$$

$$Area_{total} = n_{DAC} \times DAC_{area} + n_{ADC} \times ADC_{area} + n_{TIA} \times TIA_{area} + (n_{DDK_cell} + n_{cell}) \times Cell_{area} =$$

$$128 \times 0.068 \text{ mm}^2 + 3 \times 0.05 \text{ mm}^2 + 154 \times 0.33 \text{ mm}^2 + (1 + 2234 \times 2) \times 6.25 \times 10^{-4} \text{ mm}^2 =$$

$$62.467125 \text{ mm}^2 \quad (\text{S18})$$

$$E_{total} = n_{cell} \times E_{cell} + n_{DAC} \times E_{DAC} + n_{TIA} \times E_{TIA} + n_{ADC} \times E_{ADC} = 662656 \times 2 \times 0.04 \text{ pJ} +$$

$$1600 \times 63 \text{ pJ} + 210698 \times 2.88 \text{ pJ} + 3 \times 170 \text{ pJ} = 761132.72 \text{ pJ} \quad (\text{S19})$$

$$Area_{total} = n_{DAC} \times DAC_{area} + n_{ADC} \times ADC_{area} + n_{TIA} \times TIA_{area} + n_{cell} \times Cell_{area} =$$

$$1600 \times 0.068 \text{ mm}^2 + 3 \times 0.05 \text{ mm}^2 + 210698 \times 0.33 \text{ mm}^2 + 662656 \times 2 \times 6.25 \times 10^{-4} \text{ mm}^2 =$$

$$70467.61 \text{ mm}^2 \quad (\text{S20})$$

$$E_{total} = n_{DDK_operation} \times E_{DDK_operation} + n_{cell} \times E_{cell} + n_{DAC} \times E_{DAC} + n_{TIA} \times E_{TIA} + n_{ADC} \times$$

$$E_{ADC} = 4800 \times 0.8 \text{ pJ} + 649600 \times 2 \times 0.04 \text{ pJ} + 4800 \times 63 \text{ pJ} + 5578 \times 2.88 \text{ pJ} + 3 \times 170 \text{ pJ} =$$

$$374782.64 \text{ pJ} \quad (\text{S21})$$

$$Area_{total} = n_{DAC} \times DAC_{area} + n_{ADC} \times ADC_{area} + n_{TIA} \times TIA_{area} + (n_{DDK_cell} + n_{cell}) \times Cell_{area} =$$

$$4800 \times 0.068 \text{ mm}^2 + 3 \times 0.05 \text{ mm}^2 + 5578 \times 0.33 \text{ mm}^2 + (80 + 649600 \times 2) \times 6.25 \times 10^{-4} \text{ mm}^2 =$$

$$2979.34 \text{ mm}^2 \quad (\text{S22})$$

$$E_{total} = n_{cell} \times E_{cell} + n_{DAC} \times E_{DAC} + n_{TIA} \times E_{TIA} + n_{ADC} \times E_{ADC} = 78025792 \times 2 \times 0.04 \text{ pJ} +$$

$$602112 \times 63 \text{ pJ} + 23390218 \times 2.88 \text{ pJ} + 3 \times 170 \text{ pJ} = 111539457.2 \text{ pJ} \quad (\text{S23})$$

$$\begin{aligned}
 Area_{total} &= n_{DAC} \times DAC_{area} + n_{ADC} \times ADC_{area} + n_{TIA} \times TIA_{area} + n_{cell} \times Cell_{area} = \\
 &602112 \times 0.068 \text{mm}^2 + 3 \times 0.05 \text{mm}^2 + 23390218 \times 0.33 \text{mm}^2 + 78025792 \times 2 \times 6.25 \times \\
 &10^{-4} \text{mm}^2 = 7857247.946 \text{mm}^2 \quad (S24)
 \end{aligned}$$

The area of a single DAC, ADC, and TIA is 0.068 mm², 0.05 mm² and 0.34 mm², respectively.

In general, among the four strategies, Strategy A is the most area-saving strategy (3.23 mm²), it occupies seven times less area than Strategy D (22.68 mm²).

After comparing these four strategies, it is indicated that strategy A is the most area-saving strategy, strategies B and D achieve high energy-efficiency (2.77 TOP s⁻¹W⁻¹), and strategy C has a large performance density (210.28 GOP s⁻¹ mm⁻²).

The 10 output neurons at the output layer could have their TIA signals converted to digits with

$$n_{ADC} = \text{ceil} \left(\frac{n_{output \ dim}}{20 \ ns \times 0.2 \ GHz} \right) = 3.$$

Table S9 | Hardware parameters at 180 nm technology node in simulation

Circuit	TIA ⁴⁰	ADC ⁴¹	DAC ⁴²
Technology	180 nm	180 nm	180 nm
Resolution (bit)	8	8	8
Power (mW)	7.2	8.5	12.6
Frequency (MHz)	2500	200	200
Area (mm ²)	0.33	0.05	0.068
Voltage (V)	1.8	1.8	1.8
Current (mA)	4	4.72	7

Table S10 | Detailed metrics of each circuit module of the memristive hardware system in the mapping strategies

Mapping		Strategy A	Strategy B	Strategy C	Strategy D
Number of ADCs		16	32	96	128
Number of DACs		16	32	96	128
Number of TIAs		2	3	12	12
Area (mm ²)	ADC	0.8	1.6	4.8	6.4
	DAC	1.09	2.18	6.53	8.704
	TIA	0.66	0.99	3.96	3.96
	Array	0.68	1.81	1.36	3.62
	Sum	3.23	6.59	16.65	22.68
Energy (pJ)	ADC	7905	7225	7905	7225
	DAC	17262	17262	17262	17262
	TIA	535.68	489.6	535.68	489.6
	Array	166.82	297.89	166.82	297.89
	Sum	25869.5	25274.5	25869.5	25274.5
Latency (cycles)		6	4	1	1
Latency (ns)		120	80	20	20

Table S11 | Benchmark of the mapping strategies

	Strategy A	Strategy B	Strategy C	Strategy D
Operation (ops)	70004			
Performance (GOP s ⁻¹)	583.37	875.05	3500.2	3500.2

Power (mW)	215.58	315.93	1293.47	1263.72
Area (mm²)	3.23	6.59	16.65	22.68
Energy efficiency (TOP s⁻¹ W⁻¹)	2.71	2.77	2.71	2.77
Performance density (GOP s⁻¹ mm⁻², INT8)	180.79	133.07	210.28	154.3

a**b****c****d**
Fig. 5 | Hardware-software co-optimization results and comparison of DDK with deep learning and MFCC-based methods. a, Effectiveness of imprecision-based training and hybrid training for accuracy improvement in hardware implementations. **b,** Comparison of the DDK network (software: DDK; hardware experiment: DDK-exp) with other feature learning (DNN) and feature extraction (MFCC) techniques in various classification tasks, including audio, text, image, video and 3D object classification. The DDK neural network shows comparable performances with simultaneously greatly reduced number of parameters and operations (error bar is obtained by running simulations for 5 times). TF: Transformer. **c,** Energy consumption comparison between the DDK network and memristor-based DNN in **b.** **d,** Area comparison between the DDK network and memristor-based DNN in **b.**”

Reviewer 2

The manuscript is interesting; however, there are issues that should be solved prior to publication.

Comment 1. *At the introduction, the industrial use of memristor and current state-of-the-art features should be included, to let the reader have an exact picture of memristors in the nanoelectronics landscape.*

Answer: Thanks for your suggestion. Recently, memristors are being explored for their potential to revolutionize data storage and processing technologies. In the past few years, the switching endurance, data retention time, energy consumption, switching time, integration density, and price of memristive nonvolatile memories has been remarkably improved (depending on the materials used, values up to $\sim 10^{15}$ cycles, >10 years, ~ 0.1 pJ, ~ 10 ns, 256 gigabits per die, and $\leq \$0.30$ per gigabit have been achieved)⁴⁴. Resistive Random Access Memory (RRAM), leveraging its non-volatility, multi-bit storage, and CMOS compatibility, emerges as one of the most promising contenders for next-generation non-volatile memory (NVM).

As of the most recent advancements, major semiconductor manufacturers such as Taiwan Semiconductor Manufacturing Company (TSMC) and Semiconductor Manufacturing International Corporation (SMIC), among others, have successfully established commercially viable production lines for RRAM. TSMC has achieved mass production of chips utilizing a 40 nm node process, while its embedded RRAM with 28 nm and 22 nm nodes is currently in the production stage^{45,46}. Notably, TSMC has also implemented 32 Mb RRAM using 12 nm ultra-low power FinFET technology, showcasing its advanced capabilities in this field⁴⁷. The Institute of Microelectronics, in collaboration with SMIC, has developed a process for producing RRAM based on 14nm FinFET and 28nm technologies⁴⁸⁻⁵⁰.

Companies like Intel, Fujitsu, Infineon, and Panasonic are actively engaged in the development of embedded RRAM products. In collaboration with Interuniversity Microelectronics Centre, Panasonic has further advanced its RRAM technology and manufacturing capabilities. In 2015, Panasonic developed 40 nm RRAM chips, and by 2018, it had achieved the production of an 8 Mb

RRAM chip based on the 40 nm node⁵¹. In 2019, Intel fabricated a 7.2 Mb embedded RRAM array with 22 nm low-power FinFET (22FFL) technology. Fujitsu officially announced the launch of a 12 Mb RRAM chip, targeting its application in hearing aids and wearable devices like watches⁵². In 2022, Infineon developed a product-like 28 nm embedded RRAM demonstrator to allow fast reliability learning with high statistics⁵³. Furthermore, in 2024, Infineon's AURIX™ TC4x product series employs a 28 nm technology and introduces RRAM as the NVM for automotive applications⁵⁴. These developments underscore the rapid progress in RRAM technology and its increasing commercial viability across various applications.

Moreover, memristors are also ideal compute-in-memory (CIM) devices due to their ability to store and process data simultaneously, providing a promising non-von Neumann computing paradigm, thus eliminating the cost of data transfer^{55,56}. Memristor are recognized as promising “compute-with-physics” device that directly implements matrix-vector multiplication (MVM) using physical laws, namely Ohm’s law for multiplication and Kirchhoff’s law for summation⁴⁴. Recent studies have demonstrated in-memory MVM on RRAM chips, and have successfully achieved a range of AI tasks, including image classification, speech recognition, and image restoration, by leveraging diverse AI models such as convolutional neural networks (CNNs), long short-term memory (LSTM) networks, and probabilistic graphical models^{23,56}. As MVM is the most frequently used operation in deep learning, this implementation has resulted in greatly improved energy efficiency^{23,56-67}.

The corresponding text in the manuscript has been revised as follows: “Memristors are being explored for their potential to revolutionize data storage and processing technologies. In the past few years, the switching endurance, data retention time, energy consumption, switching time, integration density, and price of memristive nonvolatile memories has been remarkably improved (depending on the materials used, values up to $\sim 10^{15}$ cycles, >10 years, ~ 0.1 pJ, ~ 10 ns, 256 gigabits per die, and $\leq \$0.30$ per gigabit have been achieved)⁴⁴. Resistive Random Access Memory (RRAM), leveraging its non-volatility, multi-bit storage, and CMOS compatibility, emerges as one of the most promising contenders for next-generation non-volatile memory (NVM).

Recent advancements have seen major semiconductor manufacturers, including Taiwan Semiconductor Manufacturing Company (TSMC) and Semiconductor Manufacturing International Corporation (SMIC), successfully establish commercial RRAM production lines. TSMC has achieved mass production of 40 nm node chips and is currently producing embedded RRAM with 28 nm and 22 nm nodes^{45,46}. Additionally, TSMC has also implemented 32 Mb RRAM with 12 nm ultra-low power FinFET technology⁴⁷. Meanwhile, companies like Intel, Fujitsu, Infineon, and Panasonic have actively developed embedded RRAM products, showcasing significant progress in technology and increasing commercial viability⁵¹⁻⁵⁴. Fujitsu officially announced the launch of a 12 Mb RRAM chip, targeting its application in hearing aids and wearable devices like watches⁵². Infineon's AURIX™ TC4x series introducing RRAM as the NVM for automotive applications in 2024⁵⁴. These developments underscore the rapid progress in RRAM technology and its increasing commercial viability across various applications.”

Comment 2. *The fitting of beta and alpha is messy. Do the authors fit the experimental I-V curve with equations 3 and 4 and determine the parameters alpha and beta to obtain the best fit? Any algorithm to minimize the difference between experimental data and simulations? This issue should be explained in depth. In addition, the I-V curves of different pairs of alpha and beta parameters should be shown in comparison with measured I-V curves, to see the accuracy of the fitting.*

Answer: In response to your great suggestion, we have added a few sentences to explain this claim. The experimental I-V curves are fitted by equation 4 (assuming the alpha and beta are all nonnegative). The fitting is a nonlinear least squares problem. The minimization of the difference between experimental data and simulations is performed by the trust-region-reflective algorithm⁶⁸. Before fitting, the experimental data is normalized to a range from 0 to 1. The upper and lower boundaries for alpha and beta are set to 0 and 1, respectively. The initial guess values of alpha and beta are random values in the range from 0 to 1. The goodness of the fit is evaluated by the sum of the squared residuals. In the revised Fig. 3c in the manuscript (Fig. 11 in the response letter), the I-V curves of different pairs of alpha and beta parameters are shown by orange solid lines, while the measured I-V curves are depicted as blue scattered dots. The fitting accuracy is

represented as sum of the squared residuals shown in individual plots. The curve fitting details are added in the methods section in the revised manuscript.

Fig. 11 | The details of the device conductance change in the 8 utterances of the hardware feature map (blue whale). The fitted I-V curves of different pairs of alpha and beta parameters are shown by orange solid lines, while the measured I-V curves are depicted as blue scattered dots. The fitting accuracy is represented as sum of the squared residuals (represented as R) shown in individual plots.

The corresponding text in the manuscript has been revised as follows: “The conductance evolution of 8 utterances from blue whale sound is illustrated as an example in Fig. 3c. The fitted I-V curves of different pairs of alpha and beta parameters are shown by orange solid lines, while the measured I-V curves are depicted as blue scattered dots. The fitting accuracy is represented as sum of the squared residuals (represented as R) shown in individual plots.

Fig. 3 | Construction of memristor-based feature maps. **a**, A flow chart for feature map construction from 1-D waveforms. Firstly, the waveform is input into the DDK equation (algorithm) or transformed into a series of voltage pulses (hardware). Then, the output of the

equation or the conductance response of the memristor constitutes the feature map. **b**, Hardware-level implementation of feature maps from whale sounds in WMMS. The sound waveforms of blue, dwarf, fin and humpback whales (upper panel) are sub-sampled into 8 utterances and average pooled (1×60). The corresponding feature maps are shown in the lower panel. **c**, The details of the device conductance change in the 8 utterances of the hardware feature map (blue whale). The fitted I-V curves of different pairs of alpha and beta parameters are shown by orange solid lines, while the measured I-V curves are depicted as blue scattered dots. The fitting accuracy is represented as sum of the squared residuals (represented as R) shown in individual plots. **d**, Comparison of the proposed feature map construction method in terms of hardware latency and energy consumption with main-stream MFCC-based hardware. Our method shows significantly reduced latency and energy consumption.

Methods

Curve fitting

The experimental I-V curves are fitted by equation (4) (assuming the alpha and beta are all nonnegative). The fitting is a nonlinear least squares problem. The minimization of the difference between experimental data and simulations is performed by the trust-region-reflective algorithm⁶⁸. Before fitting, the experimental data is normalized to a range from 0 to 1. The upper and lower boundaries for alpha and beta are set to 0 and 1, respectively. The initial guess values of alpha and beta are random values in the range from 0 to 1.”

Comment 3. *The authors should explain how COMSOL simulations are connected to the model in equations 3 and 4? In addition, temperature plots obtained from COMSOL should be included in one or two panels (at the edge of the reset and set points) to describe the role of thermal effects in the resistive switching of the devices. Oxygen vacancies in the Ta₂O₅ layer should also be plotted in some panels for the set and reset processes. The simulation parameters are fitted to reproduce the I-V curves with just numerical criteria or they are physically based on DFT calculations or extracted from the literature? The negative activation energy for conduction in TaO_x can be supported by previous literature.*

Answer: Thank you for your suggestions. Thank you for your suggestions. The connection between COMSOL simulations and Equations (3) and (4) lies in their description of different levels of physical phenomena, but both involve the drift and diffusion of internal ions or defects during the resistive switching process of the device. Equations (3) and (4) provide simplified one-dimensional RS models with simplified drift-diffusion kinetics based on the assumptions of ohmic electron conduction, linear ion drift, linear dopant distribution in the dopant region and uniform electric field. COMSOL simulations use more complex three-dimensional models taking into account Fick diffusion flux and drift flux and Soret diffusion. Equations (3) and (4), as well as the COMSOL simulation model, both describe the drift and diffusion movements of ions or vacancies within the memristor. However, Equations (3) and (4) represent a simplified model based on highly idealized assumptions, whereas the COMSOL simulation model takes into account a wider range of practical factors, thus providing a more comprehensive and realistic representation of the device behavior. We have explained the connection between COMSOL simulations and Equations (3) and (4) in the new manuscript.

Besides, we have demonstrated the temperature and oxygen vacancy concentration (n_{V_o}) distributions in the TaO_x and HfO_x layers during the set and reset processes, including at the edges of the reset and set points (see Fig. S4-S6 in the Supplementary Information), and analyzed the temperature and oxygen vacancy changes during the set and reset processes. Fig. S5 and Fig. S6 illustrate the temperature and the distribution of n_{V_o} and in the TaO_x and HfO_x regions during the set and reset processes, respectively. As observed from Fig. S5, the highest temperature is recorded in the conductive filament (CF) area, with peak values of approximately 420°C (Fig. S5b) and 430°C (Fig. S5f) at the edge of the reset and set points, respectively. As evident from Fig. S6, during the reset process, a marked decrease of n_{V_o} is observed near the bottom electrode within the CF region at the edge of reset point (Fig. S6b, corresponding to Point B in Fig. S4b), indicating the onset of filament rupture, leading the device into a low-conductance state. Conversely, during the set process, a sudden increase of n_{V_o} near the bottom electrode within the CF region at the edge of set points (Fig. S6f, corresponding to Point F in Fig. S4b) indicates the growth of the CF, leading to the device transitioning to a high-conductance state.

These simulation parameters are derived from the literature, as shown in Table S1 in the manuscript. The negative activation energy for conduction in TaO_x is supported by previous literature (ACS Nano 2014, 8(3): 2369-2376)⁶⁹.

Fig. 12 | Finite-element modeling of the TiN/TaO_x/HfO_x/TiN device. a, Device geometry and boundary conditions. The device is modeled as a rotational symmetry cylinder. **b,** Simulated and measured *I-V* curves. The simulated results can well reproduce the RS of real devices.

Fig. 13 | The temperature distribution for the TiN/TaO_x/HfO_x/TiN device during the reset (a-d) and set processes (e-h). The temperature distribution of **a**, at the initial stage (point A in Fig. 12b), **b**, near the reset voltage (point B in Fig. 12b), **c**, at the max negative voltage (point C in Fig. 12b), **d**, after completing the negative voltage scan (point D in Fig. 12b), **e**, near set voltage (point E in Fig. 12b), **f**, at the max voltage (point F in Fig. 12b), **g**, after set (point G in Fig. 12b), **h**, after completing the positive voltage scan.

Fig. 14 | The oxygen vacancy distribution for the TiN/TaO_x/HfO_x/TiN device during the reset (a-d) and set processes (e-h). The oxygen vacancy distribution of **a**, at the initial stage (point A in Fig. 12b), **b**, at the edge of reset voltage (point B in Fig. 12b), **c**, at the max negative voltage (point C in Fig. 12b), **d**, after completing the negative voltage scan (point D in Fig. 12b), **e**, near set voltage (point E in Fig. 12b), **f**, at the edge of set voltage (point F in Fig. 12b), **g**, after set (point G in Fig. 12b), **h**, after completing the positive voltage scan.

Table 2 | Material parameters in finite-element simulation

Symbol	Value	Description
a^{69}	0.1 [nm]	Hopping distance
f^0	1×10^{12} [Hz]	Escape-attempt frequency
$E_{a_{TaO_x}}^{69}$	0.68 [eV]	Diffusion activation energy of TaO _x

T_0	293.15 [K]	Initial temperature
λ^{69}	0.1 [1/K]	Thermal coefficient
$k_{HfO_x}^{71}$	0.5 [W/(m•K)]	Thermal conductivity of HfO _x for $n_{V_o^{\bullet}} = 0 \text{ cm}^{-3}$
$k_{TaO_x}^{72}$	3 [W/(m•K)]	Thermal conductivity of TaO _x at T_0 for $n_{V_o^{\bullet}} = 1 \times 10^{21} \text{ cm}^{-3}$
$k_{electrode}$	5 [W/(m•K)]	Electrode thermal conductivity of TiN
$E_{a_{HfO_x}}^{73}$	0.65 [eV]	Diffusion activation energy of HfO _x
$E_{ac_{TaO_x}}^{69}$	-0.006 [eV]	Activation energy for conduction of TaO _x for $n_{V_o^{\bullet}} = 1 \times 10^{21} \text{ cm}^{-3}$
$E_{ac_{HfO_x}}^{74}$	0.05 [eV]	Activation energy for conduction of HfO _x for $n_{V_o^{\bullet}} = 0 \text{ cm}^{-3}$
$\sigma_{TaO_x}^{69}$	9.4×10^4 [S/m]	Electrical conductivity of TaO _x for $n_{V_o^{\bullet}} = 1 \times 10^{21} \text{ cm}^{-3}$
$\sigma_{HfO_x}^{74,75}$	3 [S/m]	Electrical conductivity of HfO _x for $n_{V_o^{\bullet}} = 0$
$\sigma_{electrode}$	3×10^4 [S/m]	Electrical conductivity of TiN

The corresponding text in the manuscript has been revised as follows: “

Methods

Fig. S5 and Fig. S6 illustrate the distribution of $n_{V_o^{\bullet}}$ and the temperature in the TaO_x and HfO_x regions during the set and reset processes, respectively. As observed from Fig. S5, the highest temperature is recorded in the conductive filament (CF) area, with peak values of approximately 420°C (Fig. S5b) and 430°C (Fig. S5f) at the edge of the reset and set points, respectively. As evident from Fig. S6, during the reset process, a marked decrease of $n_{V_o^{\bullet}}$ is observed near the bottom electrode within the CF region at the edge of reset point (Fig. S6b, corresponding to Point B in Fig. S4b), indicating the onset of filament rupture, leading the device into a low-conductance state. Conversely, during the set process, a sudden increase of $n_{V_o^{\bullet}}$ near the bottom electrode within

the CF region at the edge of set points (Fig. S6f, corresponding to Point F in Fig. S4b) indicates the growth of the CF, leading to the device transitioning to a high-conductance state.

The connection between COMSOL simulations and Equations (3) and (4) lies in their description of different levels of physical phenomena, but both involve the behavior of internal ions or defects (such as oxygen vacancies, $V_o^{\cdot\cdot}$) during the resistive switching process of the device. Equations (3) and (4) describe a simplified model that captures the fundamental dynamics of conductive filament (CF) growth and rupture during resistive switching based on the highly idealized assumptions of ohmic electron conduction, linear ion drift, linear dopant distribution in the dopant region and uniform electric field. These equations focus on the change in CF length $w(t)$ over time, where y can be considered an alternative variable for $w(t)$, and x represents the externally applied voltage or current. Equation (3) corresponds to the SET process, while Equation (4) corresponds to the RESET process. The term βx describes the drift motion of ions or defects, and the term $\frac{\alpha}{y}$ reflects the diffusion effect. On the other hand, COMSOL simulations employ a more detailed physical model. Equation (7) describes the changes in the concentration of oxygen vacancies ($n_{V_o^{\cdot\cdot}}$), considering diffusion, drift, and the influence of temperature gradients. Specifically, the term $D\nabla n_{V_o^{\cdot\cdot}}$ and $z_i \frac{D_i}{RT} F n_{V_o^{\cdot\cdot}} \nabla V$ stand for Fick diffusion flux and drift flux, respectively. The term $DS n_{V_o^{\cdot\cdot}} \nabla T$ is the Soret diffusion flux. Soret diffusion is the movement of molecules along a temperature gradient. Equations (3) and (4), as well as the COMSOL simulation model, both describe the drift and diffusion movements of ions or vacancies within the memristor. However, Equations (3) and (4) represent a simplified model based on highly idealized assumptions, whereas the COMSOL simulation model considers a wider range of practical factors, thus providing a more comprehensive and realistic representation of the device behavior.”

Supplementary Information

Fig. S4 | Finite-element modeling of the TiN/TaO_x/HfO_x/TiN device. a, Device geometry and boundary conditions. The device is modeled as a rotational symmetry cylinder. **b**, Simulated and measured *I-V* curves. The simulated results can well reproduce the RS of real devices.

Fig. S5 | The temperature distribution for the TiN/TaO_x/HfO_x/TiN device during the reset (a-d) and set processes (e-h). The temperature distribution of **a**, at the initial stage (point A in Fig. S4b), **b**, near the reset voltage (point B in Fig. S4b), **c**, at the max negative voltage (point C in Fig. S4b), **d**, after completing the negative voltage scan (point D in Fig. S4b), **e**, near set voltage (point

E in Fig. S4b), **f**, at the max voltage (point F in Fig. S4b), **g**, after set (point G in Fig. S4b), **h**, after completing the positive voltage scan.

Fig. S6 | The oxygen vacancy distribution for the TiN/TaO_x/HfO_x/TiN device during the reset (a-d) and set processes (e-h). The oxygen vacancy distribution of **a**, at the initial stage (point A in Fig. S4b), **b**, at the edge of reset voltage (point B in Fig. S4b), **c**, at the max negative voltage (point C in Fig. S4b), **d**, after completing the negative voltage scan (point D in Fig. S4b), **e**, near set voltage (point E in Fig. S4b), **f**, at the edge of set voltage (point F in Fig. S4b), **g**, after set (point G in Fig. S4b), **h**, after completing the positive voltage scan.

Table S1 Material parameters in finite-element simulation

Symbol	Value	Description
a^{69}	0.1 [nm]	Hopping distance
f^0	1×10^{12} [Hz]	Escape-attempt frequency
$E_{a_{TaO_x}}^{69}$	0.68 [eV]	Diffusion activation energy of TaO _x

T_0	293.15 [K]	Initial temperature
λ^{69}	0.1 [1/K]	Thermal coefficient
$k_{HfO_x}^{71}$	0.5 [W/(m·K)]	Thermal conductivity of HfO _x for $n_{V_O^{\bullet\bullet}} = 0 \text{ cm}^{-3}$
$k_{TaO_x}^{72}$	3 [W/(m·K)]	Thermal conductivity of TaO _x at T_0 for $n_{V_O^{\bullet\bullet}} = 1 \times 10^{21} \text{ cm}^{-3}$
$k_{electrode}$	5 [W/(m·K)]	Electrode thermal conductivity of TiN
$E_{a_{HfO_x}}^{73}$	0.65 [eV]	Diffusion activation energy of HfO _x
$E_{a_{TaO_x}}^{69}$	-0.006 [eV]	Activation energy for conduction of TaO _x for $n_{V_O^{\bullet\bullet}} = 1 \times 10^{21} \text{ cm}^{-3}$
$E_{a_{HfO_x}}^{74}$	0.05 [eV]	Activation energy for conduction of HfO _x for $n_{V_O^{\bullet\bullet}} = 0 \text{ cm}^{-3}$
$\sigma_{TaO_x}^{69}$	9.4×10^4 [S/m]	Electrical conductivity of TaO _x for $n_{V_O^{\bullet\bullet}} = 1 \times 10^{21} \text{ cm}^{-3}$
$\sigma_{HfO_x}^{74,75}$	3 [S/m]	Electrical conductivity of HfO _x for $n_{V_O^{\bullet\bullet}} = 0$
$\sigma_{electrode}$	3×10^4 [S/m]	Electrical conductivity of TiN

Notes:

Diffusion activation energy (E_a): In general, the E_a of HfO₂ is conventionally considered to be 0.7 eV⁷³. The E_a of TaO_x has been reported as 0.85 eV in some literature⁶⁹. To achieve the best fitting results between simulation data and our experimental results, we set the E_a of HfO_x ($E_{a_{HfO_x}}$) to 0.65 eV and that of TaO_x ($E_{a_{TaO_x}}$) to 0.68 eV, values which are close to those previously reported^{69,73}.

Thermal conductivity:

According to literature reports, the thermal conductivity of TaO_x (k_{TaO_x}) thin films varies between 0.9 W/m·K (low $n_{V_O^{\bullet\bullet}}$) and 4 W/m·K (high $n_{V_O^{\bullet\bullet}}$)⁷². In this study, we choose a thermal conductivity value of 3 W/m·K for TaO_x at $n_{V_O^{\bullet\bullet}} = 1 \times 10^{21} \text{ cm}^{-3}$, which is consistent with the reported values in the literature⁷².

Electrical conductivity: HfO₂ is commonly used as a high-k gate insulator in microelectronic devices, and its electrical conductivity is influenced by various factors. According to previous reports, the electrical conductivity of HfO₂ is extremely low when the n_{V_0} is zero^{74,75}. In this study, we set the electrical conductivity of HfO_x to 3 S/m for $n_{V_0}=0\text{ cm}^{-3}$, which is consistent with prior research reporting values ranging from 0 to 1000 S/m^{74,75}.”

Comment 4. *The role of cycle-to-cycle variability is not explained. How a simple model such as the one presented here can deal with it?, and how it affects the accuracy of the neural network, when variability affects the synaptic weights? See the complexity of variability compact modeling in memristors in the following reference (<https://onlinelibrary.wiley.com/doi/10.1002/aisy.202200338>).*

Answer: Thank you for the valuable suggestion. Cycle-to-cycle variability holds an essential position in the practical application of RRAM technology, as it profoundly influences the reliability and stability of RRAM devices. We have carefully studied the literature suggested by you and realized the significance of constructing variability physical models in circuit and algorithm design. These approaches for variability model construction includes two-dimensional variation coefficients, time series analysis, kinetic Monte Carlo simulations, and statistical methods. This reference and relevant ones are really helpful in understanding the origin of cycle-to-cycle variations and demonstrates the importance of cycle-to-cycle variations in practical applications, such as the accuracy of neural networks⁷⁶⁻⁸⁰. Therefore, we cite these references in our revised manuscript. In our work, cycle-to-cycle variability is not limited to affecting the precision of α and β in the feature extraction DDK layer; it also directly leads to fluctuations in synaptic weights in the classification layer, thereby impacting the performance of neural networks.

To make the model be able to deal with the above cycle-to-cycle variations, we model the variations as Gaussian noise according to the suggested reference paper and the experimental data. The noise is added to the target conductance in the RRAM synaptic devices in the classification layer and added to the α and β of the RRAM conductance in the DDK layer. The noise value equals to the standard deviation of the distribution. In the simulation process, the study range for α and β

noise is set between 0 and 0.5, while the range for synaptic noise is set between 0 and 2.5 (unit: μS).

To visually demonstrate the specific impact of cycle-to-cycle variability on neural network accuracy, Fig. 15 clearly present the impact of α noise, β noise, and synaptic noise on the classification accuracy of DDK network on the SITW dataset in a 10-speaker recognition task. Fig. 15a shows that if the network is deployed directly after off-chip training, the classification accuracy of the DDK network gradually declines as the α noise, β noise, and synaptic noise increase. Fig. 15b further indicates that when the proposed hybrid training strategy is applied, the impact of these noises on classification accuracy is significantly mitigated. This finding not only highlights the crucial influence of cycle-to-cycle variability on neural network performance but also provides evidence to the high effectiveness of the proposed hybrid training strategy.

Fig. 15 | Influence of cycle-to-cycle variation on DDK network in audio classification. a, the impact of α noise, β noise, and synaptic noise on the classification accuracy of DDK network on the SITW dataset (deployed directly after off-chip training). **b,** the impact of α noise, β noise, and synaptic noise on the classification accuracy of DDK network on the SITW dataset when α and β are fixed, and synaptic weights are fine-tuned. The classification accuracy gradually decreases as the noise increases. After fine-tuning the synaptic weights, the impact of α noise, β noise, and synaptic noise on the classification accuracy is significantly reduced.

The corresponding text in the manuscript has been revised as follows: “The impact of cycle-to-cycle variability on neural network accuracy is simulated in the speaker recognition task (Fig.

S16). The variation is modeled as Gaussian noise⁷⁶⁻⁸⁰. The noise is added to the target conductance in the RRAM synaptic devices in the classification layer and the α and β of the RRAM conductance in the DDK layer. The noise value equals to the standard deviation of the distribution. In the simulation process, the study range for α and β noise is set between 0 and 0.5, while the range for synaptic noise is set between 0 and 2.5 (unit: μS). The results further confirm the effectiveness of the proposed hybrid training strategy.

Fig. S16 | Influence of device noise on DDK network in audio classification. a, the impact of α noise, β noise, and synaptic noise on the classification accuracy of DDK network on the SITW dataset (deployed directly after off-chip training). **b**, the impact of α noise, β noise, and synaptic noise on the classification accuracy of DDK network on the SITW dataset when α and β are fixed, and synaptic weights are fine-tuned. The classification accuracy gradually decreases as the noise increases. After fine-tuning the synaptic weights, the impact of α noise, β noise, and synaptic noise on the classification accuracy is significantly reduced.

”

Comment 5. *The typical model developed in the classical paper of S. Williams group, and revisited in this manuscript, could be good enough to fit the I-V curves. However, in the back-propagation algorithm, the synaptic weights derivatives are needed (see Figure 4 in the main manuscript). The authors should comment on this issue. A plot on the capacity to reproduced experimental current derivatives would be welcome to clarify this issue.*

Answer: Thank you for the valuable suggestion. The backpropagation is performed on CPU using experimentally measured device conductance and current. During the backpropagation phase, the loss is defined as the deviation between the experimental current values and the actual target values. The gradient calculation for each layer involves the gradient of the loss function with respect to the conductance of the RRAM devices. The method for calculating the gradient of the loss function with respect to synaptic weights is described by equations 21 and 22 in the main text. The formula 21 describes specifically the error in the kth layer is the product of the synaptic weights of the (k+1)th layer and the error of the (k+1)th layer, followed by the dot product with the derivative of the activation function in the kth layer. Formula 22 describes specifically the gradient of the loss function with respect to the weights is the product of the error in the kth layer and the output of the (k-1)th layer. The gradient of the loss function with respect to the conductance of the RRAM devices is calculated in the kth layer as formula (34-35) in response letter (replace $\frac{\partial C(\alpha, \beta, W)}{\partial W}$ in formula 22 in the main text with $\frac{\partial C(\alpha, \beta, W)}{\partial G}$, and W^{k+1} in formula 21 in the main text with G^{k+1}). The calculation of the error in the kth layer as follows:

$$\delta^k = G^{k+1} \delta^{k+1} \odot \sigma'(z^k) \quad (34)$$

δ^{k+1} is the error in the (k+1)th layer, which is transmitted back to the (k+1)th layer by the gradient of the loss function to softmax. G^{k+1} is value of the conductance of the synaptic weight mapping in the (k+1)th layer. $\sigma'(z^k)$ is the derivative of the activation function with respect to output current in the kth layer. The calculation of gradient of the loss function with respect to the conductance of the RRAM devices as follow:

$$\frac{\partial C(\alpha, \beta, W)}{\partial G} = \delta^k a^{k-1} \quad (35)$$

a^{k-1} is the output in the (k-1)th layer. Specifically, it is the value of the output current of the (k-1)th layer after it has been processed by the activation function. The activation function is implemented on CPU.

To better demonstrate the effectiveness of the DDK network, we have taken your advice and provided a gradient plot here to show the changes in synaptic weight gradients of the DDK network as the number of iterations increases (in the 10-speaker recognition task of the SITW dataset). As illustrated in Fig. 16, with the increase in the number of iterations, the gradients of the hidden layer

weight (weight1_grad) and output layer weight (weight_grad), all exhibit a downward trend, indicating that the training process is proceeding smoothly and gradually converging.

Fig. 16 | The changes in weight gradients of the DDK network in SITW dataset during the training. The gradients of the hidden layer weight (weight1_grad) and output layer weight (weight_grad) exhibit a downward trend as the number of iterations increases.

The corresponding text in the manuscript has been revised as follows: “

Methods

Backpropagation-based feature learning

In hardware implementation, the backpropagation is performed on CPU using experimentally measured device conductance and current. The gradient of the loss function with respect to the conductance of the RRAM devices is calculated as follows:

$$\delta^k = G^{k+1} \delta^{k+1} \odot \sigma'(z^k) \quad (30)$$

δ^{k+1} is the error in the $(k+1)$ th layer, which is transmitted back to the $(k+1)$ th layer by the gradient of the loss function to softmax. G^{k+1} is value of the conductance of the synaptic weight mapping

in the $(k+1)$ th layer. $\sigma'(z^k)$ is the derivative of the activation function with respect to output current in the k th layer. The calculation of gradient of the loss function with respect to the conductance of the RRAM devices as follow:

$$\frac{\partial C(\alpha, \beta, W)}{\partial G} = \delta^k a^{k-1} \quad (31)$$

a^{k-1} is the output in the $(k-1)$ th layer. Specifically, it is the value of the output current of the $(k-1)$ th layer after it has been processed by the activation function. The activation function is implemented on CPU.”

Comment 6. *In the final part of the manuscript, the neural network is described. However, the authors use a confusing writing style where software and hardware parts are intertwined. It is necessary to explain which part of the network are hardware-based, and the corresponding experimental data obtained, and which part are software-based. How is the connection between experimental data and software-based data done?*

Answer: Thank you for your suggestions. The entire DDK neural network is experimentally realized in hardware. The difference exists in how the different parts of the DDK neural network, such as the feature extraction DDK layer and the classification layer, are trained. In the proposed hybrid training technique, the DDK layer is firstly trained on software and then deployed on hardware, while the classification layer is trained *in-situ* on hardware. We have rewritten the content about the network and clearly explained the implementation process of the network. In Fig. 5a, all blue-filled bar charts are software simulation results, while the orange-filled bar charts are hardware experimental result.

The corresponding text in the manuscript has been revised as follows: “When the entire network is exclusively trained based on software without non-idealities (referred to as software training), the accuracy of software testing can achieve 94.8%. Nevertheless, when simulating the hardware inference process by incorporating experimental noise into the aforementioned network (imprecise inference after software training), the accuracy is merely 15.5%. The reason is that software training cannot account for hardware imperfections, leading to a diminished level of

network resilience. If the DDK and classification layer are all trained on software with noises (imprecision-based training), the test accuracy can reach 91.2%. The performance can be enhanced further by adopting the fine-tuning training method on the classification layer (imprecision-based hybrid training). This approach involves preserving or fixing the pre-trained parameters of the DDK layers, which were originally obtained through software training with noise, while solely retraining the classification layer on software with noise. Using this strategy, we ultimately achieve an enhanced robustness and attain an accuracy of 92.8%, reflecting a substantial improvement in performance. The DDK network was experimentally trained on memristive 1T1R chips using the proposed hybrid training method (Fig. 4c). The feature maps of the ten speakers in dataset are experimentally obtained with memristors, as shown in Fig. S14. The DDK layer is firstly trained in software and then deployed on hardware, while the classification layer is trained *in-situ* on hardware (experimental hybrid training). Training procedure and parameters are shown in Fig. S15 and Table S4, respectively. This experimental accuracy of 92.5% demonstrates the efficacy of this strategy in enhancing classification accuracy on practical memristive hardware.”

Comment 7. *Readme files are included. The code is sorted out.*

Answer: Thanks. The readme files are now added and the code is sorted out.

Reviewer 3

The authors fabricated a bilayer memristor with structure TiN/TaO/HfO/TiN, and built a 2 parameter-related (drift-diffusion) device-level model based on its electrical characteristics. The model is further developed to implement feature learning for pattern classification with higher energy efficiency. The hardware-based feature learning is interesting and the topic is suitable for the journal. I would in principle recommend publication, while some minor revisions are still needed. The following suggestions and questions may be helpful to the authors.

Comment 1. *Please clarify why the device needs a higher gate voltage for RESET than SET? Especially when the RESET current is lower than the SET current. The same gate voltage in both SET and RESET process could ease the control. See Nature 2023 Hybrid 2D–CMOS microchips for memristive applications. <https://www.nature.com/articles/s41586-023-05973-1>.*

Answer: Thank you. This suggestion is really helpful since we indeed neglect the study on gate voltages in RESET process. Following your suggestion and the mentioned excellent paper, we carefully studied the influence of the RESET gate voltage on the resistance states of the memristor device. After forming, we firstly switched the device to a low resistance state at a fixed SET gate voltage (1.8 V in our experiment) and then measured the device resistance states after RESET at different RESET gate voltages (from 0.8 V to 2.4 V with a 0.1 V step). The experiment results are shown in Fig. 17. It is obvious that the device can be operated at various RESET gate voltages, the value of the RESET gate voltage only influences the high resistance state that can be achieved after the RESET operation. The higher the RESET gate voltage, the greater the resistance states following RESET. Therefore, we highly agree with you that the same gate voltage in both SET and RESET process could be used and ease the control. We replaced the I-V curve in Fig. 2a in the original manuscript with a new I-V curve measured with the same gate voltage at 1.5 V in both SET and RESET process, as shown in Fig. 18 in the response letter.

Fig. 17 | Experimentally measured resistance states of the memristor device switched at different pairs of set and reset gate voltages. The set gate voltage is fixed at 1.8 V, while the reset gate voltage varies from 0.8 V to 2.4 V with a step of 0.1 V. The resistance states after RESET increase in proportion to the RESET gate voltage.

Fig. 18 | A typical I - V curve of the bipolar RS behavior of the memristor device. The SET process is more abrupt than the RESET one. Gate voltages for SET ($V_{g,set}$) and RESET ($V_{g,reset}$) are both 1.5 V.

The corresponding text in the manuscript has been revised as follows: “

Fig. 2 | Characterization of TiN/TaO_x/HfO_x/TiN memristive crossbar array cells. a, A typical I - V curve of the bipolar RS behavior. The SET process is more abrupt than the RESET one. **Gate voltages for SET ($V_{g,set}$) and RESET ($V_{g,reset}$) are both 1.5 V.** **b, c and d,** Tunability of parameters α and β in the proposed algorithm with input pulse configurations applied on a single memristor at various initial resistances. The influences from pulse amplitude (b), pulse width (c) and initial resistance (d) are studied in cycle-to-cycle test (100 cycles), respectively. Statistics of α and β are

presented in error band diagrams (95% confident interval). Parameter α is almost insensitive to input pulse and initial resistance, while parameter β is positively proportional to the above factors. Moreover, when the initial resistance becomes larger, β saturates.”

Comment 2. *The pulsed I-V-t curves of the memristor are needed apart from Fig.2a, as Fig.2b-c are based on pulse operation, with different voltage/pulse width/initial resistance. So the readers can see the real operation region of the device.*

Answer: In response to your suggestion, we have provided the pulsed I-V-t curves, as shown in Fig. 8. The programming scheme is shown in Fig. 8a. For the long-term potentiation (LTP) process, the voltage of the transistor gate (WL) increases linearly (from 0.9 V to 2.0 V), where the voltage of the top electrode is fixed at 2.2 V and the voltage of the source is ground. For the long-term depression (LTD) process, each operation includes a SET and RESET pulse. For each operation, the RESET pulse is applied first, then the SET pulse is applied. The RESET voltages of the gate, source, and top electrode are fixed (3.7 V, ground, and -3.3 V separately). The SET voltage of the gate decreases linearly (from 2.0 V to 0.9 V). The SET voltage of the top electrode is fixed at 2.2V, while the source is grounded. The pulse width in the measurement is fixed at 5 ns. The conductance is read at 0.2 V. Fig. 8b shows a typical LTP/LTD curve of the memristor device. These results indicate linear and symmetric conductance change with an operation region from 6.1 μS to 202.1 μS .

Fig. 8 | The pulsed I - V - t curves of the memristor. a, Programming scheme. For the LTP process, the voltage of the transistor gate (WL) increases linearly (from 0.9 V to 2.0 V), where the voltage of the top electrode is fixed at 2.2 V and the voltage of the source is ground. For the LTD process, each operation includes a SET and RESET pulse. For each operation, the RESET pulse is applied first, then the SET pulse is applied. The RESET voltages of the gate, source, and top electrode are fixed (3.7 V, ground, and -3.3 V separately). The SET voltage of the gate decreases linearly (from 2.0 V to 0.9 V). The SET voltage of the top electrode is fixed at 2.2V, while the source is grounded. The pulse width in the measurement is fixed at 5 ns. The conductance is read at 0.2 V. **b**, A typical LTP/LTD curve of the memristor device.

The corresponding text in the manuscript has been revised as follows: “The pulsed I - V - t curves are shown in Fig. S7, showing linear and symmetric conductance change with an operation region from 6.1 μS to 202.1 μS .”

Supplementary Information

Fig. S7 | The pulsed I - V - t curves of the memristor. a, Programming scheme. For the LTP process, the voltage of the transistor gate (WL) increases linearly (from 0.9 V to 2.0 V), where the voltage of the top electrode is fixed at 2.2 V and the voltage of the source is ground. For the LTD process, each operation includes a SET and RESET pulse. For each operation, the RESET pulse is applied first, then the SET pulse is applied. The RESET voltages of the gate, source, and top electrode are fixed (3.7 V, ground, and -3.3 V separately). The SET voltage of the gate decreases linearly (from 2.0 V to 0.9 V). The SET voltage of the top electrode is fixed at 2.2V, while the source is grounded. The pulse width in the measurement is fixed at 5 ns. The conductance is read at 0.2 V. b, A typical LTP/LTD curve of the memristor device.”

Comment 3. Please define “ y ” more specifically, like resistivity or conductivity? Figure 1 and Fig.S2 should be better explained with real physical meaning. For example, Instead of describing the clear mathematic relation like “Fig. S2, α is close related to an asymptote position of y , while β mainly controls the decrease speed... With the increasing of β , the output y decreases more rapidly to asymptote so that it keeps almost constant even under subsequent input signals.”, which could be further explained as “dopants mobility could induce higher conductance or resistance change” “dopants diffusion coefficient determines the final conductance or resistance level could be achieved”, if I understand correctly.

Answer: Thanks. Upon your recommendation, we have explained the Fig. 1 and Fig. S2 with real physical meaning. In Fig. S2, the diffusion coefficient of dopants ($D = \alpha$) determines the final conductance, while the mobility of dopants ($\mu = \beta L$) determines the rate of conductivity change.

The corresponding text in the manuscript has been revised as follows: “In memristors, $\alpha = D$, $\beta = \mu/L$, $y = w(t)$ and $x = v(t)$. y is output or the state variable of the memristor device that represents the length of the doped region within the resistive switching layer after the diffusion and migration of dopants. From Equation (1), it is evident that since R_{on} is significantly smaller than R_{off} ($R_{on} \ll R_{off}$), the resistance of the device is primarily influenced by the second term ($R_{OFF} \left(1 - \frac{w(t)}{L}\right)$). Specifically, as the variable $w(t)$ increases, the resistance of the device decreases, clearly illustrating a negative correlation between w and the device resistance. Therefore, the y in Equations (3) and (4) is positively correlated with the conductivity of the memristor device. α and β are positive constants controlling increase and decrease speed of y , respectively. Equations (3) and (4) describe the dynamics behavior of y at input x .

As shown in Fig. S2, α is close related to an asymptote position of y , while β mainly controls the decrease speed. This indicates that the diffusion coefficient of dopants ($D = \alpha$) mainly determines the final conductance, while the mobility of dopants ($\mu = \beta L$) influences primarily the rate of conductivity change. With the increasing of β , the output y decreases more rapidly to asymptote so that it keeps almost constant even under subsequent input signals. This observation suggests that as the mobility of dopants increases, the rate of conductance declines more precipitously. Therefore, combination of diffusion coefficient of dopants and mobility of dopants can effectively control not only the feature learning process, but also the length of information where feature learning is performed.”

Comment 4. *The authors define $\alpha=D$ =dopant diffusion coefficient, $\beta=u/L$ =dopant mobility/ L , while D and u normally are fixed number once the material combination has been determined. The authors need to explain how pulse configurations changes the dopant diffusion coefficient and*

dopant mobility. If not, please specify which methods could be used to adjust u and D , as variables.

Answer: Thank you for the insightful suggestion. The different tunability of α and β are discussed in the following. From the physical meanings in equation (2) in the revised manuscript, α ($=D$) and β ($=\mu/L$) should be insensitive to external electric field. However, the normalized $x(t)$, other than the real voltage configuration it is mapped to, is used as input in the curve-fitting, thus, the impact of pulse configuration or initial state is reflected in β value. This method can overcome difficulties in representing pulse configuration and initial state that are physical multiparameter as a single mathematical value. This can be understood as follows:

$$\frac{dy}{dt} = \frac{\alpha}{y} - \beta X, \quad (36)$$

where X is the unnormalized input. After normalization:

$$X = \gamma x, \quad (37)$$

where x is the normalized input and γ is the scale factor. Therefore,

$$\frac{dy}{dt} = \frac{\alpha}{y} - \beta\gamma x = \frac{\alpha}{y} - Bx, \quad (38)$$

where $B = \beta\gamma$ is the fitted value of β when using normalized input x .

The corresponding text in the manuscript has been revised as follows: “The different tunability of α and β are discussed in the following. From the physical meanings in equation (2), α ($=D$) and β ($=\mu/L$) should be insensitive to external electric field. However, the normalized $x(t)$, other than the real voltage configuration it is mapped to, is used as input in the curve-fitting, thus, the impact of pulse configuration or initial state is reflected in β value. This method can overcome difficulties in representing pulse configuration and initial state that are physical multiparameter as a single mathematical value.”

Comment 5. Please provide a semi-log curve for Fig. S1.

Answer: Thanks. The semi-log curves for Fig. S1 are provided in the revised manuscript.

The corresponding text in the manuscript has been revised as follows: “

Fig. S1 | Typical RS curves obtained by the simplified drift-diffusion model. a, Abrupt RS. When the doped region arrives the boundary of the device, abrupt RS occurs. Typical parameters are $v_0 = 5$ V, $\omega_0 = 10$, $\mu = 6.3 \times 10^{-14}$ m²V⁻¹s⁻¹, $R_{\text{off}}/R_{\text{on}} = 450$ and $L = 10$ nm. **b,** NDR-type RS. When the doped region approaches the boundary of the device, NDR-type RS occurs. Typical parameters are $v_0 = 2.4$ V, $\omega_0 = 10$, $\mu = 6.3 \times 10^{-14}$ m²V⁻¹s⁻¹, $R_{\text{off}}/R_{\text{on}} = 450$ and $L = 10$ nm.”

Comment 6. *Modify Fig. S4c-e using the same scale to show better visualization.*

Answer: Thank you for your suggestion. We have revised Fig. S4c-e in the previous manuscript and adopted the same scale bar in the new manuscript. The Fig. S4c-e are now expanded to Fig. S5 and Fig. S6 in the revised version to show more details in the resistive switching.

The corresponding text in the manuscript has been revised as follows: “

Fig. S4 | Finite-element modeling of the TiN/TaO_x/HfO_x/TiN device. a, Device geometry and boundary conditions. The device is modeled as a rotational symmetry cylinder. b, Simulated and measured I - V curves. The simulated results can well reproduce the RS of real devices.

Fig. S5 | The temperature distribution for the TiN/TaO_x/HfO_x/TiN device during the reset (a-d) and set processes (e-h). The temperature distribution of a, at the initial stage (point A in Fig. S4b), b, near the reset voltage (point B in Fig. S4b), c, at the max negative voltage (point C in Fig. S4b), d, after completing the negative voltage scan (point D in Fig. S4b), e, near set voltage (point

E in Fig. S4b), **f**, at the max voltage (point F in Fig. S4b), **g**, after set (point G in Fig. S4b), **h**, after completing the positive voltage scan.

Fig. S6 | The oxygen vacancy distribution for the TiN/TaO_x/HfO_x/TiN device during the reset (a-d) and set processes (e-h). The oxygen vacancy distribution of a, at the initial stage (point A in Fig. S4b), b, at the edge of reset voltage (point B in Fig. S4b), c, at the max negative voltage (point C in Fig. S4b), d, after completing the negative voltage scan (point D in Fig. S4b), e, near set voltage (point E in Fig. S4b), f, at the edge of set voltage (point F in Fig. S4b), g, after set (point G in Fig. S4b), h, after completing the positive voltage scan.”

Reference

- 1 Yao, S. *et al.* FastDeepIoT: Towards understanding and optimizing neural network execution time on mobile and embedded devices. In *Proceedings of the 16th ACM Conference on Embedded Networked Sensor Systems*. 278-291 (ACM).
- 2 Jierun Chen, S.-h. K., Hao He, Weipeng Zhuo, Song Wen, Chul-Ho Lee, S.-H. Gary Chan. Run, Don't Walk: Chasing Higher FLOPS for Faster Neural Networks. In *Proceedings of the IEEE/CVF conference on computer vision and pattern recognition*. 2303 (IEEE).
- 3 Kaiming He, X. Z., Shaoqing Ren, Jian Sun. Deep residual learning for image recognition. In *Proceedings of the IEEE conference on computer vision and pattern recognition*. 770-778 (IEEE).
- 4 Menghani, G. Efficient Deep Learning: A Survey on Making Deep Learning Models Smaller, Faster, and Better. *ACM Computing Surveys* **55**, 1-37 (2023).
- 5 Johannes Getzner, B. C., Stephan Günnemann. Accuracy is not the only Metric that matters: Estimating the Energy Consumption of Deep Learning Models. *arXiv preprint arXiv:2304.00897* (2023).
- 6 Shan, W. *et al.* A 510-nW Wake-Up Keyword-Spotting Chip Using Serial-FFT-Based MFCC and Binarized Depthwise Separable CNN in 28-nm CMOS. *IEEE J. Solid-State Circuits* **56**, 151-164 (2021).
- 7 Giraldo, J. S. P., Lauwereins, S., Badami, K. & Verhelst, M. Vocell: A 65-nm Speech-Triggered Wake-Up SoC for 10- μ W Keyword Spotting and Speaker Verification. *IEEE J. Solid-State Circuits* **55**, 868-878 (2020).
- 8 Li, Q. *et al.* MSP-MFCC: Energy-Efficient MFCC Feature Extraction Method With Mixed-Signal Processing Architecture for Wearable Speech Recognition Applications. *IEEE Access* **8**, 48720-48730 (2020).
- 9 Jo, J., Yoo, H. & Park, I.-C. Energy-efficient floating-point MFCC extraction architecture for speech recognition systems. *IEEE Transactions on Very Large Scale Integration Systems* **24**, 754-758 (2016).
- 10 Ramos-Lara, R., López-García, M., Cantó-Navarro, E. & Puente-Rodríguez, L. Real-time speaker verification system implemented on reconfigurable hardware. *J. Signal Process. Sys.* **71**, 89-103 (2013).
- 11 Haofeng, K., Weijia, S., Lane, I. & Chong, J. Efficient MFCC feature extraction on Graphics Processing Units. In *Constantinides International Workshop on Signal Processing*. 1-4 (IET, 2013).
- 12 Techpowerup. GPU Database of NVIDIA GeForce GTX 580, <<https://www.techpowerup.com/gpu-specs/geforce-gtx-580.c270>> (2016).
- 13 Manikandan, J., Venkataramani, B., Girish, K., Karthic, H. & Siddharth, V. Hardware Implementation of Real-Time Speech Recognition System Using TMS320C6713 DSP. In *24th International Conference on VLSI Design*. 250-255 (IEEE, 2011).
- 14 TexasInstruments. TMS320C6711D, C6712D, C6713B Power Consumption Summary, <<https://www.ti.com/lit/pdf/spra889>> (2004).
- 15 Intel. *The Specification of Intel® Pentium® 4 Processor 1.50 GHz*, <<https://www.intel.com/content/www/us/en/products/sku/27423/intel-pentium-4-processor-1-50-ghz-256k-cache-400-mhz-fsb/specifications.html>> (2000).
- 16 Zhao, M. *et al.* Characterizing Endurance Degradation of Incremental Switching in Analog RRAM for Neuromorphic Systems. In *2018 IEEE International Electron Devices Meeting (IEDM)*. 20.22.21-20.22.24 (IEEE).
- 17 Kim, S. *et al.* Experimental Demonstration of a Second-Order Memristor and Its Ability to Biorealistically Implement Synaptic Plasticity. *Nano Letters* **15**, 2203-2211 (2015).
- 18 Kim, S. H. *et al.* Experimental Demonstration of a Second-order Memristor and its Ability to Biorealistically Implement Synaptic Plasticity. *Nano Lett.* **15**, 2203-2211 (2015).

- 19 Shin, J. H., Jeong, Y. J., Zidan, M. A., Wang, Q. & Lu, W. D. Hardware Acceleration of Simulated Annealing of Spin Glass by RRAM Crossbar Array. In *2018 IEEE International Electron Devices Meeting (IEDM)*. 3.3.1-3.3.4 (IEEE).
- 20 Cao, Q., Goswami, S. & Karniadakis, G. E. Laplace neural operator for solving differential equations. *Nat. Mach. Intell.* **6**, 631-640 (2024).
- 21 Conklin, A. A. & Kumar, S. Solving the big computing problems in the twenty-first century. *Nat. Electron.* **6**, 464-466 (2023).
- 22 Xia, Q. & Yang, J. J. Memristive crossbar arrays for brain-inspired computing. *Nat. Mater.* **18**, 309-323 (2019).
- 23 Wan, W. *et al.* A compute-in-memory chip based on resistive random-access memory. *Nature* **608**, 504-512 (2022).
- 24 Zhou, Z. *et al.* A Survey on Efficient Inference for Large Language Models. *arXiv preprint arXiv:2404.14294* (2024).
- 25 Yi, S.-i., Kendall, J. D., Williams, R. S. & Kumar, S. Activity-difference training of deep neural networks using memristor crossbars. *Nat. Electron.* **6**, 45-51 (2023).
- 26 McLaren, M., Ferrer, L., Castan, D. & Lawson, A. The Speakers in the Wild (SITW) Speaker Recognition Database. In *Interspeech*. 818-822 (ISCA, 2016).
- 27 Kim, T., Lee, J. & Nam, J. Sample-Level CNN Architectures for Music Auto-Tagging Using Raw Waveforms. In *2018 IEEE International Conference on Acoustics, Speech and Signal Processing (ICASSP)*. 366-370 (IEEE).
- 28 Rao, M. *et al.* Thousands of conductance levels in memristors integrated on CMOS. *Nature* **615**, 823-829 (2023).
- 29 Wang, Z. *et al.* In situ training of feed-forward and recurrent convolutional memristor networks. *Nat. Mach. Intell.* **1**, 434-442 (2019).
- 30 Sharma, G., Umamathy, K. & Krishnan, S. Trends in audio signal feature extraction methods. *Applied Acoustics* **158**, 107020 (2020).
- 31 Turab, M., Kumar, T., Bendechache, M. & Saber, T. Investigating Multi-Feature Selection and Ensembling for Audio Classification. *arXiv preprint arXiv:2206.07511* (2022).
- 32 Chauhan, N., Isshiki, T. & Li, D. Speaker Recognition Using LPC, MFCC, ZCR Features with ANN and SVM Classifier for Large Input Database. In *2019 IEEE 4th International Conference on Computer and Communication Systems (ICCCS)*. 130-133 (IEEE).
- 33 Wang, Y. & Hu, W. in *Proceedings of the 2nd International Conference on Computer Science and Application Engineering* 88 (ACM, 2018).
- 34 Tirumala, S. S., Shahamiri, S. R., Garhwal, A. S. & Wang, R. Speaker identification features extraction methods: A systematic review. *Expert Syst. Appl.* **90**, 250-271 (2017).
- 35 Mada Sanjaya, W. S., Anggraeni, D. & Santika, I. P. Speech Recognition using Linear Predictive Coding (LPC) and Adaptive Neuro-Fuzzy (ANFIS) to Control 5 DoF Arm Robot. *J. Phys: Conf. Ser.* **1090**, 012046 (2018).
- 36 Swain, A. K. & Abdulla, W. Estimation of LPC parameters of speech signals in noisy environment. In *2004 IEEE Region 10 Conference TENCN*. 139-142 (IEEE).
- 37 Pazhanirajan, S. & Dhanalakshmi, P. EEG Signal Classification using Linear Predictive Cepstral Coefficient Features. *International Journal of Computer Applications* **73**, 28-31 (2013).
- 38 Le Gallo, M. *et al.* A 64-core mixed-signal in-memory compute chip based on phase-change memory for deep neural network inference. *Nat. Electron.* **6**, 680-693 (2023).
- 39 Shi, T. *et al.* Stochastic neuro-fuzzy system implemented in memristor crossbar arrays. *Science Advances* **10**, ead13135 (2024).
- 40 Lavasani, H. M., Pan, W., Harrington, B., Abdolvand, R. & Ayazi, F. A 76dB Ω 1.7GHz 0.18 μ m CMOS tunable transimpedance amplifier using broadband current pre-amplifier for high frequency

- lateral micromechanical oscillators. In *2010 IEEE International Solid-State Circuits Conference - (ISSCC)*. 318-319 (IEEE).
- 41 Brooks, L. & Lee, H.-S. A Zero-Crossing-Based 8-bit 200 MS/s Pipelined ADC. *IEEE J. Solid-State Circuits* **42**, 2677-2687 (2007).
- 42 Idros, N., Rosli, A., Abdul Aziz, Z. A., Rajendran, J. & Marzuki, A. A 1.8 V high-speed 8-bit hybrid DAC with integrated rail-to-rail buffer amplifier in CMOS 180 nm. *Microelectronics International* **38**, 46-54 (2021).
- 43 Ambrogio, S. *et al.* Equivalent-accuracy accelerated neural-network training using analogue memory. *Nature* **558**, 60-67 (2018).
- 44 Lanza, M. *et al.* Memristive technologies for data storage, computation, encryption, and radio-frequency communication. *Science* **376**, eabj9979 (2022).
- 45 Chiu, Y. C. *et al.* A 40nm 2Mb ReRAM Macro with 85% Reduction in FORMING Time and 99% Reduction in Page-Write Time Using Auto-FORMING and Auto-Write Schemes. In *2019 Symposium on VLSI Technology*. T232-T233.
- 46 Lee, C. F., Lin, H. J., Lien, C. W., Chih, Y. D. & Chang, J. A 1.4Mb 40-nm embedded ReRAM macro with 0.07 μ m² bit cell, 2.7mA/100MHz low-power read and hybrid write verify for high endurance application. In *2017 IEEE Asian Solid-State Circuits Conference (A-SSCC)*. 9-12 (IEEE).
- 47 Huang, Y.-C. *et al.* in *2024 IEEE International Solid-State Circuits Conference (ISSCC)* 288-290 (IEEE, 2024).
- 48 Wang, L. *et al.* A 14nm 100Kb 2T1R Transpose RRAM with >150X resistance ratio enhancement and 27.95% reduction on energy-latency product using low-power near threshold read operation and fast data-line current stabling scheme. In *2021 Symposium on VLSI Technology*. 1-2 (IEEE).
- 49 Ye, W. *et al.* A 28-nm RRAM Computing-in-Memory Macro Using Weighted Hybrid 2T1R Cell Array and Reference Subtracting Sense Amplifier for AI Edge Inference. *IEEE J. Solid-State Circuits* **58**, 2839-2850 (2023).
- 50 Zhao, L. *et al.* in *2022 IEEE Symposium on VLSI Technology and Circuits (VLSI Technology and Circuits)* 316-317 (IEEE, 2022).
- 51 Yoneda, S. *et al.* in *2018 International Conference on Solid State Devices and Materials* 91-92 (2018).
- 52 Fujitsu. *Fujitsu launches 12Mbit ReRAM—largest memory density in ReRAM family*, <www.fujitsu.com/jp/group/fsm/en/products/reram/spi-12m-mb85as12mt.html> (2022).
- 53 Peters, C., Adler, F., Hofmann, K. & Otterstedt, J. Reliability of 28nm embedded RRAM for consumer and industrial products. In *2022 IEEE International Memory Workshop (IMW)*. 1-3 (IEEE).
- 54 Infineon. *Infineon's new MCU AURIX™ TC4x is about to go into mass production*, <https://www.zenitron.com.tw/tw/news/ifx_mcu_autrix_tc4x> (2024).
- 55 Ielmini, D. & Wong, H. S. P. In-memory computing with resistive switching devices. *Nat. Electron.* **1**, 333-343 (2018).
- 56 Yao, P. *et al.* Fully hardware-implemented memristor convolutional neural network. *Nature* **577**, 641-646 (2020).
- 57 Lee, S., Sohn, J., Jiang, Z., Chen, H.-Y. & Wong, H.-S. P. Metal oxide-resistive memory using graphene-edge electrodes. *Nat. Commun.* **6**, 8407 (2015).
- 58 Xu, X. *et al.* Fully CMOS compatible 3D vertical RRAM with self-aligned self-selective cell enabling sub-5 nm scaling. In *2016 IEEE Symposium on VLSI Technology*. 1-2 (IEEE, 2016).
- 59 Wang, R. *et al.* Implementing in-situ self-organizing maps with memristor crossbar arrays for data mining and optimization. *Nat. Commun.* **13**, 2289 (2022).
- 60 Wei, J. *et al.* A neuromorphic core based on threshold switching memristor with asynchronous address event representation circuits. *Sci. China Inform. Sci.* **65**, 122408 (2021).

- 61 Duan, Q. *et al.* Spiking neurons with spatiotemporal dynamics and gain modulation for monolithically integrated memristive neural networks. *Nat. Commun.* **11**, 3399 (2020).
- 62 Cai, F. *et al.* A fully integrated reprogrammable memristor–CMOS system for efficient multiply–accumulate operations. *Nat. Electron.* **2**, 290-299 (2019).
- 63 Wang, Z. *et al.* Reinforcement learning with analogue memristor arrays. *Nat. Electron.* **2**, 115-124 (2019).
- 64 Wang, C. *et al.* Scalable massively parallel computing using continuous-time data representation in nanoscale crossbar array. *Nat. Nanotechnol.* **16**, 1079-1085 (2021).
- 65 Chen, B. *et al.* Ge-Based Asymmetric RRAM Enable $8F^2$ Content Addressable Memory. *IEEE Electron Device Lett.* **39**, 1294-1297 (2018).
- 66 Li, C. *et al.* Analog content-addressable memories with memristors. *Nat. Commun.* **11**, 1638 (2020).
- 67 Wang, X. *et al.* A 4T2R RRAM Bit Cell for Highly Parallel Ternary Content Addressable Memory. *IEEE Trans. Electron Devices* **68**, 4933-4937 (2021).
- 68 Coleman, T. F. & Li, Y. An Interior Trust Region Approach for Nonlinear Minimization Subject to Bounds. *SIAM J. Optim.* **6**, 418-445 (1996).
- 69 Kim, S., Choi, S. & Lu, W. Comprehensive Physical Model of Dynamic Resistive Switching in an Oxide Memristor. *ACS Nano* **8**, 2369-2376 (2014).
- 70 Leighton, P. A. Electronic Processes in Ionic Crystals (Mott, N. F.; Gurney, R. W.). *J. Chem. Educ.* **18**, 249 (1941).
- 71 Panzer, M. A. *et al.* Thermal Properties of Ultrathin Hafnium Oxide Gate Dielectric Films. *IEEE Electron Device Lett.* **30**, 1269-1271 (2009).
- 72 Landon, C. D. *et al.* Thermal transport in tantalum oxide films for memristive applications. *Appl. Phys. Lett.* **107**, 023108 (2015).
- 73 Capron, N., Broqvist, P. & Pasquarello, A. Migration of oxygen vacancy in HfO_2 and across the HfO_2 / SiO_2 interface: A first-principles investigation. *Appl. Phys. Lett.* **91**, 192905 (2007).
- 74 Larentis, S., Nardi, F., Balatti, S., Gilmer, D. C. & Ielmini, D. Resistive Switching by Voltage-Driven Ion Migration in Bipolar RRAM—Part II: Modeling. *IEEE Trans. Electron Devices* **59**, 2468-2475 (2012).
- 75 Pahinkar, D. G. *et al.* in *2019 18th IEEE Intersociety Conference on Thermal and Thermomechanical Phenomena in Electronic Systems* 219-225 (IEEE, 2019).
- 76 Roldán, J. B. *et al.* Variability in Resistive Memories. *Advanced Intelligent Systems* **5**, 2200338 (2023).
- 77 Gonzalez-Cordero, G., González, M. B., Jiménez-Molinos, F., Campabadal, F. & Roldán, J. B. New method to analyze random telegraph signals in resistive random access memories. *Journal of Vacuum Science & Technology B* **37**, 012203 (2019).
- 78 González-Cordero, G. *et al.* Neural network based analysis of random telegraph noise in resistive random access memories. *Semicond. Sci. Technol.* **35**, 025021 (2020).
- 79 Ruiz-Castro, J. E., Acal, C., Aguilera, A. M. & Roldán, J. B. A Complex Model via Phase-Type Distributions to Study Random Telegraph Noise in Resistive Memories. *Mathematics* **9**, 390 (2021).
- 80 Pazos, S. *et al.* High - Temporal - Resolution Characterization Reveals Outstanding Random Telegraph Noise and the Origin of Dielectric Breakdown in h-BN Memristors. *Adv. Funct. Mater.* **34**, 2213816 (2024).

REVIEWERS' COMMENTS

Reviewer #1 (Remarks to the Author):

The reviewer is satisfied with the author's reply. I recommend this paper for publication.

Reviewer #2 (Remarks to the Author):

The authors have addressed the requirements included in my revision. I recommend the paper for publication.

Reviewer #3 (Remarks to the Author):

The authors have properly addressed my previous concerns. Now it is recommended to publish as it is.